# Identification of covalent modifications regulating immune signaling complex composition and phenotype

Annika Frauenstein[1], Stefan Ebner[1], Fynn M Hansen[2], Ankit Sinha[2], Kshiti Phulphagar[1], Kirby Swatek[3], Daniel Hornburg[4] iD, Matthias Mann[2] iD & Felix Meissner[1,5,*] iD

## Abstract

Cells signal through rearrangements of protein communities governed by covalent modifications and reversible interactions of distinct sets of proteins. A method that identifies those post-transcriptional modifications regulating signaling complex composition and functional phenotypes in one experimental setup would facilitate an efficient identification of novel molecular signaling checkpoints. Here, we devised modifications, interactions and phenotypes by affinity purification mass spectrometry (MIP-APMS), comprising the streamlined cloning and transduction of tagged proteins into functionalized reporter cells as well as affinity chromatography, followed by MS-based quantification. We report the time-resolved interplay of more than 50 previously undescribed modification and hundreds of protein–protein interactions of 19 immune protein complexes in monocytes. Validation of interdependencies between covalent, reversible, and functional protein complex regulations by knockout or site-specific mutation revealed ISGylation and phosphorylation of TRAF2 as well as ARHGEF18 interaction in Toll-like receptor 2 signaling. Moreover, we identify distinct mechanisms of action for small molecule inhibitors of p38 (MAPK14). Our method provides a fast and cost-effective pipeline for the molecular interrogation of protein communities in diverse biological systems and primary cells.

**Keywords** mass spectrometry; posttranslational modifications; protein interactions; proteomics; signaling networks
**Subject Categories** Immunology; Proteomics
**Mol Syst Biol. (2021) 17: e10125**

## Introduction

Cellular functions rely on complex molecular networks that are mainly composed of proteins (Seet *et al*, 2006; Pan *et al*, 2012). Cell type- and context-specific functions require a tight orchestration of signaling, and their dysregulation is often associated with pathology (Arkin *et al*, 2014). Experimental approaches that quantitatively capture the mechanisms of dynamic signaling networks are therefore highly valuable for establishing causal links to cellular phenotypes and the development of strategies for targeted interference.

Traditionally, the analysis of signal transduction mechanisms has focused on proteins with annotated functions in a given biological pathway. Pathway activation is probed with antibodies that determine the abundance of posttranslational modifications (PTMs) or interaction of selected proteins (protein–protein interactions, PPIs). Although valuable for testing pre-defined molecular states of selected proteins, the utility of this approach is limited by antibody availability, and prior knowledge of molecular and functional relationships. While employing antibodies would be applicable irrespective of the cell type, the discrimination of direct and indirect, as well as antibody-bound and bait-bound protein interactors, is often challenging because of limited antibody specificity (Marcon *et al*, 2015). Conversely, while epitope tagging of selected proteins provides an alternative that guarantees specific enrichment with stable background binders—a defined set of proteins adhering to the affinity matrix—not all cell types are amenable to efficient genetic manipulations. An optimal strategy would therefore combine efficient and antibody-independent enrichment with universal applicability for eukaryotic cell types.

Mass spectrometry (MS)-based proteomics allows the detection of PTM and PPIs without prior knowledge. In recent years, MS-based proteomics has advanced tremendously and transitioned from identifying only a few proteins to comprehensively quantifying cellular proteomes and identifying modified proteins and protein interactions on a large scale (Larance & Lamond, 2015; Aebersold &

1 Experimental Systems Immunology, Max Planck Institute of Biochemistry, Martinsried, Germany
2 Department of Proteomics and Signal Transduction, Max Planck Institute of Biochemistry, Martinsried, Germany
3 Department of Molecular Machines and Signaling, Max Planck Institute of Biochemistry, Martinsried, Germany
4 Department of Genetics, School of Medicine, Stanford University, Stanford, CA, USA
5 Institute of Innate Immunity, Department of Systems Immunology and Proteomics, Medical Faculty, University of Bonn, Bonn, Germany
*Corresponding author (lead contact). Tel: +49 22828751210; E-mail: felix.meissner@uni-bonn.de

Mann, 2016). As such, it provides systems-wide views of cellular states with immense discovery potential, as indicated by large-scale efforts to map the entire interactomes in yeast (Gavin et al, 2002; Ho et al, 2002; Krogan et al, 2006), drosophila (Guruharsha et al, 2011), and human (Hein et al, 2013; Hein et al, 2015; Huttlin et al, 2015), kinase and phosphatase interactomes (Gingras et al, 2007; Couzens et al, 2013; Yao et al, 2017; Buljan et al, 2020) as well as global views of specific PTMs (Choudhary et al, 2009; Humphrey et al, 2015; Lescarbeau et al, 2016; Liu et al, 2018).

Although it is well appreciated that the interplay of PTMs and PPIs determines how biological responses are regulated, MS-based technologies are almost always used to investigate PTMs and PPIs separately, and rely on distinct biochemical and analytical strategies. Hence, the analysis of PPIs is bait-centric, and selected proteins are affinity-enriched together with their interacting partners (Paul et al, 2011). By contrast, PTM analysis generally focuses on a single modification type (e.g., phosphorylation), wherein modified peptides of all cellular proteins are affinity-enriched. Alternatively, in order to classify PTMs on specific proteins, affinity purification mass spectrometry (APMS) approaches with stringent washing and lysis conditions have been performed at the expense of PPI elucidation (Stutz et al, 2017; Pankow et al, 2019; Karayel et al, 2020). Consequently, these two molecular modes of protein regulations are experimentally disconnected, hampering the discovery of the relationships between PTMs and PPIs in cellular signaling pathways. Furthermore, easy methods to simultaneously monitor different PTM types in a single sample are missing. Conventional enrichment strategies for distinct PTMs vary widely, and hence, mapping of multiple PTMs usually requires several sequential or parallel biochemical steps. This requires large amounts of starting material and results in low-sample throughput, while comprehensiveness is still limited as the enrichment strategies are tailored toward known biochemical properties of selected PTMs. A method that would capture in an unbiased manner all detectable PTMs in protein complexes of interest is therefore needed so as to comprehensively pinpoint molecular signaling checkpoints in complex biological systems.

The functional evaluation of emerging PTMs and PPIs is a common bottleneck in systems-wide discovery approaches. While initial screens are often performed in an experimental system that closely resembles cellular physiology, experimental validation of hits among all discovered candidates frequently relies on loss- or gain-of-function experiments in cell lines to achieve the necessary throughput. However, desirable would be an experimental setup that facilitates both discovery and validation in primary cells.

To develop a method for the systematic dissection of cellular signaling checkpoints by simultaneous PTM and PPI mapping in one experiment, we devised a streamlined pipeline—Modifications, Interactions and Phenotypes by APMS (MIP-APMS). We evaluated and technically optimized all steps of MIP-APMS, comprising (i) the epitope tagging of proteins of interest and mammalian cell transduction, (ii) affinity purification conditions for optimal interaction network and PPI enrichment, (iii) followed by MS-based PTM and PPI quantification and identification, and (iv) ultimate biochemical and phenotypic validation of interactors and PTMs in primary human immune cells. Integration of multiple MIP-APMS experiments generates dynamic signal transduction networks and pinpoints time-resolved co-regulations of PTMs and PPIs in

sequential signal transduction steps. We show the discovery potential of our pipeline by interrogating dynamically assembling protein communities in human monocyte immune signaling using Toll-like receptor (TLR) 2 activation and MAP kinase MAPK14 inhibition as paradigms. Our screen encompassing 19 protein complexes identified more than 50 previously undescribed PTMs, including phosphorylation, acetylation, methylation, ISGylation as well as other less well-described chemical modifications and elucidated an interaction network spanning more than 300 PPIs. We used the modular concept of MIP-APMS to test emerging data-driven hypotheses to validate PTMs and PPIs regulating immune signaling in reporter and primary cells. In this way, MIP-APMS enables the streamlined validation of crosstalk between different layers of protein regulation with broad applicability.

# Results

## Experimental and proteomics strategy for interrogating dynamic signal transduction networks

We devised MIP-APMS for the identification and perturbation of the functional checkpoints of cellular signaling pathways. MIP-APMS involves the following four stages with the indicated time frames (see Graphical Abstract, Figs 1 and EV1A):

1   Cloning of genes encoding epitope-tagged proteins and transduction of specialized cell types.
2   Streamlined quantification of various types of PTMs together with PPIs.
3   Implementation of an analytical strategy to pinpoint genetic or pharmacological signaling network perturbations.
4   Direct biochemical and functional evaluation of novel biological regulations in the same experimental system.

### Universal cloning and transduction strategy

To enable interrogation of signaling cascades, we employed a cost-effective method for epitope tagging of proteins with a restriction enzyme-free approach, called restriction enzyme-free seamless ligation cloning extract (SLiCE) cloning (Zhang et al, 2012). A modified weak phosphoglycerate kinase (PGK) promoter controls the expression of the GOIs, which are flanked by attL sites. Thereby, our vector system is compatible with commercial DNA assembly cloning strategies such as the NEBuilder platform or Gateway, which had been used before (Lambert et al, 2014). As shown previously, employing lentiviral transduction for amphotrophic gene transfer extends the scope from readily transfectable cell lines, e.g., human embryonic kidney (HEK) cells, to non-dividing and terminally differentiated cells of primary origin (Huttlin et al, 2015; Samavarchi-Tehrani et al, 2018). In particular for application with primary immune cells, transduction is advantageous as other methods can activate innate immune signaling pathways and induce cell death (Fernandes-Alnemri et al, 2009; Hornung et al, 2009; Gaidt et al, 2017). As a relevant and challenging experimental model system, we chose human monocytes, because these cells are not easily transfectable and execute a broad spectrum of cellular programs by the dynamic intracellular propagation of molecular signals downstream of cell surface receptors. For method development and

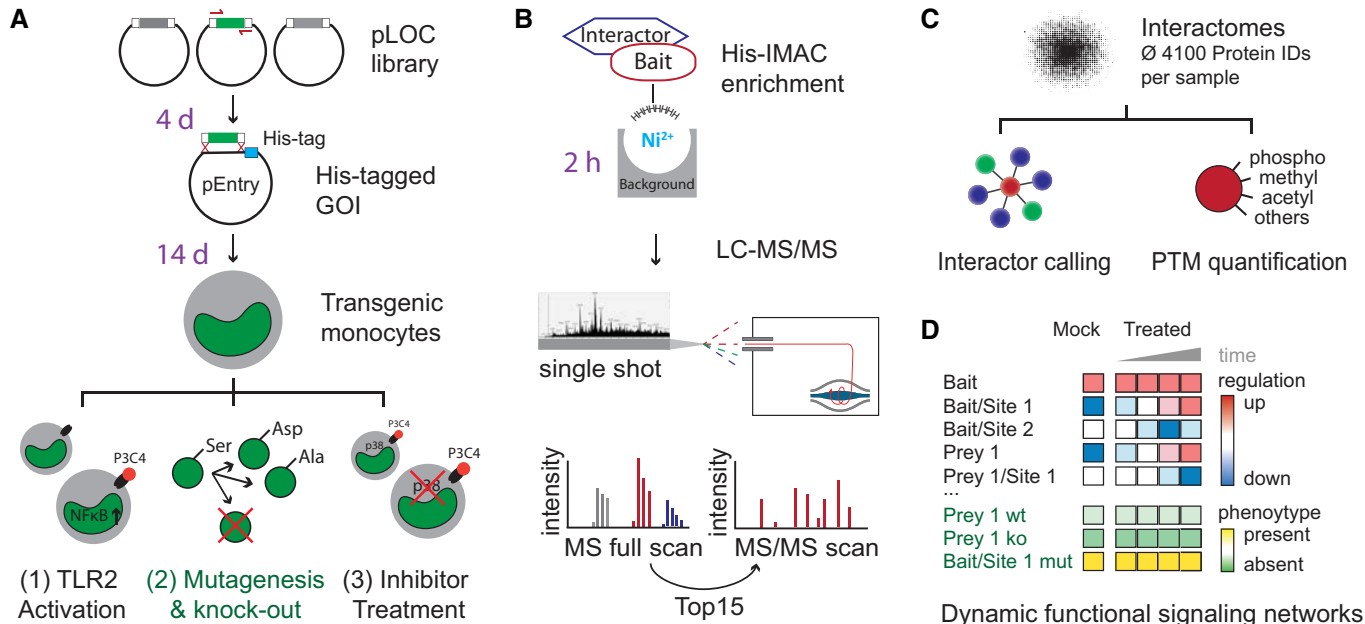

**Figure 1.  Experimental strategy for interrogating dynamic signal transduction networks with MIP-APMS.**

A   Generation of polyclonal transgenic cell lines by lentiviral transduction of genes encoding epitope-tagged wild-type or variant bait proteins. Analysis of PTMs and PPIs upon cellular activation (exemplified for the TLR1/2 activation by the agonist Pam3CSK4, P3C4), or pharmacological signal perturbation (exemplified by MAPK14 inhibitors). Time frames for the individual steps are indicated in violet.

B   Single-step His-IMAC affinity enrichment and single-run liquid chromatography–tandem mass spectrometry (LC-MS/MS).

C   Investigation of dynamic signal network topologies by simultaneous analysis of PPIs and multiple different PTMs. The numbers indicate analysis steps.

D   Schematic representation of PTM and PPI dynamics as a function of cellular phenotypes. Wt, wild type; ko, knockout; mut, mutation.

Data information: See also Figs EV1–EV4.

phenotypic screening, we employed the monocytic cell line U937 and validated our results with primary cells. We achieved 92 (± 5) % cellular transduction efficiency after antibiotic marker selection (Fig EV1B). We further demonstrated the universality of our approach with primary human macrophages differentiated from peripheral monocytes (Fig EV1C and D, Table EV1) and primary human T cells (Fig EV1E, Table EV1).

We carefully characterized the functional properties of generated cell lines: The average copy numbers of the endogenous protein counterparts to the tagged proteins were 3.1 million per cell, increasing only slightly to an average of 4.3 million copies upon transduction (Fig EV2A). Importantly, global protein expression levels remained stable within cells upon expression of epitope-tagged bait proteins (Fig EV2B and C). We specifically confirmed that transduced cells exhibit no background immune activation by assessing expression levels of proteins involved in immune- and infection-associated pathways (Fig EV2D and E) and retain their full activation potential by assessing NFkB activity with Luciferase reporter assays (Fig EV2F).

### Simultaneous enrichment of PTMs and PPIs

Next, to study interdependency of PTMs and PPIs in signaling cascades, we evaluated biochemical enrichment strategies for epitope-tagged proteins with MS-based proteomics using high-performance liquid chromatography (HPLC) coupled to a linear quadrupole Orbitrap mass spectrometer (Q Exactive HF, Thermo)

operated in a data-dependent acquisition mode (Fig 1B) (Scheltema *et al*, 2014). We systematically compared typical short epitope tags: Flag-tag (Hopp *et al*, 1988), Strep-tag (Schmidt & Skerra, 2007), and polyhistidine tag (Hochuli *et al*, 1988). To quantitatively compare epitope tag-based enrichments, identification and label-free quantification (LFQ) were performed in the MaxQuant environment (Cox & Mann, 2008). While > 1,000 proteins were shared between all three enrichments (Fig EV3B), His-IMAC enrichment identified more background binding proteins. Exemplified for MAPK14, our results show high overlap of known interactors for Strep-tag and His-tag IPs with on-bead digestion, whereas Flag-tag and Strep-tag with elution yielded lower numbers of significant interactors (Fig EV3A, Table EV1). Notably, the highest median bait protein sequence coverage (Fig EV3C), highest intensity of MAPK14 (Fig EV3D), and highest number of significantly interacting proteins were achieved with His-IMAC.

Accordingly, we incorporated His-IMAC in the MIP-APMS protocol and further optimized the protocol for high bait enrichment and high-sequence coverage by titrating imidazole concentration in lysis and wash buffers, respectively (Fig EV3E and F). Following method optimization, the respective bait proteins were among the highest enriched proteins after MIP-APMS (Fig EV3G, Table EV1). We achieved a median sequence coverage of 70% for bait proteins (Fig EV3H), opening up the possibility of directly identifying and quantifying PTMs, such as phosphorylation, acetylation, or methylation as well as other less well-studied covalent protein modifications on

any bait protein (Fig 1C). Differently modified peptides were not analyzed separately, as in typical proteomics workflows, but instead the selected enriched proteins represented all present and detectable proteoforms. This made it possible to simultaneously quantify the differently modified and unmodified versions of peptides. MIP-APMS enables the efficient and cost-effective and robust analysis of PTMs and PPIs in a single experiment.

### Dynamic signaling network analysis

To study how signaling networks rearrange upon cellular activation, we integrated quantitative PTM and PPI information from multiple MIP-APMS experiments. This enabled quantitative analysis of sequential steps in signal transduction, since it allowed for dynamic PTM and PPI crosstalk to be resolved providing a basis to identify molecular switches in signal transduction networks. We observed enrichments and de-enrichment of prey proteins in protein complex of interest and also dynamically regulated PTMs on both bait and prey proteins (Fig 1D, regulation up/down).

### Biochemical and functional evaluation of novel biological regulations in the same experimental system

To validate our findings in follow-up studies, we employed the same experimental system used for discovery. We investigated the alterations in dynamic signaling networks of proteins mutated on single amino acid sites discovered in our study. Furthermore, by transforming our model system into NFkB reporter cells, we were able to reveal functional effects on NFKB activation of novel PTMs and PPIs by CRISPR-Cas9-mediated gene knockout and site-specific gene mutations, respectively (Fig 1D, phenotype, Fig 1A). As described below in more detail, we were able to derive functional molecular checkpoints in monocyte signal transduction networks.

### Signaling networks of kinases, signaling adapters, and caspases in monocytes

We tested our MIP-APMS approach by interrogating the molecular composition of protein communities in mammalian cells *in situ*. Specifically, we investigated innate immune signaling complexes, assembled various protein classes, such as kinases, caspases, and tumor necrosis factor (TNF) receptor-associated factors (TRAFs) in human monocytes.

We generated 19 transgenic monocytic U937 cell lines and analyzed them with MIP-APMS, as described above. This identified and quantified an average of 4,106 proteins per measurement, including non-specifically binding proteins as expected for non-stringent APMS conditions (Trinkle-Mulcahy *et al*, 2008; Rees *et al*, 2011). We observed high median intra-bait and inter-bait Pearson correlations (> 0.9) between biological replicates (Fig EV3I) and between different cell lines (Fig EV3J). This highlights the overall reproducibility of the devised workflow. To discriminate specifically interacting proteins from background binders common to all baits, we compared enrichments from single vs. all other cell lines with a standard statistical test (two-sided *t*-test) at a stringent false discovery rate (FDR) of 1% to correct for multiple hypothesis testing (Hein *et al*, 2015; Keilhauer *et al*, 2015; Hubel *et al*, 2019). This resulted in a small fraction of significantly interacting proteins (378 proteins in total, with a median of 16 interactors per bait) compared to a large proportion of background binders (Table EV1, Fig EV4A). Notably, distinct protein intensity differences and *P*-values clearly distinguish specific bait and prey from unspecific background proteins (Fig EV4B and C). MIP-APMS prioritizes bait-specific preys, as proteins enriched in multiple experiments—including interconnected interactors—show lower enrichment differences and *P*-values (Fig EV4D) by unbiased statistical interactor calling (see Materials and Methods). We compared our LFQ intensity and *t*-test-based strategy to the results of the SAINT algorithm (spectral count based) exemplary for MAPK14 and identified largely similar interactors (Fig EV4E).

The identified interactors included previously described as well as novel proteins (Fig 2A, Table EV2). Unsupervised hierarchical clustering of label-free quantification (LFQ) intensity profiles of the significant interactors grouped specific interactors of corresponding bait proteins together (Fig EV4F). To determine the topology of the detected protein interaction network, we assembled proteins according to shared interactors. This enabled the identification of signaling hubs through common connections of bait and prey proteins that clustered together in the network (Fig 2B). The analysis recapitulated many known interactions, including the TRAF2–BIRC2 CORUM complex (Ruepp *et al*, 2010) involving the binary interaction of TRAF2 and BIRC2, supplemented by such players as TRAF1, TBK1, TANK, and IKBKE (Wu *et al*, 2005). Some TRAF2 interactors, such as RIPK1, CASP8, and TNF (Hsu *et al*, 1996), were not detected in this experiment perhaps because they require distinct context-dependent cellular activation, e.g., through TNFR. These

**Figure 2. Dissection of protein signaling networks in human monocytes using MIP-APMS.**

A  Percentage of previously described interactors (green) and novel interactors (blue), and the count of significant interactors (FDR < 0.01, enrichment > 2) per bait protein (median interactor count: 16).

B  Protein–protein interaction network of clustered interaction data. Edges indicate interactions, with shared interactions connecting the individual MIP-APMS experiments. Red nodes correspond to bait proteins, green nodes to interactors reported in the literature, and blue nodes to novel interactors.

C  Numbers of acetylations, methylations, and phosphorylations identified on bait proteins and interactors.

D  Percentage of PTMs identified on bait proteins and interactors.

E  Numbers of PTMs on bait proteins/interactors of individual pull-downs.

F  Numbers of novel and described (Uniprot-annotated) acetylations, methylations, and phosphorylations.

G  Unsupervised clustering (Pearson correlation) of the z-scored intensity profiles of all PPIs (357) and PTMs (37) upon TLR2 activation, partitioned in seven clusters.

H  Dynamic profiles of co-regulated PTMs and PPIs with close network proximity, from the indicated clusters; median z-scored intensity of each time point (blue: median, gray: confidence interval = 0.95, method: loess); *n*, number of proteins in clusters 1–7. Selected proteins from each cluster are indicated, with the bait proteins in parentheses.

Data information: See also Fig EV3, Table EV2 for PPIs, and Table EV3 for PTMs.

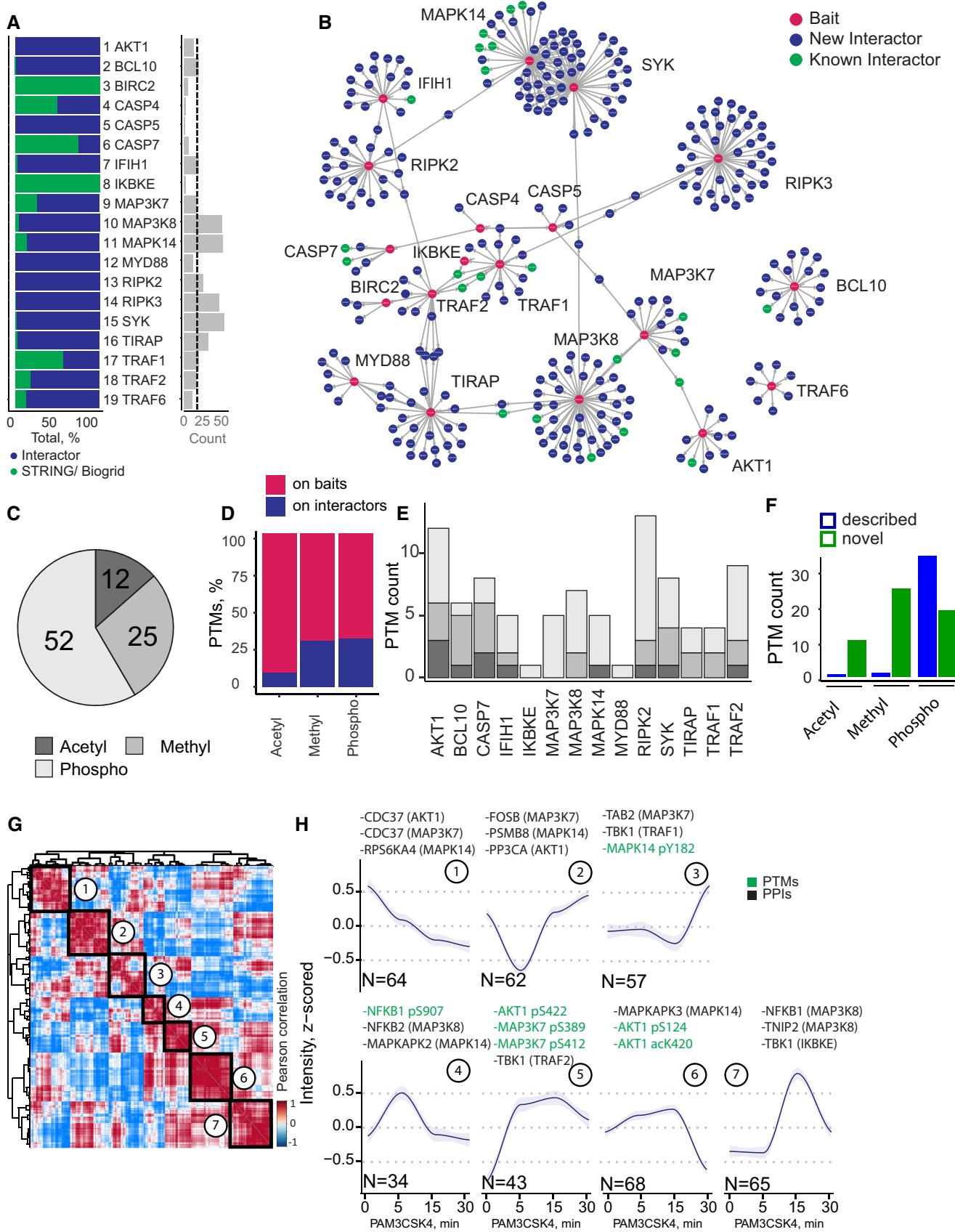

Figure 2.

observations validated the utility of MIP-APMS for the interrogation of intracellular signaling networks.

To identify and quantify PTMs in the same experimental setup, we re-analyzed our data with phosphorylation, acetylation, and methylation as variable posttranslational modifications. Even though we did not enrich for PTMs, we identified and quantified PTMs spanning phosphorylations, acetylations, and methylations on baits as well as prey proteins (88 PTMs on 19 bait proteins). Phosphorylation was the most abundant PTM in the dataset (52 sites), followed by methylation (25 sites) and acetylation (11 sites) (Fig 2C). While the majority of PTMs were detected on bait proteins, some (31 PTMs on 10 proteins) were also detected on prey proteins (26% of all known PTMs; Fig 2D). A remarkable 74% of the studied bait proteins or their respective interactors were posttranslationally modified, with some proteins, e.g., AKT1 and RIPK2, harboring more than 10 PTMs (Fig 2E). Notably, MIP-APMS identified 52 previously undescribed PTMs, in particular methylation and acetylation sites (Fig 2F). Furthermore, an unbiased analysis of covalent peptide modifications using the dependent peptide algorithm in MaxQuant, the string-based search algorithm Taggraph—based on a de novo search in PEAKS—and MS Fragger (Devabhaktuni et al, 2019)—revealed a series of less well-described covalent modifications on MAPK14 (Fig EV4G). Twenty-six modifications were shared between search engines (2.3% of all modifications for dependent peptides, 1.5% PEAKS/Taggraph, and 0.9% MS Fragger, Fig EV4H). Out of these 26 modifications, six were reproducibly identified and quantified in all replicates (Fig EV4I). To distinguish biologically regulated from other—for example—sample preparation-introduced modifications, we quantified the identified modifications upon cell activation with specific searches in MaxQuant. Notably, only MAPK14 phosphorylation was differentially regulated between conditions. Moreover, acetylation, methylation, and phosphorylation detected on TRAF2, MAPK14, and MAP3K7 with specific searches were missed by open searches (dependent peptides of MaxQuant and PEAKS/Taggraph; Fig EV4J). This demonstrates that MIP-APMS can discover novel PTMs in signaling complexes; however, comparisons across search engines and confirmation with specific search strategies are advisable to increase confidence.

To capture the dynamics of cellular signal transduction, we next analyzed how the intracellular networks rearrange upon cellular activation via cell surface receptors. We stimulated cells via TLR2, as this pattern recognition receptor is prominently expressed in monocytes and induces a robust pro-inflammatory program that involves activation of the transcription nuclear factor (NF) κB pathway (Oliveira-Nascimento et al, 2012; Rieckmann et al, 2017). We analyzed the dynamic signaling networks downstream of TLR2 using time course experiments in biological quadruplicates. Upon stimulation with the lipopeptide Pam3CysK (PAM3CSK4), cellular signaling was activated (Fig EV4K), and stable vs. dynamic PTMs and PPIs could be distinguished. Because of the short time frame of kinetic investigations (within 30-min post-cellular activation), we did not normalize protein levels to expression-induced protein abundance changes. On average, we detected two statistically significant dynamic PPIs and one dynamic PTMs per bait (Fig EV5A; Table EV2 and EV3). Our data suggest that phosphorylation is the most

---

**Figure 3.  N-Terminal phosphorylation of TRAF2 and ISG15 is dynamic functional regulators downstream of TLR2.**

A   Volcano plot representing the interactome of TRAF2 (measured 15× in biological replicates) compared against all other pull-downs in the control group. The results of the t-tests are represented in volcano plots, which show the protein enrichment versus the significance of the enrichment. Numbers indicate enrichment ranks with the heatmap labels of (C) serving as the legend. Significant interactors of TRAF2 (two-tailed t-test, FDR < 0.01, enrichment > 4) are colored in blue (novel interactors) and green (known interactors).

B   Interactors of TRAF2 (blue: novel interactors, green: known interactors) with interconnecting proteins between different baits colored in gray.

C   Hierarchical clustering of significant interactors of TRAF2 upon activation with significant hits in at least one time point denoted with an asterisk. Cell activation was performed for 5, 15, and 30 min with the TLR2 ligand PAM3CSK4 (P3C4).

D   Intensity profile of the TRAF2 interactor TANK upon activation, normalized to TRAF2 bait LFQ intensity.

E   Hierarchical clustering of the TRAF2 PTMs (acetylation, methylation, and phosphorylation) upon activation, with significant hits (t-test) in at least one time point denoted with an asterisk.

F   Intensity profile of the phosphorylation of TRAF2 on Thr7 upon activation, normalized to TRAF2 bait intensity. Central band of the boxplot shows the median, boxes represent the IQR, 3 biological replicates were performed for UT, and 4 biological replicates were performed for additional time points. P-values were calculated by t-test. Asterisks indicate significant differences. *P-value < 0.05.

G   Intensity profile of TRAF2 interactors ISG15 and TRAF1 in different TRAF2 phospho-variants, normalized to TRAF2 wild-type intensities. Central band of the boxplot shows the median, boxes represent the IQR, and 4 biological replicates were performed for every condition. P-values were calculated by t-test. Asterisks indicate significant differences **P-value < 0.01, ***P-value < 0.001.

H   Induction of NFκB determined based on luciferase luminescence in TLR2-activated U937 NFκB reporter cells transfected with genes encoding different TRAF2 phospho-variants. Bar represents the median, error bars represent the standard deviation, and 4 biological replicates were performed for additional time points. P-values were calculated by t-test. Asterisks indicate significant differences. ***P-value < 0.001.

I   MS/MS Spectrum containing GlyGly modification K320 on TRAF2 after GlyGly enrichment on TRAF2 MIP-APMS.

J   Differences and P-values of ISG15 intensity in TRAF2 K->R mutants compared against TRAF WT

K   Intensity profile of TRAF2 interactors ISG15 and TANK in TRAF2 K->R mutants, normalized to TRAF2 wild-type intensities. Central band of the boxplot shows the median, boxes represent the IQR, and four biological replicates were performed for every condition. P-values were calculated by t-test. Asterisks indicate significant differences. ***P-value < 0.001.

L   Induction of NFκB determined based on luciferase luminescence in TLR2-activated U937 NFκB reporter cells transfected with genes encoding different TRAF2 K→R mutants (each bar represents a mean from three independent measurements; error bars represent the standard deviation; ***P-value < 0.001).

M   Intensity profile of TRAF2 interactors ISG15 and TANK in TRAF2-K389R and S11D mutants in human primary macrophages. Central band of the boxplot shows the median, boxes represent the IQR, and three biological replicates were performed for every condition. P-values were calculated by t-test. Asterisks indicate significant differences **P-value < 0.01, *P-value < 0.05, ***P-value < 0.001.

Data information: Experiments in (A–L) were performed in U937 cell lines. Gray boxes indicate missing values. IQR stands for interquartile range and represents the 25th to 75th percentile. See also Table EV1–EV4.

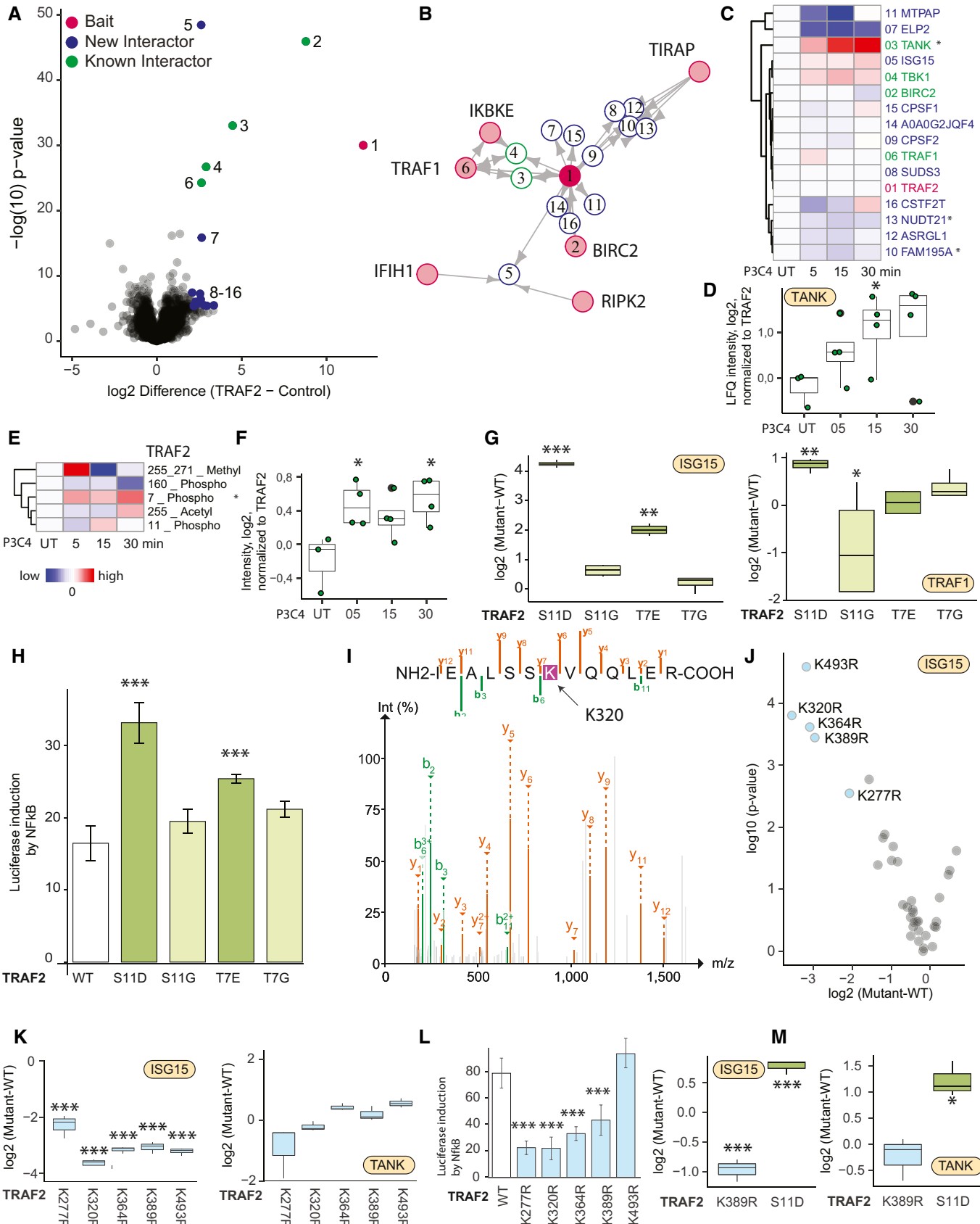

**Figure 3.**

dynamic PTM in the tested setting, followed by methylation (Fig EV5B).

Next, to study PTM and PPI interdependency, we correlated all PTM and PPI intensities and clustered them unbiasedly over the time course of TLR2 activation (Fig 2G, Table EV3). We detect the dynamic co-regulation on both molecular layers (PPIs and PTMs), identifying correlating and anti-correlating PTMs and PPIs during signaling pathway activation. We identified seven clusters with distinct kinetics, some peaking early (Fig 2H, Cluster: 4,5) and others late (Fig 2H, Cluster: 7) upon pathway activation, as well as up- (Fig 2H, Cluster: 4,5,6,1) vs. down-regulated (Fig 2H, Cluster: 1,2,3) PTMs and PPIs. Interestingly, interactors identified in more than one MIP-APMS experiment (e.g., CDC37: Cluster 1) were in close network proximity. Our approach facilitated an unbiased discovery of time-resolved molecular connections between dynamic PTMs and PPIs, exemplified by the correlated interaction of MAP3K8 interactors (NFKB1, NFKB2) and NFκB1 phosphorylation (Cluster 4,7), or the anti-correlated phosphorylation of the C-terminal kinase domain of AKT1 and the interaction with CDC37 (Clusters 1, 5). This demonstrates that the sensitivity and robustness of MIP-APMS enable the simultaneously determination of cellular signaling network rearrangements by PPIs, PTMs, and their interplay. We conclude that MIP-APMS is sufficiently sensitive and robust to capture dynamic signaling networks in mammalian cells *in situ*. It generates highly reproducible data that may be used for the discovery of novel dynamic PTMs in signal transduction cascades, and simultaneous evaluation of multiple PTMs and PPIs in signaling networks.

### Dynamic phosphorylations and ISGylations regulate TRAF2 downstream of TLR2

We next evaluated MIP-APMS for the discovery of novel molecular checkpoints in intracellular immune signaling. We focused on significantly regulated PPIs (FDR < 0.01) and PTMs (*P*-value < 0.05) identified for TRAF2 and MAP3K7, and examined their biochemical and phenotypic relevance through network perturbations mediated by gain- and loss-of-function mutations.

TRAF2 is a central adaptor protein in TNF signaling and regulates pro-inflammatory cytokine production through NFκB and JNK signaling pathways (Borghi *et al*, 2016). As described above, the MIP-APMS analysis confirmed previously reported TRAF2 interactors, such as TNF receptor-associated factor TRAF1, baculoviral IAP repeat-containing protein BIRC2 (cIAP2), TRAF family member-associated NFκB activator TANK, and serine/threonine-protein kinase TBK1. In addition, we identified ELP2 and ISG15 as novel components of the TRAF2 complex (Fig 3A and B) and TANK, a negative regulator of TRAF2 (Cheng & Baltimore, 1996), as dynamically recruited to the TRAF2 complex. By contrast, the majority of other TRAF2 interactors remained unchanged upon activation (Fig 3 C and D). While most other PTMs remained unchanged upon signal pathway activation, the analysis revealed dynamic N-terminal phosphorylations on Thr7 and Ser11 of TRAF2 (Fig 3E and F). Thus, the interactome and PTMs of TRAF2 are dynamically regulated upon NFκB activation via TLR2.

To test whether these dynamic N-terminal phosphorylations affected the composition and function of the TRAF2 protein complex, we used the MIP-APMS streamlined workflow to generate

protein phospho-variants, in which specific Ser or Thr residues were changed to Gly, or to Asp/Glu to mimic phosphorylation. We probed the resulting signaling network rearrangements using MIP-APMS and found a specific enrichment for ubiquitin-like protein ISG15 and TRAF1 by the phospho-mimetic TRAF2 variants compared to wild-type TRAF2 (Fig 3G; Table EV4). These data suggest that N-terminal phosphorylation of TRAF2 at both Thr7 and Ser11 stabilizes a protein complex with ISG15 and TRAF1.

To further assess the functional relevance of the N-terminal TRAF2 phosphorylation on cellular regulation, we introduced the phospho-mimetic and phospho-dead TRAF2 variants into NFκB reporter monocyte cell lines. TLR2-induced NFκB activation was elevated with TRAF2 N-terminal phospho-mimetics, whereas the phospho-dead variants showed activation comparable to that of wild-type TRAF2, indicating that N-terminal phosphorylation boosted downstream signal transduction (Fig 3H).

ISG15 is a ubiquitin-like protein that covalently modifies target proteins on lysine residues in a process called ISGylation (Loeb & Haas, 1992; Zhang & Zhang, 2011). After tryptic digest, isgylated peptides harbor GlyGly modifications on lysines that can be readily detected by LC-MS/MS. As we did not directly detect GlyGly-modified peptides, we combined MIP-APMS with GlyGly enrichment and indeed identified two GlyGly modification sites on TRAF2 (Positions K27, K320; Fig 3I). To deduce the impact of ISGylation on the TRAF2 interaction network, we performed site-directed mutagenesis of TRAF2 lysines and subjected the K→R mutant cell lines to MIP-APMS. Out of the total 32 K→R mutants, 5 showed strong (more than 4×) and significant depletion of ISG15 in the TRAF2 complex (Fig 3J, Table EV4). Interestingly, the most regulated site—K320— was also identified by our initial GlyGly enrichment, suggesting an ISGylation of TRAF2. Reduced ISG15 levels in the interactomes of certain K→R mutants further support this observation. In contrast, TANK levels—a TRAF2 complex member—remained unaltered in the different TRAF2 mutants, pointing toward a specific partial perturbation of the TRAF2 protein community by K→R site-directed mutagenesis (Fig 3K). We excluded potential clonal or TRAF2 mutant expression effects on ISG15 levels by comparing ISG15 levels of transgenic monocyte interactomes to full proteomes (Fig EV5D). Unchanged ISG15 intensities upon stringent MIP-APMS conditions (6 M GdmCl) in a TRAF2 MIP-APMS experiment as well as no evident interaction of recombinant ISG15 and TRAF2 in a size exclusion-based binding assay further support the covalent ISGylation of TRAF2 (Fig EV5E). Functional analysis of the K→R mutants revealed reduction in NFkB activation for K277R, K320R, K364R, and K389R mutants, suggesting that ISGylation of TRAF2 may act as a positive regulator downstream of TLR2 (Fig 3L). To expand our findings to primary human macrophages, we selected the novel phospho-mimetic TRAF2 mutant S11D and lysine-mutant K389R. These experiments confirm ISG15 enrichment in the TRAF2-S11D complex and depletion in the TRAF2-K389R complex (Fig 3M).

### ARHGEF18 and FOSB are functional regulators downstream of TLR2

To further explore the utility of MIP-APMS for discovery of new interactors, we evaluated functional interactions of MAP3K7. MAP3K7 (TAK1) is a central kinase of the MAPK signaling pathway, with crucial roles in the activation of TRAF6 downstream of TLRs

and other receptors (B-cell receptor, TNF receptor) (Landström, 2010) and known as a major regulator of NFkB signaling (Sato *et al*, 2005). The MIP-APMS analysis recapitulated the TNFα/NFκB signaling complex 7 (CORUM) consisting of TAB1, TAB2, TAB3, and CDC37 (Fig 4A and B). Upon TLR2 activation, TAB1 and SNX17 were depleted from the MAP3K7 complex (Fig 4C), while phosphorylation of MAP3K7 on Ser389 increased, significantly (Fig 4D). This revealed dynamic regulation of both PTMs and PPIs during pathway execution.

From the nine previously unknown interactors, we selected the guanine nucleotide exchange factor ARHGEF18 and the transcriptional

regulators FOSB and FOXK2 for functional hypothesis testing. Because MAP3K7 is implicated in NFκB activation, we used CRISPR to knock out the respective genes in monocytic NFκB reporter cells and determined the pathway activity by luciferase induction that directly correlates with the activation of NFkB (Fig 4E). Upon deletion of genes encoding TLR2 and MYD88 (the receptor and proximal adaptor of PAM3CSK4, respectively (Li *et al*, 2010)), we observed an almost complete inhibition of NFκB activation. CRISPR knockout of *MAP3K7*, and the interactors *ARHGEF18* and *FOSB*, led to a partial reduction of NFκB activation, thereby linking this PPI to a functional downstream phenotype in the signaling cascade (all

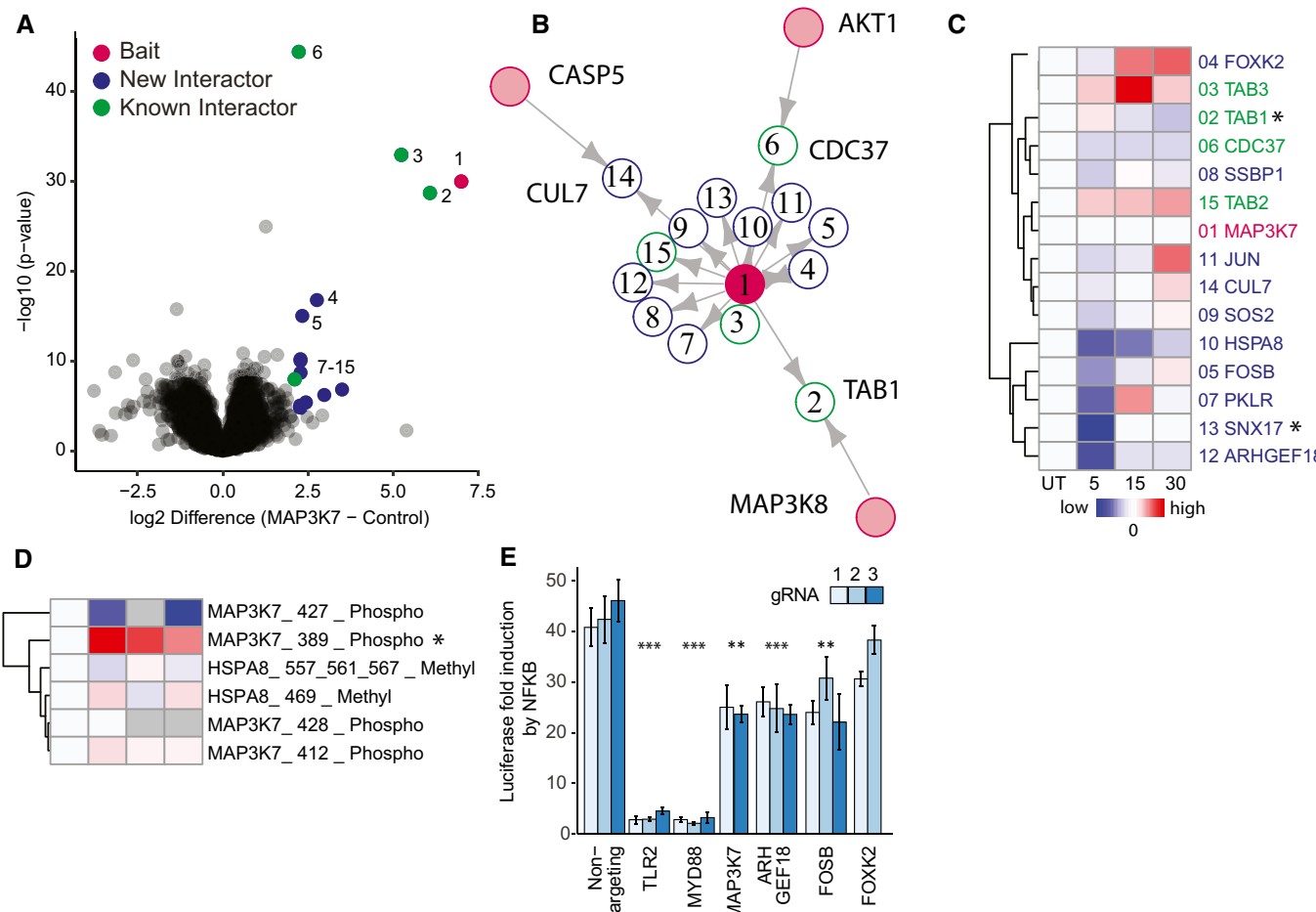

**Figure 4. ARHGEF18 and FOSB are functional regulators downstream of TLR2.**

A Volcano plot representing the interactome of MAP3K7 (measured 16× in biological replicates) compared against all other pull-downs in the control group. The results of the *t*-tests are represented in volcano plots, which show the protein enrichment versus the significance of the enrichment. Numbers indicate enrichment ranks with the heatmap labels of (C) serving as the legend. Significant interactors of MAP3K7 (two-tailed *t*-test, FDR < 0.01, enrichment > 4) are colored in blue (novel interactors) and green (known interactors).

B Interactors of MAP3K7 (blue: novel interactors, green: known interactors) with interconnecting proteins between different baits colored in gray.

C Heatmap of significant interactors of MAP3K7 upon activation, with significant hits in at least one time point (*t*-test, *P*-value < 0.05) denoted with an asterisk. Cell activation was performed for 5, 15, and 30 min with the TLR2 ligand PAM3CSK4 (P3C4).

D Heatmap of MAP3K7 PTMs (phosphorylation) upon activation, with significant hits (*t*-test, *P*-value < 0.05) in at least one time point denoted with an asterisk.

E Induction of NFκB determined based on luciferase luminescence in U937 NFκB reporter cells with CRISPR-Cas9 knockouts of the potential novel interactors of MAP3K7 upon TLR2 activation (each bar represents a mean of four independent measurements; error bars represent the standard deviation; *P*-values were calculated by *t*-test. Asterisks indicate significant differences. ***P*-value < 0.001, ***P*-value < 0.01).

Data information: Gray boxes indicate missing values. See also Appendix Figs S1–S14, Tables EV1 and EV2.

PTMs and PPIs of the characterized bait proteins are available in Appendix Figs S1–S14). We verified CRISPR-KO of ARHGEF18 and FOSB by Western blot analysis (Fig EV5F).

Hence, our MIP-APMS strategy can interrogate the functional relevance of individual molecular switches in a streamlined manner on the levels of PTMs as well as PPIs in signal transduction networks.

### Dissecting drug mode of action for MAPK14 inhibitors with MIP-APMS

Small molecules are often used to interfere with specific cellular functions and are the mainstay of the drug industry. Definition of the target engagement of small molecules is a major challenge in drug discovery and novel proteomics approaches have been devised for this purpose (Schirle *et al*, 2012). We reasoned that MIP-APMS could enable the identification of signaling network rearrangements induced by small molecules, providing a unique proteomic perspective on the mode of drug action. We selected previously described pharmacological inhibitors of p38 kinase (MAPK14) (JX-401 (Friedmann *et al*, 2006), sorafenib (Edwards & Emens, 2010), and skepinone-L (Koeberle *et al*, 2011)) and analyzed their mode of perturbation of the cellular signaling network assemblies involving MAPK14 (Fig 5A–C).

The obtained data indicated that skepinone-L and sorafenib interfered with the physiological intracellular signaling network of MAPK14 to a greater extent than JX-401 (Fig 5D; Table EV5). Further, interestingly, sorafenib and skepinone-L perturbed the interactions within the core complexes differently. While MAPKAPK5, a downstream substrate of MAPK14 (New *et al*, 1998), was depleted in the MAPK14 protein complex upon treatment with both sorafenib and skepinone-L (Fig 5D), only sorafenib reduced the binding of RPS6KA4 (MSK1) and PTPN7 to MAPK14, and even more so upon cellular activation with TLR2 ligands (Fig 5E). Further, both sorafenib and skepinone-L induced hyper-phosphorylation of the MAPK14 phospho-loop on Tyr182, whereas an N-terminal phosphorylation site (Ser2) remained unaltered (Fig 5B and F). This indicated that PTMs and PPIs of MAPK14 are altered upon inhibitor treatment.

Further, MIP-APMS also allowed testing of drug off-target effects (Fig EV5G–I). MAP3K7 phosphorylation on Ser367, Ser412, and Ser445 was significantly altered, and both JUN and TAB2 were depleted from the MAPK14 complex upon treatment with sorafenib. This suggests that the MAP3K7 protein complex, reported to be an upstream activator of MAPK14 (Martín-Blanco, 2000), is in part targeted by MAPK14-specific inhibitors. Enrichment of ELP2 (JX-401, Skepinone-L) and TBK1 (JX-401) was observed in the TRAF2 signaling complex. PTMs on TRAF2 were not affected by the inhibitor treatment. Hence, MIP-APMS can be used to dynamically resolve the interactome and PTM changes upon small molecule treatment and provides information on molecular relationships in signal transduction networks that facilitate understanding of drug mode of action.

## Discussion

Cellular processes are orchestrated by signal transduction pathways that depend on PTMs and PPIs. However, how PTMs and PPIs collaborate in structuring the dynamic signaling network topologies remains incompletely understood, in part because of the laborious experimental approaches involved in dissecting these interactions. Here, we describe MIP-APMS, a combined streamlined cell line generation and proteomics approach to interrogate functional signal transduction networks in intracellular signaling pathways. We quantified more than 370 PPIs and 80 PTMs across innate protein signaling cascades in human monocytes upon receptor activation or drug treatment. Among these are 50 previously undescribed PTMs, including those for which specific enrichment methods are less streamlined, such as ISGylation. Our approach revealed biochemical connections between PTMs and PPIs, as well as protein subnetworks that regulate cellular programs dependent on site-specific PTMs.

We employed MIP-APMS for streamlined and selected interference with protein subnetworks. Demonstrating this principle for the site-specific manipulation of protein phosphorylation as well as ISGylation on TRAF2 yielded differential interactomes of mutated proteins as well as altered cellular physiology. In this way, structural insights into interaction interfaces between protein complexes and crucial PTMs for stabilizing interacting proteins can be revealed. To our knowledge, this is the first description of protein ISGylation augmenting NFkB activity. We disturbed protein interaction networks of the kinase MAPK14 with small molecules to shed additional light on the drug mode of action of kinase inhibitors. Both skepinone-L and sorafenib changed the protein interaction network by a different mode of action, whereas both inhibitors lead to phosphorylation of the MAPK14 phospho-loop.

Modifications, interactions and phenotypes-APMS experiments with temporal resolution further allow the elucidation of co-regulations at different biochemical layers—adding to our understanding of molecular connections along the sequential steps of signal transduction. By further increasing temporal resolution, it may become possible to resolve the causalities between regulation on the PTM and PPIS levels in even greater detail.

For epitope tagging, we employed constructs from the pLOC library (GE Healthcare); however, other cDNA libraries or gene synthesis can readily be employed with polymerase chain reaction (PCR) to obtain DNA fragments with respective homologous overhangs. We employed the cost-effective, non-commercial SLICE cloning strategy; however, commercial solutions using NEBuilder or Gateway are possible with our vector system. An advantage of the small peptide tag chosen for the enrichment strategy in the current study is that it results in little steric interference with physiological protein–protein interactions.

According to our evaluation, Strep-tag and His-tag-based enrichments resulted in high bait sequence coverage; however, His-tag captured known interaction partners most comprehensive. By optimizing a non-stringent lysis procedure with low detergent and salt concentrations and also low temperature in the MIP-APMS protocol, we aimed to capture PTMs together with stable as well as transient interactions. According to our analysis, on average 12.3% of the PTMs and 5.5% of the PPIs are dynamic; however, as biochemical procedures impact recovery of interactors and different thresholds for significance calling are employed, comparability of PTM and PPI dynamics across studies remains challenging. Incorporating chemical cross-linking approaches (Holding, 2015; Liu & Heck, 2015)

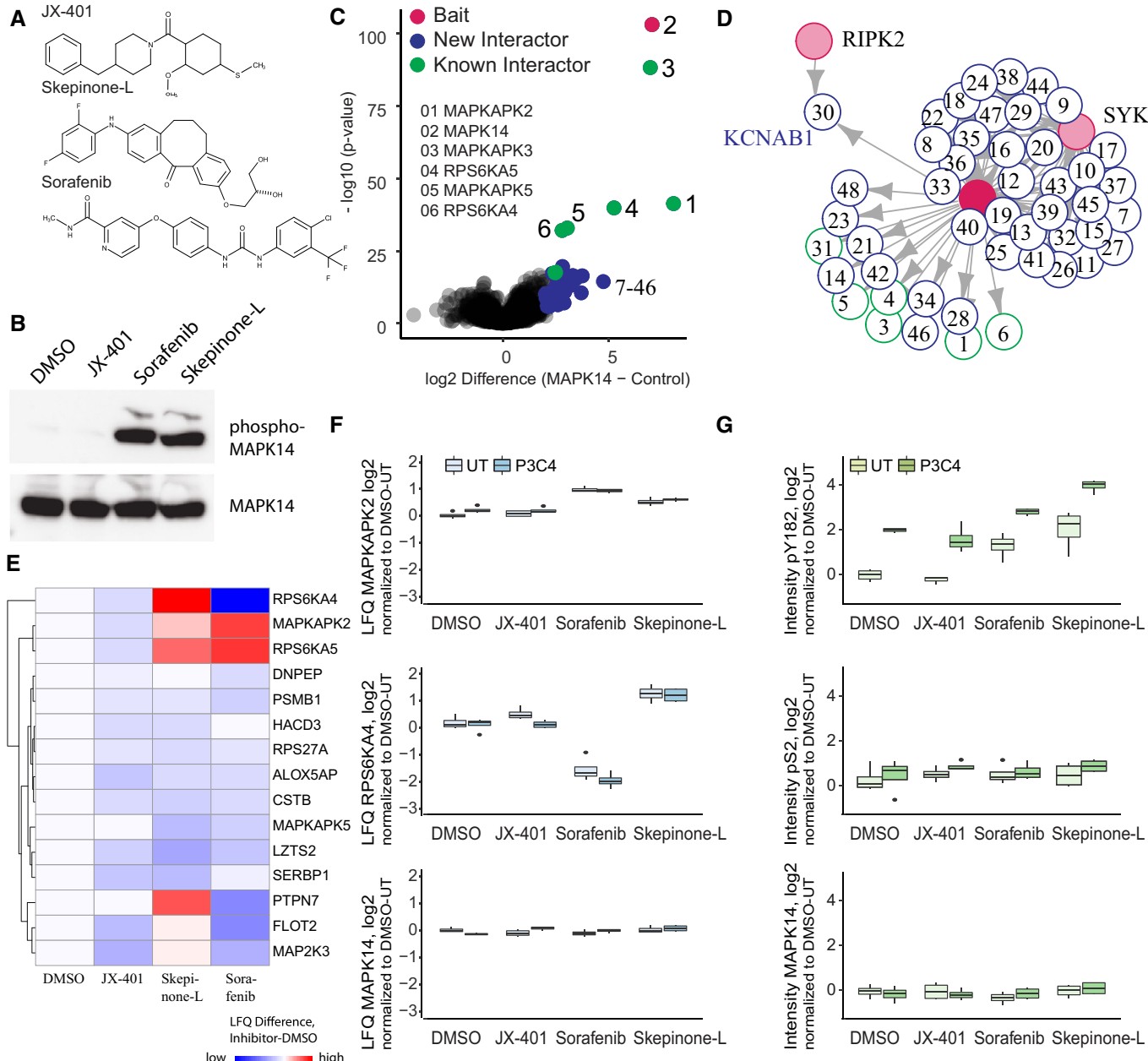

**Figure 5. Dissecting drug mode of action for MAPK14 inhibitors with MIP-APMS.**

A   Chemical structures of MAPK14 inhibitors JX-401, skepinone-L, and sorafenib.

B   Phosphorylation of MAPK14 in U937 WT after treatment with the inhibitors, analyzed by Western blotting using an alpha-phospho-MAPK14 antibody. Total MAPK14, detected by alpha-MAPK14 antibody, was used as a loading control.

C   The interactome of MAPK14 compared against all other pull-downs in the control group. The results of the *t*-tests are represented in volcano plots, which show the protein enrichment versus the significance of the enrichment. Numbers indicate enrichment ranks with the heatmap labels of (C) serving as the legend. Only the top interactors of MAPK14 are numbered. The complete list can be found in Table EV1.

D   Interactors of MAPK14 (blue: novel interactors, green: known interactors) with interconnecting proteins between different baits colored in gray.

E   Heatmap of MAPK14 interactors significantly altered upon treatment with the different MAPK14 inhibitors, with significant hits in at least one treatment (*t*-test, *P*-value < 0.05) denoted with an asterisk. Treatments were normalized to DMSO control. The complete list can be found in Table EV5.

F   LFQ intensity profiles of the MAPK14 interactors RPS6KA4 and MAPKAPK2 and MAPK14 after treatment with different MAPK14 inhibitors, normalized to MAPK14 bait intensity. Drug mode of action was analyzed in the presence (P3C4, 0.5 μg/ml, 30 min) or absence of P3C4 after inhibitor treatment. Central band of the boxplot shows the median, boxes represent the IQR, and 4 biological replicates were performed for every condition.

G   Intensity profiles of MAPK14 phosphorylation on positions Ser2 and Tyr182 and MAPK14 protein intensity after treatment with different MAPK14 inhibitors, normalized to MAPK14 bait intensity. Drug mode of action was analyzed in the presence (P3C4, 0.5 μg/ml, 30 min) or absence of P3C4 after inhibitor treatment. Central band of the boxplot shows the median, boxes represent the IQR, and 4 biological replicates were performed for every condition.

Data information: Gray boxes indicate missing values. Bars represent median, error bars s.d. See also Fig EV5, Table EV5.

could further stabilize transient interactors. Combining MIP-APMS with structural information of the bait protein as well as its interactome can potentially reveal distinct interaction interfaces of protein complexes that are perturbed by site-specific covalent modification or drug action. It would be particularly interesting to integrate protein cross-linking with the PTM status, e.g., of wild-type vs. mutant variant proteins of interest, so that altered structural interaction interfaces can be resolved in addition to differential PTM-dependent PPIs.

We have already explored the strategy of disturbing protein interaction networks using small molecules to determine the effect of drugs on protein complexes. Whereas other proteomics approaches are aimed at identifying drug targets (Molina *et al*, 2013), MIP-APMS elucidates changes in protein communities involving the selected targets. It may thus serve as an additional drug discovery tool to resolve target protein network properties or off-target effects.

Despite its advantages, MIP-APMS currently has some limitations, which can be addressed by developing the method further in the future. These include the possibility that bait protein levels are different from those of endogenous proteins and incomplete protein sequence coverage. MIP-APMS is based on epitope-tagged bait proteins, which are introduced into target cells by lentiviral cellular transduction. Although this strategy enables rapid experiments and functional interrogation with high bait throughput, protein production levels may deviate from endogenous levels with ectopic expression instead of genome editing (Lackner *et al*, 2015). To address this and avoid excessive overproduction of the bait proteins (see Fig EV1B), we employed an engineered weak PGK promoter, as opposed to the commonly used strong cytomegalovirus (CMV) promoter (Qin *et al*, 2010). In general, we recommend total proteome measurements as described in the current study to evaluate whether normalization of changes caused by bait introduction is required.

Using MIP-APMS, we achieved 70% sequence coverage for bait proteins using a single-enzyme protein digestion strategy combined with data-dependent acquisition. To further increase sequence coverage and map PTMs on bait proteins even more comprehensively, additional proteases, e.g., chymotrypsin or GluC, could be used. As MIP-APMS does not include a second enrichment step, the method preferentially quantifies abundant PTMs on bait and prey proteins. Ubiquitinylation, neddylation, and ISGylation are known as sub-stoichiometric PTMs and special biochemical enrichment or MS methods are commonly used for their detection (Kim *et al*, 2011; Wagner *et al*, 2011; Bustos *et al*, 2012; Hansen *et al*, 2021). We show that MIP-APMS combined with GlyGly enrichment facilitates the bait-centric identification of ubiquitin-like modification sites, exemplified for TRAF2. In the future, the total measuring time per sample at a comparable proteomics depth may be further reduced by using data-independent acquisition strategies and short LC gradients (Bruderer *et al*, 2017; Bache *et al*, 2018). Reproducibility, precision and accuracy of modified peptide quantification may be increased further by using isobaric labeling strategies as opposed to LFQ (Hogrebe *et al*, 2018; Virreira Winter *et al*, 2018).

In conclusion, MIP-APMS provides a versatile platform for parallel and time-resolved determination of PPIs and all PTMs of protein complexes in all transducible cells. It quantitatively resolves dynamic signaling network topologies and has broad applicability for the monitoring of virtually all coordinated intracellular programs. Owing to its conceptual design, emerging hypotheses on PTM and PPI involvement in selected signaling cascades are readily testable by protein mutation or loss of function impact on cellular phenotypes.

# Materials and Methods

### Reagents and Tools table

| Reagent or Resource | Source | Identifier |
| --- | --- | --- |
| **Experimental Models: Cell Lines** | | |
| Human: HEK293T | ATCC | CRL-3216 |
| Human: U937 | ATCC | CRL-1593.2 |
| **Recombinant DNA** | | |
| CRISPR vector | Transomics | TELA1002 |
| Gene synthesis | This paper | Appendix Table S6 |
| pLOC vectors | GE Healthcare | Appendix Table S5 |
| pMD2.G | Addgene | #12259 |
| psPAX | Addgene | #12260 |
| **Antibodies** | | |
| Anti-rabbit IgG, HRP-linked antibody | Cell Signaling | 7074 |
| GAPDH (14C10) Rabbit mAb | Cell Signaling | 2118 |
| Phospho-p38 MAPK (Thr180/Tyr182) antibody | Cell Signaling | 9211 |

**Reagents and Tools table** (continued)

| Reagent or Resource | Source | Identifier |
|---|---|---|
| ARHGEF18 | Sigma | HPA042689 |
| MAP3K7 | R&D | MAB5307 |
| FOSB | R&D | AF2214 |
| **Bacterial and Virus Strains** | | |
| Cignal Lenti NFκB Reporter (luc) | Qiagen | CLS-013L |
| XL1-Blue Competent Cells | Agilent Technologies | 200249 |
| **Oligonucleotides** | | |
| PCR and cloning primers | This study | Appendix Table S1–S4 |
| **Chemicals, enzymes, and other reagents** | | |
| Blasticidin | Invivogen | ant-bl-1 |
| cOmplete™, Mini Protease Inhibitor Cocktail | Sigma | 4693132001 |
| DMEM | Life Technologies | 31966047 |
| JX-401 | Santa Cruz Biotechnology | CAS 349087-34-9 |
| LysC | Wako-Chemicals | 129-02541 |
| Ni-IDA Agarose | Jena Bioscience | AC-310-25 |
| PAM3CSK4 | Invivogen | tlrl-pms |
| Passive Lysis 5X Buffer | Promega | E1941 |
| Phosstop—20 TABLETS | Sigma | 4906837001 |
| Phusion® High-Fidelity DNA Polymerase | New England Biolabs | M0530S |
| Polybrene | Sigma | 107689 |
| Polyethylenimine, Linear, MW 25000, Transfection Grade (PEI 25K) | Polysciences | 23966-1 |
| Puromycin | Invivogen | ant-pr-5 |
| RPMI-1640 | Life Technologies | 72400054 |
| Skepinone-L | Merck | 506174-5MG |
| Sorafenib | Santa Cruz Biotechnology | CAS 284461-73-0 |
| SwaI | New England Biolabs | R0604L |
| T4 DNA ligase reaction buffer | New England Biolabs | B0202S |
| Trypsin | Sigma | T6567-1mg |
| Critical commercial assays | | |
| Dual-Luciferase® Reporter Assay System | Promega | E1910 |
| QUIKChange II XL Site-Directed Mutagenesis Kit | Agilent | #200521 |
| Oligonucleotides | | |
| Oligos for pLOC cloning, site-directed mutagenesis, and gRNA cloning | This paper | Appendix Table S6 |
| **Software and Algorithms** | | |
| MaxQuant | (Cox & Mann, 2008) | http://www.biochem.mpg.de/5111795/maxquant |
| Perseus | (Tyanova et al, 2016) | http://www.biochem.mpg.de/5111810/perseus |
| R | NA | https://www.r-project.org/ |
| Ggplot2 | (Wickham, 2016) | https://cran.r-project.org/web/packages/ggplot2/ggplot2.pdf |
| Igraph | NA | http://igraph.org/r/ |
| CHOPCHOP | (Labun et al, 2016) | http://chopchop.cbu.uib.no/index.php |

## Methods and Protocols

### Experimental design

All experiments were performed in replicate. No aspect of the study was blinded. Sample size was not predetermined, and no outliers were excluded from analyses.

### Molecular biology

#### Entry vector design

We made use of restriction enzyme-free seamless ligation cloning extract (SLiCE) cloning using universal primer pairs to insert coding sequences of genes from the Precision LentiORF Collection (pLOC) library (GE Healthcare) into target vectors under the control of a modified weak phosphoglycerate kinase (PGK) promoter, introducing C-terminal epitope tags into the encoded proteins (Zhang *et al*, 2012). Our vector system is compatible with commercial DNA assembly cloning strategies such as the NEB Builder platform or Gateway due to Attl sites flanking the GOIs.

Entry vectors for SLiCE cloning were derived from the pLOC library (GE Healthcare). The vector s include a blasticidin resistance cassette for antibiotic-assisted cell-line selection and an IRES-GFP for FACS sorting. An efficient entry site for SLiCE cloning of the GOI was integrated by SLiCE cloning: The original pLOC vector was PCR-amplified using primers 1 and 2 containing overhangs with a SwaI restriction enzyme (New England Biolabs) cutting site for plasmid linearization, attL1/2 sites for homologous recombination-based SLiCE cloning, and a His-GSG-Flag-tag for GOI epitope tagging. SLiCE cloning was performed as previously described (Zhang *et al*, 2012). Briefly, 300 ng of the amplified pLOC vector, 1:10 (v/v) SLiCE extract (in-house), and 1:10 (v/v) T4 ligase buffer (New England Biolabs) was incubated for 1 h at 37°C. After incubation, the SLiCE mixture was used to transform XL1-blue bacteria (in-house) by heat shock. The transformants were selected on LB plates supplemented with 100 μg/ml ampicillin (LB-Amp plates) after overnight incubation at 37°C. Positive clones were identified by sequencing using primer 3.

The CMV promoter in the modified pLOC vector was exchanged for a weak PGK promoter by SLiCE cloning: the modified pLOC vector was PCR-amplified using primers 4 and 5 (see Appendix Table S1), the weak PGK promoter with homologous ends to the modified pLOC vector was *de novo* synthesized (see Appendix Table S6), and the two fragments were combined by SLiCE, as described before (Zhang *et al*, 2012). Briefly, 300 ng of the amplified pLOC vector, 100 ng of the synthesized weak PGK promoter fragment, 1:10 (v/v) SLiCE extract, and 1:10 (v/v) T4 ligase buffer (New England Biolabs) were incubated for 1 h at 37°C. After incubation, the SLiCE mixture was used to transform XL1-blue bacteria by heat shock. The transformants were selected on LB-Amp plates after overnight incubation at 37°C. Positive clones were identified by sequencing using primer 6 (see Appendix Table S1), as above. The obtained vector was used in subsequent cloning steps as an entry vector, called pLOC entry vector (pLOC-PGKweak-C-HisGSGFlag-BLASTICIDIN).

#### Cloning for epitope tagging

Open-reading frame (ORF) clones were obtained from the Precision LentiORF Collection. GOI (see Appendix Table S5) were PCR-amplified from the pLOC library (GE Healthcare) using the universal primers 7, 8, and 9. The CDS of MAP3K7 with attL1/attL2 overhangs was obtained by gene synthesis (Thermo Fisher Scientific). The pLOC entry vector was digested with the restriction enzyme SwaI (New England Biolabs) according to the manufacturer's instructions. Then, 300 ng of linearized pLOC entry vector, 100 ng of amplified GOI, 1:10 (v/v) SLiCE extract, and 1:10 (v/v) T4 ligase buffer (New England Biolabs) were incubated for 1 h at 37°C. After incubation, the SLiCE mixture was used to transform XL1-blue bacteria by heat shock. The transformants were selected on LB-Amp plates supplemented with 10% (v/v) glucose after overnight incubation at 37°C. Positive clones were identified by sequencing using primers 4 and 10 (see Appendix Table S1).

#### Site-directed mutagenesis of selected phosphosites

For the site-directed mutagenesis of the N-terminal TRAF2 phosphosites and TRAF2 K→R mutants, the QUIKChange II XL site-directed mutagenesis kit (Agilent) was employed. The site-directed mutagenesis was performed by PCR amplification of pLOC-TRAF2 using specific primers (see Appendix Table S2), according to the manufacturer's instructions.

#### Molecular biology and protein purification for TRAF2-ISG15 binding assays

ISG15, TRAF2, and influenza B virus NS1B were cloned into pCoofy vector as a N-terminal His-GST fusion. Plasmids were transformed into Rosetta (DE3) pLacI cells, grown in TB medium, and expression induced with 200 μM IPTG at $OD_{600}$ 0.4–0.8. After induction, cultures were grown for 16 h at 18°C. Cells were re-suspended in lysis buffer (50 mM Tris pH 8.0, 150 mM NaCl, 2 mM β-mercaptoethanol, protease inhibitor cocktail [Roche]) and lysed by sonication. Proteins were purified in tandem with His- and glutathione resin. Purified proteins were cleaved overnight with His-3C PreScission protease at 4°C. Following cleavage, the His-GST tag and His-3C protease were removed by a His pull-down. Proteins were either further purified by SEC (Superdex 75 10/300 GL, GE Life Sciences) or immediately buffer exchanged into storage buffer (50 mM Tris, 150 mM NaCl, 2 mM DTT). Proteins were concentrated and flash-frozen in liquid nitrogen.

### Cell biology

#### Cell culture

U937 cells (CRL-1593.2) were purchased from the ATCC. The cells were cultured according to the manufacturer's instructions, in RPMI-1640 medium (Life Technologies) supplemented with 100 U/ml penicillin (GIBCO), 100 μg/ml streptomycin (GIBCO), and 10% (v/v) heat-inactivated fetal bovine serum (GIBCO; complete RPMI medium). The cells were incubated at 37°C under 5% $CO_2$.

HEK293T cells (CRL-3216) were purchased from ATCC. The cells were cultured according to the manufacturer's instructions, in DMEM (Life Technologies) supplemented with 100 U/ml penicillin (GIBCO), 100 μg/ml streptomycin (GIBCO), 1× Glutamax (GIBCO), and 10% heat-inactivated fetal bovine serum (complete DMEM medium). The cells were incubated at 37°C under 5% $CO_2$.

Primary human monocytes were obtained by culturing primary human monocytes enriched from buffy coats as described previously (Rieckmann *et al*, 2017). Primary human macrophages were differentiated in RPMI-1640 medium (Life Technologies) supplemented with 100 U/ml penicillin (GIBCO), 100 μg/ml streptomycin (GIBCO),

10% (v/v) heat-inactivated fetal bovine serum (GIBCO), and 50 ng/ml M-Csf. The cells were incubated at 37°C under 5% $CO_2$.

### NFκB reporter cell lines

U937 cell lines were transduced with Cignal Lenti NFκB-reporter constructs (Qiagen) according to the manufacturer's instructions. The transductants were selected in the presence of puromycin (5 µg/ml) for 14 days to establish stable cell lines.

### Cell lines for epitope-tagged bait proteins

For lentivirus production, HEK293T cells ($2 \times 10^6$, one six-well) were transfected with sequence-validated pLOC-GOI vectors using polyethyleneimine (Polysciences) as a transfection reagent. Helper plasmids pMD2.G, psPAX, and the pLOC vector harboring the GOI were combined in a ratio of 1:1.5:2. After 4-h incubation in complete RPMI medium at 37°C under 5% $CO_2$, the transfection mix was removed and fresh complete RPMI medium was added. Lentiviral supernatant was collected after 48-h incubation at 37°C under 5% $CO_2$, centrifuged (500 *g*, 5 min), filtered (0.45 µm), and supplemented with 8 µg/ml polybrene (Sigma). Then, the virus (complete supernatant of one six-well) was added to 0.2 Mio U937 cells or U937-NFkB Reporter Cell lines, incubated for 4 h at 37°C under 5% $CO_2$, following which fresh medium was added. Selection pressure with blasticidin (10 µg/ml; Invivogen) was introduced after 48 h. The cells were cultured for 2 weeks under the selective pressure and then directly used in MIP-APMS experiments.

Transduction of primary human macrophages was performed as previously described (Berger *et al*, 2011). In short, 10 Mio macrophages were transduced with a mix of VPX-Vlps and pLOC lentivirus (v/v, 50%) in the presence of polybrene (8 µg/ml), incubated for 4 h at 37°C under 5% $CO_2$, following which fresh medium was added. Cells were harvested after 72h and then directly used in MIP-APMS experiments.

HEK293T (10 Mio) cells were transfected with pLOC-MAPK14-HisGSGFlag or pLOC-MAPK14-Strep using polyethyleneimine (Polysciences) as a transfection reagent. After 4-h incubation in complete DMEM medium at 37°C under 5% $CO_2$, the transfection mix was removed and fresh complete DMEM medium was added. Cells were harvested after 72h and then directly used in MIP-APMS, Flag-MS, and Strep-MS experiments.

### TLR2 activation of U937 cells

Cells (5 Mio suspension) were seeded in deep-well 24-well plates, with one plate was used per cell line. TLR2 activation with PAM3CSK4 (0.5 µg/ml; Invivogen) was performed in a reverse time course and in quadruplicate, for 30, 15, 5, and 0 min at 37°C under 5% $CO_2$. The 0 min time point was not treated with PAM3CSK4. The cells were harvested by centrifugation and flash-frozen and stored at −80°C until MIP-APMS.

### Drug mode of action on MAPK14

Cells (5 Mio) were seeded in deep-well 24-well plates. The cells were treated with MAPK14 inhibitors (sorafenib: 10 µM; skepinone-L: 80 nM; and JX-401: 10 µM) for 2 h at 37°C under 5% $CO_2$ in quadruplicate. Inhibitor-treated cells and controls either harvested directly or were activated with PAM3CSK3 (P3C4, 0.5 µg/ml; Invivogen) for 30 min at 37°C under 5% $CO_2$. Cells were harvested by centrifugation and frozen until MIP-APMS.

### CRISPR/Cas9 Knockout

CRISPR knockout experiments were performed to identify potential novel interactors of MAP3K7 (see Appendix Table S3). For effective delivery of gRNA and Cas9, the transEDIT gRNA Plus Cas9 Expression vector with blasticidin was purchased from Transomics. For the experiment, gRNAs were designed using the web tool CHOP-CHOP (Labun *et al*, 2016) and cloned into the transEDIT vector according to the manufacturer's instructions (the primer list is provided in Appendix Table S3). Virus for each gRNA was produced as explained above (Cell Lines for Epitope-Tagged Bait Proteins). The U937-NFκB reporter cells were transduced and co-selected using puromycin (5 µg/ml) and blasticidin (10 µg/ml) at 37°C under 5% $CO_2$.

### Luciferase reporter assay

U937-NFκB reporter cells ($5 \times 10^4$) were seeded in quadruplicate on the day before the experiment. The cells were activated with PAM3CSK4 (0.5 µg/ml; Invivogen) for 6 h and harvested in passive lysis buffer (Promega). Luminescence of *Renilla* luciferase was determined in a dual-luciferase reporter assay (Promega), according to the manufacturer's instructions, using a microplate reader (Tecan).

## Biochemistry

### Western blots

One million U937 cells were stimulated, washed in PBS, and lysed in buffer (4% SDS, 40 mM HEPES [pH 7.4, 10 mM DTT] supplemented with protease inhibitors [Sigma-Aldrich, 4693159001]). Samples were centrifuged (16,000 *g*, 10 min), Li-LDS sample buffer was added to a final concentration of 1×, and the supernatant was incubated (5 min, 95°C). Proteins were separated on 12% Novex Tris-glycine gels (Thermo Fisher Scientific, XP00120BOX) and transferred onto PVDF membranes (Merck Millipore, IPVH00010) or Nitrocellulose membranes (Amersham, 10600002). Membranes were blocked in 5% BSA in PBST, and antibodies were diluted in 2% BSA in PBST. Antibodies used for immunoblotting were as follows (diluted 1:1,000): phospho-p38 MAPK (Thr180/Tyr182) antibody (Cell Signaling, 9211), GAPDH (14C10) rabbit mAb (Cell Signalling, 2118), p38 MAPK (R&D, AF8691), ARHGEF18 (Sigma, HPA042689), MAP3K7 (R&D, MAB5307), FOSB (R&D, AF2214) and anti-rabbit IgG, HRP-linked antibody (Cell Signaling, 7074).

### His-IMAC enrichment

Frozen pellets of 19 cell lines with bait proteins containing 9x His-tags in deep-well 24-well plates were defrosted (5 min, 37°C). The cells were re-suspended in 800 µl of lysis buffer (10 mM HEPES [pH 7.5; Gibco], 50 mM NaCl [Sigma], 20 mM imidazole [Sigma], 0.05% NP-40 [Thermo Fisher], 1 mM $MgCl_2$ [Sigma], 50 U/ml benzonase [in-house], protease inhibitors [Roche, 1 tablet per 50 ml], and phosphatase inhibitors [Roche, 1 tablet per 50 ml]), incubated for 15 min on ice, and cleared by centrifugation (500 *g*, 5 min, 4°C). Supernatants were transferred to deep-well 96-well plates already containing equilibrated Ni-IDA beads (JenaBioScience GmbH, 50 µl slurry per well). The plates were incubated at 4°C for 1 h, shaking. The beads were washed three times (10 mM HEPES [pH 7.5], 50 mM NaCl, and 20 mM imidazole), and the supernatant was removed completely before proceeding.

### Flag-enrichment

Frozen pellets of HEK293T-MAPK14-HisGSGFlag (1xFlag) and control cell lines were defrosted (5 min, 37°C). The cells were re-suspended in 800 µl of lysis buffer (10 mM HEPES [pH 7.5; Gibco], 50 mM NaCl [Sigma], 0.05% NP-40 [Thermo Fisher], 1 mM $MgCl_2$ [Sigma], 50 U/ml benzonase [in-house], protease inhibitors [Roche, 1 tablet per 50 ml], and phosphatase inhibitors [Roche, 1 tablet per 50 ml]), incubated for 15 min on ice, and cleared by centrifugation (500 g, 5 min, 4°C). Supernatants were transferred to deep-well 96-well plates already containing equilibrated anti-Flag M2 agarose gel (Sigma, 50 µl slurry per well). The plates were incubated at 4°C for 1 h, with shaking at over 1,500 rpm. The beads were washed three times (10 mM HEPES [pH 7.5], 50 mM NaCl) and the supernatant was removed completely before proceeding.

### Strep-enrichment

Frozen pellets of HEK293T-MAPK14-Strep (1× Strep-tag II) and control cell lines were defrosted (5 min, 37°C). The cells were re-suspended in 800 µl of lysis buffer (10 mM HEPES [pH 7.5; Gibco], 50 mM NaCl [Sigma], 0.05% NP-40 [Thermo Fisher], 1 mM $MgCl_2$ [Sigma], 50 U/ml benzonase [in-house], protease inhibitors [Roche, 1 tablet per 50 ml], and phosphatase inhibitors [Roche, 1 tablet per 50 ml]), incubated for 15 min on ice, and cleared by centrifugation (500 g, 5 min, 4°C). Supernatants were transferred to deep-well 96-well plates already containing equilibrated MagStrep "type3" beads (iba, 50 µl slurry per well). The plates were incubated at 4°C for 1 h, with shaking at over 1,500 rpm. The beads were washed three times (10 mM HEPES [pH 7.5], 50 mM NaCl), and the supernatant was removed completely before proceeding with sample preparation for on-bead digestion. For elution, beads were incubated with 50 µl 1× buffer BXT (IBA Lifesciences) and purified proteins were eluted at room temperature for 30 min with constant shaking at 1,100 rpm on a ThermoMixer C incubator as described previously (Gordon et al, 2020). Proportional amounts of bead and elution were analyzed.

### Combination of MIP-APMS with GlyGly enrichment

We used 500 Mio TRAF2-U937 cells and performed His-IMAC enrichment as described above adjusted for higher input. The sample was digested as explained below under sample preparation. Peptide desalting was performed on SepPack C18 columns as per the manufacturer's instruction. After elution, peptides were lyophilized overnight. The lyophilized sample was reconstituted in 900 µl cold immunoaffinity purification buffer (IAP; 50 mM MOPS, pH 7.2, 10 mM $Na_2HPO_4$, 50 mM NaCl). For the enrichment of diGly remnant containing peptides, antibodies of the PTMScan® Ubiquitin Remnant Motif (K-ε-GG) Kit (Cell Signaling Technology [CST]) were first cross-linked to beads). For this, one vial of antibody-coupled beads was washed three times with 1 ml cold cross-linking buffer (100 mM sodium tetraborate decahydrate, pH 9.0), followed by 30-min incubation in 1 ml cross-linking buffer (20 mM dimethylpimip-imidate in cross-linking wash buffer) for 30 min at room temperature and gentle agitation. After two consecutive washes with 1 ml cold quenching buffer (200 mM ethanolamine, pH 8.0) and 2-h incubation in 1 ml cold quenching, crosslinked beads were washed three times with 1 ml cold IAP buffer and 1/24 was immediately used for immunoaffinity purification. For this, peptides were added to crosslinked antibody beads and incubated for 2 h at 4°C under

gentle agitation. After incubation, beads were sequentially washed two times with cold IAP buffer and five times with cold $ddH_2O$ in GF-StageTips. Thereafter, peptides were eluted twice with 50 µl 0.15% TFA into SDB-RPS StageTips. Eluted peptides were loaded onto stationary material and washed once with 200 µl 0.2% TFA and once with 200 µl 0.15% TFA/ 2% ACN. Peptides were eluted from SDB-RPS StageTips with 60 µl 1.25% ammonium hydroxide ($NH_4OH$)/80% ACN and dried using a SpeedVac centrifuge (Eppendorf, Concentrator plus). For mass spectrometry, dried peptides were re-suspended in 9 µl A* (2% ACN, 0.1% TFA).

### Analytical size-exclusion chromatography binding assays

Binding assays were performed with ISG15 (1–157 aa) and TRAF2 variants (1–185 aa) on a Vanquish HPLC system (Thermo Fisher Scientific) using an AdvanceBio size-exclusion chromatography column (Agilent Technologies). As a positive control for ISG15 binding, the influenza B virus NS1B protein (1–103 aa) was used. Prior to analytical sizing, the column was pre-equilibrated with SEC buffer (50 mM Tris pH 7.5, 150 mM NaCl, 2 mM DTT). ISG15 (30 µM) was mixed with TRAF2 variants or NS1B (25 µM) prior to injection on the column. Fractions were mixed with SDS sample buffer and resolved on a 4–20% gradient SDS/PAGE. Gels were visualized by Coomassie staining.

### Quantitative proteomics analysis

#### MIP-APMS sample preparation

After His-IMAC, the beads were re-suspended in 50 µl of 8 M urea and 40 mM HEPES (pH 8.0). LysC digestion (Wako, 0.5 µg/µl, 1 µl) was performed for 3 h at room temperature - 25°C (with shaking, 1,500 rpm). Afterward, the samples were diluted (1:6) with water and digested with trypsin (Sigma; 0.5 µg/µl, 1 µl) for 16 h (room temperature, with shaking, 1,500 rpm). The digests were centrifuged (5 min, 500 g), and the supernatants were transferred to new 96-well plates. Cysteines were reduced by the addition of dithiothreitol (1 mM, room temperature, 1,500 rpm, 30 min), before proceeding to cysteine alkylation with iodoacetamide (55 mM, room temperature, 30 min, dark). Excess iodoacetamide was quenched by adding thiourea (100 mM, room temperature, 10 min) prior to acidification for peptide desalting with trifluoroacetic acid (TFA; final concentration: 1% v/v). Peptides were loaded onto C18 StageTips (EmporeTM, IVA-Analysentechnik). They were then eluted with 80% acetonitrile, dried using a SpeedVac, and re-suspended in a solution of 2% acetonitrile, 0.1% TFA, and 0.5% acetic acid.

#### Whole-proteome MS sample preparation

Cells were lysed in SDC-lysis buffer and digested with LysC and trypsin, as described previously (Kulak et al, 2017). Peptides were desalted on stacked poly(styrene-divinylbenzene) reversed-phase sulfonate plugs and eluted with a mixture of 80% acetonitrile, 19% $ddH_2O$, and 1% ammonia. MS measurements were performed in replicate ($n = 3$) using Q Exactive HF (Thermo Fisher Scientific).

#### LC-MS/MS

Peptides were separated using an EASY-nLC 1200 HPLC system (Thermo Fisher Scientific) coupled online to the Q Exactive HF and Q Exactive HF-X mass spectrometer via a nanoelectrospray source (Thermo Fisher Scientific), as described before (Scheltema et al,

2014; Kelstrup *et al,* 2018). Peptides were loaded in buffer A (0.5% formic acid) on in-house packed columns (75 μm inner diameter and 20 cm long; packed with 1.9-μm C18 particles from Dr. Maisch GmbH, Germany). Peptides were eluted using a nonlinear 95-min gradient of 5–60% buffer B (80% acetonitrile and 0.5% formic acid) at a flow rate of 300 nl/min and a column temperature of 55°C. The operational parameters were monitored in real-time by using the SprayQC software (in-house) (Scheltema & Mann, 2012). The Q Exactive HF and Q Exactive HF-X were operated in a data-dependent acquisition positive mode with a survey scan range of 300–1,650 $m/z$ and a resolution of 60,000–120,000 at $m/z$ 200. Up to 15 most abundant isotope patterns with a charge of > 1 were isolated using a 1.8 Thomson (Th) isolation window and subjected to high-energy collisional dissociation fragmentation at a normalized collision energy of 27. Fragmentation spectra were acquired with a resolution of 15,000 at $m/z$ 200. Dynamic exclusion of sequenced peptides was set to 20 s to reduce repeated peptide sequencing. Thresholds for ion injection time and ion target values were set to 20 ms and 3E6 for the survey scans, and 55 ms and 1E5 for the MS/MS scans. Data were acquired using the Xcalibur software (Thermo Scientific).

### Quantification and statistical analysis

#### Peptide identification and LC-MS/MS data analysis

MaxQuant software (version 1.5.3.16) was used to analyze MS raw files. MS/MS spectra were searched against the human Uniprot FASTA database (version July 2015, 91,645 entries) and a common contaminants database (247 entries) by the Andromeda search engine (Cox & Mann, 2008). Cysteine carbamidomethylation was set as a fixed modification, and N-terminal acetylation and methionine oxidation were set as variable modifications. To identify and quantify phosphorylation, acetylation, and methylation, variable modification search was consecutively performed. Enzyme specificity was set to trypsin, with a maximum of two missed cleavages and a minimum peptide length of seven amino acids. FDR of 1% was applied at the peptide and protein level. Peptide identification was performed with an allowed initial precursor mass deviation of up to 7 ppm and an allowed fragment mass deviation of 20 ppm. Nonlinear retention time alignment of all analyzed samples was performed using MaxQuant. Peptide identifications were matched across all samples within a time window of 1 min of the aligned retention times. Protein identification required at least one "razor peptide" in MaxQuant. A minimum ratio count of 1 was required for valid quantification events using the MaxQuant's LFQ algorithm (MaxLFQ). Data were filtered for the presence of common contaminants and peptides only identified by site modification, and hits to the reverse database (Cox & Mann, 2008) were excluded from further analysis.

Dependent peptide in MaxQuant analysis was performed to analyze unbiased PTMs on MAPK14 with standard parameters (FDR < 0.01, Mass bin size 0.0065 Da). For TagGraph analysis, sequence interpretations were first analyzed with the de novo search engine Peaks. Peaks analysis was performed with 10 ppm precursor mass tolerance and 0.01 Da fragment mass tolerance (Ma *et al,* 2003). TagGraph analysis was performed using human Uniprot FASTA database (version July 2015, 91,645 entries), with FDR cutoff of 0.1, and all other settings remained to unchanged as present in the software distribution.

#### Interactor calling

We integrate differences in intensity and abundance as described before (Keilhauer *et al,* 2015). We employ an AE-MS workflow with quantitative MS, which means that we use not only the information for protein identification but also for protein quantification for post-experiment interactor calling. To determine which proteins are substantially enriched (i.e., bait and prey proteins), AE-MS employs standard statistical testing (*t*-test) with a multiple hypothesis correction (FDR 0.01 for multiple hypothesis testing). In detail, each quantified protein had to be identified with more than one peptide and in more than 60% of replicates of at least one cell line to be considered valid. Protein LFQ intensities were log-transformed to the base of 2 and missing values imputed from a random normal distribution centered on the detection limit (width = 0.3, Down Shift:1.8). Samples were clustered by using Pearson correlation into different control groups in the Perseus environment leading to three separate groups (see Fig EV3K). To identify the interactors, a two-tailed Student's *t*-test (permutation-based FDR < 0.01 with 250 randomizations, enrichment > 2) with a minimum of 10 valid values in the first group was performed in the Perseus environment, using all other cell lines in the respective control group (Tyanova *et al,* 2016). Here, the baits were loaded as first group and second group mode was selected as "complement". Significant interactors were compared to the STRING and Biogrid databases (Szklarczyk *et al,* 2015; Chatr-Aryamontri *et al,* 2017) and overlaps were denoted in the Figs.

#### SAINT analysis via crapome

MAPK14 His IPs and controls (U937 transduced with His-Tag) were performed in triplicates and uploaded to the SAINT-based Crapome server (https://reprint-apms.org) (Mellacheruvu *et al,* 2013). As Experiment Type, we selected single-step epitope tag APMS and spectral counts as quantitation Type. As external controls, we selected PBMC (cell/tissue type), agarose (affinity support), and Q Exactive (Instrument type). The primary empirical fold change score (FC-A) was calculated by user controls using average for combining replicates (number of virtual controls = 10). The secondary fold change score was calculated by all controls (user + external controls) using geometric mean for combining replicates (number of virtual controls = 3). The probabilistic SAINT Score was calculated by user controls (combining replicates: average) and 10 virtual controls. Saint options were 2,000 n-burn, 4,000 n-iter, 0 LowMode, 1 MinFold, and 1 Normalize.

#### Analysis of dynamic PTMs and PPIs

Prior to the analysis of dynamic PPIs, LFQ intensities of significant interactors of each replicate were normalized to the LFQ intensities of the respective bait proteins to avoid loading artifacts.

$$\text{LFQ} - \text{intensity}\,(\text{prey} - \text{protein})_{\text{normalized}} = \frac{\text{LFQ intensity}\,(\text{prey} - \text{protein})}{\text{LFQ intensity}\,(\text{bait} - \text{protein})}.$$

A *two-tailed Student's t*-test (*P*-value < 0.05) was performed on the previously identified significant interactors comparing unactivated conditions versus activated conditions at different time points. Significant dynamic preys were reported with an asterisk in the heatmaps.

Conversely, intensities of modified peptides of each replicate were normalized to the intensity of the respective protein intensity to decrease the total coefficient of variation. PTMs that had valid values in at least 3 replicates of at least one time point were considered for the analysis. No imputation was performed.

$$\text{Intensity(modified peptide of protein}\,X)_{\text{normalized}} = \frac{\text{Intensity(modified peptide of protein}\,X)}{\text{Intensity(protein}\,X)}.$$

A *two-tailed Student's t*-test (*P*-value < 0.05) was performed on the previously identified significant interactors and modified peptides, respectively, comparing un-activated conditions versus activated conditions at different time points. Significant dynamic preys/PTMs were reported with an asterisk in the heatmaps.

### Unsupervised clustering

Intensities of dynamically regulated PPIs (357) and PTMs (178) upon TLR 2 activation were filtered for at least 70% valid values and normalized per time point (PPIs to bait protein intensity and PTMs to protein intensity of the modified protein as explained above). The median of each time point was calculated and then *Z*-scored. Pearson correlation was calculated between each of the PPIs and PTMs, and results were visualized by hierarchical clustering. The data were clustered and median z-scored intensities (confidence interval: 0.95) were plotted against the time course of TLR2 activation (method = loess, *y ~ x*). *N* shows the number of PPIs/PTMs corresponding to each cluster.

### Analysis of whole-proteome data

Full proteomes were measured in triplicates as described under peptide identification and LC-MS/MS data analysis. Data were filtered for the presence of common contaminants and peptides only identified by site modification, and hits to the reverse database (Cox & Mann, 2008) were excluded from further analysis. As a requirement, each quantified protein had to be identified with more than one peptide and in more than 60% of replicates of at least one cell line to be considered valid. Protein LFQ intensities were log-transformed to the base of 2 and missing values imputed from a random normal distribution centered on the detection limit (width = 0.3, Down Shift:1.8). To identify differentially expressed proteins between wildtype and transduced cell lines, a two-tailed Student's *t*-test (permutation-based FDR < 0.05 with 250 randomizations, enrichment > 2) with a minimum of two valid values in the first group was performed in the Perseus environment, using all other cell lines in the respective control group (Tyanova *et al*, 2016). Copy numbers were calculated with the Perseus Plugin Proteomic Ruler, which normalizes protein intensity to the molecular mass of each protein (Wićniewski *et al*, 2014).

## Data availability

The mass spectrometry proteomics data have been deposited to the ProteomeXchange Consortium via the PRIDE partner repository with the dataset identifier PXD010996. The datasets produced in this study are available in the following database: https://www.ebi.ac.uk/pride/

Project accession: PXD010996 (http://www.ebi.ac.uk/pride/archive/projects/PXD010996).

Further information and requests for resources and reagents should be directed to and will be fulfilled by the Lead Contact, Felix Meissner (felix.meissner@uni-bonn.de).

**Expanded View** for this article is available online.

### Acknowledgements

We thank I. Paron, C. Deiml, J. Mueller, and K. Mayr for MS assistance; G. Sowa, J. Baum, V. Boeck S. Dewitz, and A. Ullrich for technical assistance; M. Oroshi for computer and database support; and M. Tanzer, J. Swietlik, J. Rieckmann, M. Steger, J. Bader, and C. Luber for helpful discussions. This work was funded by the Max Planck Society for the Advancement of Science and the Deutsche Forschungsgemeinschaft (DFG, German Research Foundation) Project-ID 165054336 (SFB 914), Project-ID 360372040 (SFB 1335), and Project-ID 408885537 (TRR274). Open Access funding enabled and organized by Projekt DEAL.

### Author contributions

AF, SE, FMH, KP, and KS performed experiments. AF developed and implemented bioinformatics methods. AF and FM conceived the data analysis and interpreted the data. AS and DH assisted in data analysis. FM and MM conceived the study. FM supervised the experiments. AF and FM wrote the manuscript.

### Conflict of interest

The authors declare that they have no conflict of interest.

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
