## [Review Process File · Molecular Systems Biology]

Identification of Covalent Modifications Regulating Immune Signaling Complex Composition and Phenotype

Annika Frauenstein, Stefan Ebner, Fynn Hansen, Ankit Sinha, Kshiti Phulphagar, Kirby Swatek,
Daniel Hornburg, Matthias Mann, and Felix Meissner

DOI: [10.15252/msb.202010125](https://doi.org/10.15252/msb.202010125)

Corresponding author: Felix Meissner (meissner@biochem.mpg.de)

Review Timeline:

Submission Date:	16th Nov 20
Editorial Decision:	7th Jan 21
Revision Received:	9th Jun 21
Editorial Decision:	1st Jul 21
Revision Received:	8th Jul 21
Accepted:	8th Jul 21

Editor: Jingyi Hou

Transaction Report:

7th Jan 2021

Manuscript Number: MSB-2020-10125

Title: Identification of Covalent Modifications Regulating Immune Signaling Complex Composition and Phenotype

Author: Annika Frauenstein

Stefan Ebner

Ankit Sinha

Phulphagar Kshiti

Kirby Swatek

Daniel Hornburg

Matthias Mann

Felix Meissner

Dear Dr Meissner,

Thank you for submitting your work to Molecular Systems Biology. We have now heard back from the three reviewers who agreed to evaluate your manuscript. As you will see below, the reviewers acknowledge that the presented findings could be relevant as a resource. They raise however a series of concerns, which we would ask you to address in a major revision.

Without reiterating all the points raised in the reviews below, some of the more substantial issues are the following:

- In light of reviewers' comments, the statements regarding the overall methodological advance need to be toned down.
- Additional analyses, controls and validation experiments are required, in order to enhance the conclusiveness and the biological insights provided by the study. Reviewer #3 provides constructive suggestions in this regard.

All other issues need to be addressed as well. As you may already know, our editorial policy allows in principle a single round of major revision, so it is essential to provide responses to the reviewers' comments that are as complete as possible. Please feel free to contact me in case you would like to discuss in further detail any of the issues raised by the reviewers.

On a more editorial level, please do the following:

- Please provide a .docx formatted version of the manuscript text (including legends for main figures, EV figures and tables). Please make sure that the changes are highlighted to be clearly visible.
- Please provide individual production quality figure files as .eps, .tif, .jpg (one file per figure).
- Please provide a .docx formatted letter INCLUDING the reviewers' reports and your detailed point-by-point responses to their comments. As part of the EMBO Press transparent editorial process, the point-by-point response is part of the Review Process File (RPF), which will be published alongside your paper.

-Please note that all corresponding authors are required to supply an ORCID ID for their name upon submission of a revised manuscript.

-We replaced Supplementary Information with Expanded View (EV) Figures and Tables that are collapsible/expandable online (see examples in <http://msb.embopress.org/content/11/6/812>). A maximum of 5 EV Figures can be typeset. EV Figures should be cited as 'Figure EV1, Figure EV2' etc... in the text and their respective legends should be included in the main text after the legends of regular figures.

Additional Tables/Datasets should be labeled and referred to as Table EV1, Dataset EV1, etc. Legends have to be provided in a separate tab in case of .xls files. Alternatively, the legend can be supplied as a separate text file (README) and zipped together with the Table/Dataset file.

For the figures and tables that you do NOT wish to display as Expanded View figures, they should be bundled together with their legends in a single PDF file called *Appendix*, which should start with a short Table of Content. Each legend should be below the corresponding Figure/Table in the Appendix. Appendix figures and tables should be referred to in the main text as: "Appendix Figure S1, Appendix Figure S2, Appendix Table S1" etc. See detailed instructions regarding expanded view here: <https://www.embopress.org/page/journal/17444292/authorguide#expandedview>.

-Before submitting your revision, primary datasets (and computer code, where appropriate) produced in this study need to be deposited in an appropriate public database (see <https://www.embopress.org/page/journal/17444292/authorguide#dataavailability>).

The accession numbers and database should be listed in a formal "Data Availability " section (placed after Materials & Method) that follows the model below (see also <https://www.embopress.org/page/journal/17444292/authorguide#dataavailability>). Please note that the Data Availability Section is restricted to new primary data that are part of this study.

Data availability

- We would encourage you to include the source data for figure panels that show essential quantitative information. Additional information on source data and instruction on how to label the files are available at < <https://www.embopress.org/page/journal/17444292/authorguide#sourcedata> >.

- All Materials and Methods need to be described in the main text. We would encourage you to use 'Structured Methods', our new Materials and Methods format. According to this format, the Material and Methods section should include a Reagents and Tools Table (listing key reagents,

experimental models, software and relevant equipment and including their sources and relevant identifiers) followed by a Methods and Protocols section in which we encourage the authors to describe their methods using a step-by-step protocol format with bullet points, to facilitate the adoption of the methodologies across labs. More information on how to adhere to this format as well as downloadable templates (.doc or .xls) for the Reagents and Tools Table can be found in our author guidelines: <

<https://www.embopress.org/page/journal/17444292/authorguide#researcharticleguide>>. An example of a Method paper with Structured Methods can be found here: .

- Regarding data quantification:

Please ensure to specify the name of the statistical test used to generate error bars and P values, the number (n) of independent experiments (please specify technical or biological replicates) underlying each data point and the test used to calculate p-values in each figure legend. Discussion of statistical methodology can be reported in the materials and methods section, but figure legends should contain a basic description of n, P and the test applied.

Graphs must include a description of the bars and the error bars (s.d., s.e.m.).

- Please provide a "standfirst text" summarizing the study in one or two sentences (approximately 250 characters, including space), three to four "bullet points" highlighting the main findings and a "synopsis image" (550px width and max 400px height, jpeg format) to highlight the paper on our homepage.

Here are a couple of examples:

<https://www.embopress.org/doi/10.15252/msb.20199356>

<https://www.embopress.org/doi/10.15252/msb.20209475>

<https://www.embopress.org/doi/10.15252/msb.209495>

When you resubmit your manuscript, please download our CHECKLIST

(<http://bit.ly/EMBOPressAuthorChecklist>) and include the completed form in your submission.

Please note that the Author Checklist will be published alongside the paper as part of the transparent process

(<https://www.embopress.org/page/journal/17444292/authorguide#transparentprocess>).

If you feel you can satisfactorily deal with these points and those listed by the referees, you may wish to submit a revised version of your manuscript. Please attach a covering letter giving details of the way in which you have handled each of the points raised by the referees. A revised manuscript will be once again subject to review and you probably understand that we can give you no guarantee at this stage that the eventual outcome will be favorable.

Yours sincerely,

Jingyi Hou

Editor

Molecular Systems Biology

If you do choose to resubmit, please click on the link below to submit the revision online *within 90

days*.

Link Not Available

IMPORTANT: When you send your revision, we will require the following items:

1. the manuscript text in LaTeX, RTF or MS Word format
2. a letter with a detailed description of the changes made in response to the referees. Please specify clearly the exact places in the text (pages and paragraphs) where each change has been made in response to each specific comment given
3. three to four 'bullet points' highlighting the main findings of your study
4. a short 'blurb' text summarizing in two sentences the study (max. 250 characters)
5. a 'thumbnail image' (550px width and max 400px height, Illustrator, PowerPoint or jpeg format), which can be used as 'visual title' for the synopsis section of your paper.
6. Please include an author contributions statement after the Acknowledgements section (see <https://www.embopress.org/page/journal/17444292/authorguide>)
7. Please complete the CHECKLIST available at (<https://bit.ly/EMBOPressAuthorChecklist>). Please note that the Author Checklist will be published alongside the paper as part of the transparent process (<https://www.embopress.org/page/journal/17444292/authorguide#transparentprocess>).
8. Please note that corresponding authors are required to supply an ORCID ID for their name upon submission of a revised manuscript (EMBO Press signed a joint statement to encourage ORCID adoption). (<https://www.embopress.org/page/journal/17444292/authorguide#editorialprocess>)

Currently, our records indicate that the ORCID for your account is 0000-0003-1000-7989.

Link Not Available

The system will prompt you to fill in your funding and payment information. This will allow Wiley to send you a quote for the article processing charge (APC) in case of acceptance. This quote takes into account any reduction or fee waivers that you may be eligible for. Authors do not need to pay any fees before their manuscript is accepted and transferred to the publisher.

*** PLEASE NOTE *** As part of the EMBO Press transparent editorial process initiative (see our Editorial at <https://dx.doi.org/10.1038/msb.2010.72>), Molecular Systems Biology publishes online a Review Process File with each accepted manuscripts. This file will be published in conjunction with your paper and will include the anonymous referee reports, your point-by-point response and all pertinent correspondence relating to the manuscript. If you do NOT want this File to be published, please inform the editorial office at msb@embo.org within 14 days upon receipt of the present letter.

Reviewer #1:

The current manuscript by Frauenstein et al. presents a proteomics approach to dissect molecular

signaling pathways through the simultaneous detection of protein complexes and post-translational modification. The authors call this methodology MIP-APMS (Modifications, Interactions and Phenotypes by Affinity Purification Mass Spectrometry) and demonstrate proof-of-concept through application of 19 baits (i.e. protein complexes) in monocytes (both immortalized and primary). MIP-APMS combines tagging of bait proteins of interest through a rapid lentivirus approach (His tags used here), purification of protein complexes, LC-MS-based proteins detection and bioinformatic analysis of obtained results. The methodology, mainly through tricks with spectral matching, attempts to detect both protein complexes and all possible post-translational modifications at the same time. Computational analysis of the resulting data is aimed to provide detailed insights into immune cell relevant signaling nodes by providing both information on protein complex identify and post-translational modifications. Rapid genetic and/or pharmacologic intervention will then enable to specifically investigate specific signaling nodes. Some higher-level proof-of-concept results are shown in the manuscript.

Overall this is an interesting and technically rigorous manuscript that attempts to find an appropriate balance between technology, systems biology and proof-of-concept follow-up of novel discoveries. This manuscript presents a large amount of data and is appropriate for Molecular Systems Biology. Below I suggest a couple of minor modifications for the authors to further strengthen this manuscript.

Comments:

- 1) It's certainly interesting that putative interactors and PTMs are analyzed at the same time. Pulldowns are somewhat less complex than cellular lysates which would then suggest that PTMs can be detected and in an unbiased manner (i.e. without enrichment). I suggest that the authors describe the data a bit more careful.
 - a. These pulldowns are actually quite complex: > 4000 proteins seem to be detected in each bait purification. This will certainly limit the number of PTMs that can be effectively detected in each analysis. Many will be missed and would only be detectable if tedious enrichments are performed (as discussed by the authors).
 - b. I assume the dependent peptide algorithm in MaxQuant is similar to an open search (i.e. MSFragger). It might be interesting to run such an algorithm for comparative reasons - at least on a subset of these data to provide additional confirmation.
 - c. Supplemental Table S1 was not completely clear. I assume the authors ran multiple replicates and possibly also a background controls (i.e. lysates without the His-tag)? It might be appropriate to show this table with more granularity and show results from all the replicates, rather than a merged dataset?
- 2) Figure 2F/G: These figures and resulting data need more explanation. As presented data interpretation is unclear. What exactly went into this heatmap? How do we get from the correlation clusters to these temporal, quantitative profiles?
- 3) There are several data normalizations throughout the manuscript that require more explanation.
 - a. For example, the data in Fig 3A ("The interactome of TRAF2 (measured 16x in biological replicates) compared against all other pull-downs in the control group"). Does this mean that a pulldown of a given bait is compared to the pulldowns from all the other baits in this study? So basically, bait controls are used to infer bait specific association. Are there some steps to remove simple non-specific bead interaction? These data may have been buried somewhere in the manuscript, but were unclear to this reviewer.
 - b. Fig 3C: For these dynamic protein network analyses a certain prey is normalized to the bait at different time-points/treatments. Intuitively this makes sense, but is the abundance of the bait

actually stable in these dynamic experiments and if not how would this change results?

Overall this is an interesting manuscript that presents a pipeline to rapidly generate tagged bait proteins through viral transduction. The focus is on 19 baits in immortalized and primary immune cells, providing a significant resource for immunologists, this alone makes this paper useful to the community. Besides individual candidates, the novelty of this study is the parallel generation and analysis of dynamic protein complexes (and PTMs). Proof-of-concept validation provides confidence of the generated data and the MIP-APMS technology.

Reviewer #2:

In this manuscript, Frauenstein et al describe a method in which they pull down a number of known key players in signalling pathways downstream of Toll-like receptors (TLR). They analyse the pulldowns not only for protein changes but also for post-translational modifications. They provide a number of examples, including 19 proteins followed by dynamics around TRAF2 complex changes upon TLR activation. They then show that knocking out proteins of the TAK1 complex affects NF κ B signalling and finally, they show that three different inhibitors of MAPK14 affect the protein complex of this protein.

Major points:

1. The authors try to sell AP MS and the subsequent search for PTMs in the pulldowns as a new method. This has been done since many years in the community and therefore, this is not a new method and it definitely does not need a new acronym. If required, this reviewer can start looking for papers that are at least 10 years old where this was done. It appears that the authors believe that their "streamlined" cell line generation is worth highlighting, but then many others, including Anne-Claude Gingras have shown similar approaches for AP-MS.
2. There is an overall connecting story missing. The manuscript reads as a collection of many, many experiments that are somehow pulled together.

Minor points:

1. Supplementary tables should contain the intensity values of all replicates. Suppl Table 1 for example has only the "difference" and the p-value.
2. Having values with 5 digits in the supplementary tables makes relatively little sense as the last 3 are definitely not accurate.
3. The transduction part of the paper is valuable, but too long. The authors may want to shorten this part significantly.
4. I don't understand figure 2G. which protein pulldowns are we looking at that see reductions in the PTMs and the complex partners?
5. Figure 3A: please make sure that 8,9,10 aren't overlaying in the volcano plot.
6. Figure 3C&E. some of the points in this figure are identical in their values. Did the authors have indeed such a reproducibility?
7. Figure 3: please add the protein ID to D. it is not obvious which proteins these PTMs belong to.
8. ISGylation of Traf2: it is quite harsh mutating away lysines. Did the authors find any GlyGly modifications on these residues?
9. Considering that the authors didn't prove that the change in NF κ B activation in K>R mutants is really down to ISGylation, I suggest that the statement is a little less strong: Please change "Functional analysis of the K->R mutants revealed reduction in NF κ B activation for K277R, K320R, K364R and K389R mutants, indicating ISG15 as a positive regulator downstream of TLR2." To "Functional analysis of the K->R mutants revealed reduction in NF κ B activation for K277R, K320R,

K364R and K389R mutants, suggesting that ISGylation of TRAF2 may act as a positive regulator downstream of TLR2."

10. Figure 4 and the associated paragraph: it is long known (PMID: 10675530) that TAK1 plays a major role in NFkB signalling and that the knockout has reduced NFkB activation (PMID: 16186825). While making all these CRISPR knockouts was certainly a lot of work, this chapter does not seem to provide much novelty.

11. How many replicates were made of every pulldown. It wasn't clear from the manuscript.

12. What did the authors do when there was only 1 or 2 missing values? Did the authors still impute values?

13. Please add a list to PRIDE explaining what each RAW file is. I could not find this.

Reviewer #3:

Summary:

In their manuscript "Identification of Covalent Modifications Regulating Immune Signaling Complex Composition and Phenotype," Frauenstein et al. have created a framework to define "protein communities" - PPIs and PTMs associated with affinity purified complexes - and investigate their rearrangement in response to specific perturbations. To fully test the boundaries of these methods, the authors have chosen to interrogate molecular signaling in human monocyte cell lines and primary cells. Starting from the cloning of GOIs into His-tagged lentiviral constructs for transduction into a broad spectrum of cell types, to the optimization of affinity purification and sample preparation for MS, and MS analysis to simultaneously investigate PPIs and the PTMs of purified proteins, these steps collectively form MIP-APMS (Modifications, Interactions and Phenotypes by Affinity Purification Mass Spectrometry), a "fast and cost-effective pipeline" to characterize PPIs and PTMs in an iterative manner.

The authors have delineated a thoughtful strategy to clone GOIs for AP-MS studies. Using a previously published method for cloning, SLiCE, a lentivirus vector was modified to temper gene expression (modified weak PGK promoter) and mediate compatibility with commercial cloning systems (flanking ~ 100 bp attL sites). A major advantage of this protocol is that PTM analysis requires no additional enrichment steps; instead, modifications are preserved through the mild lysis conditions employed, and their identification is enabled through the high sequence coverage obtained for bait proteins. Dependent searches, performed through PEAKS and MaxQuant, identify ~ 50 novel PTMs. Incorporation of a variable modification search allowed the identification of dynamic N-term phosphorylation of TRAF2, with verification of these sites via phosphor variants revealing significant interactome remodeling. Frauenstein et al. further demonstrate the utility of MIP-APMS to profile drugs and identify potential off target effects by probing alterations of PPI networks following treatment with small molecule inhibitors.

Recommendation:

Although the underlying premise is commendable, the actual execution and results fall short of expectations due, in part, to technical limitations; moreover, the individual parts are not generally novel while their sum is overstated with conclusions overreaching at times. Beyond PPIs, MIPAPMS reveals both bait and prey PTMs. However, the correlations between PPIs and PTMs are generally not convincingly clear in the time course experiments. Nevertheless, some of the approaches and subsequent findings are interesting (especially isgylation of TRAF2); therefore, we would recommend a qualified acceptance pending significant revisions to clarify text and figures and additional experiments to further support conclusions.

Major Comments:

1) The author's do not incorporate any negative controls into their AP-MS workflow, instead removing proteins common across samples as background interactors

a) Do the author's incorporate intensity or abundance differences between samples in determining what gets labeled a background interactor?

b) Interconnected proteins (e.g. TRAF2 and BIRC2) are also baits; how does this affect determination of background proteins?

c) Similarly, were any scoring systems used when determining interactors (SAINT, compPASS)? We would suggest incorporating negative controls along with established algorithms (SAINT, compPASS) for determining interactors, particularly as the authors are introducing MIP-APMS as a novel methodology. Also, are there any challenges in incorporating PTM analysis with these algorithms? We would appreciate a comment from the authors.

2) The authors identify ~4,000 interactors in an experiment, which seems high. Can this be attributed to the less-stringent lysis methods used to capture PTMs?

3) Do the author's variable modification searches (phosphorylation, acetylation, methylation) match the modification searches captured by the dependent searches using MaxQuant and PEAKS?

4) Fig. S4D and Fig. 5D: It is important to show total p38 (MAPK14) across all samples and that phosphorylation increase is not primarily a change in protein.

5) Fig. 4: MAP3K7

a) Fig. 4B and 4C:

i) Do grey boxes in Fig. 4C indicate missing values?

ii) In general, no clear upward or downward PPI trends are observed across time points (except perhaps 427_Phospho). Does this reflect the rapid dynamics of the signaling pathways?

iii) Do asterisks indicate significant hits at all "timepoints"?

b) Fig. 4B and 4C: The authors indicated interplay between PPIs and PTMs, specifically highlighting the loss of TAB1 and SNX17 PPIs due to MAP3K7 Ser389 phosphorylation. Changes do not clearly correlate in the same time points or to the same degree.

i) In the absence of follow up experiments to confirm this connection (such as the S to D/G point mutations in Fig. 3), this is a bit premature.

ii) We were also curious about ARHGEF18 and 389_Pi. This looks similar to SNX17 in heat map, but is not indicated to be significant.

c) Fig. 4D: CRISPR KO in monocytic U937 NFkB reporter cells - It is important to confirm gene KO/loss of protein by TIDE analysis/western blot, especially since effect is only partial. We did not find any mention of this in the text, legend, or materials and methods.

Minor Comments:

1) Can the authors comment on the effect the dependent search has on false discovery rates

(FDR) - as this is a less complex dataset than analysis of whole cell lysate, for example, is the effect on FDR minimized?

2) How similar are the results obtained with PEAKS and MaxQuant?

3) Are the author's employing any sort of stringency cutoffs (e.g. minimum number of peptides) for PTMs identified using the variable search - this information is provided at the protein level, but I'm not clear if the same cutoffs are applied for PTMs ("Each quantified protein had to be identified with more than one peptide and in more than 60% of replicates of at least one cell line to be considered valid")?

4) It would be helpful to include a table with the number of PTMs and intensity for each modification. Is there any downside to high bait enrichment (do prey IDs suffer?)

5) Please specify western blot protocol and source of antibodies used in Fig. S4D and Fig. 5D in Materials and Methods.

6) MIP-APMS: the P stands for Phenotypes, the authors also talk about "biochemical and phenotypic validation in primary human immune cells". To us, "phenotype" doesn't sound right here. We would rather say "functional validation", "intracellular pathway dissection" or "network topologies".

7) Page 10, first line: "To further explore the utility of MIP-APMS for discovery, ..." We would appreciate more precision about which kind of discovery.

8) Fig. 1D: Please explain in more detail in the main text for clarity.

9) Fig. 2:

a) Fig. 2A: Please add more space between bar charts to separate 'novel vs unknown' (blue/green) from total interactors (grey) or place bait labels between bar charts.

b) Fig. 2B:

i) Please reference Table S2 in the main text and/or figure legend for a clear list of proteins in the 19 interactomes.

ii) Upon closer examination, found that nodes are actually labeled; however, blue text could not be discerned over blue background for novel interactions. Please change the text color to make it visible.

iii) Are the data for TRAF2, MAP3K8, and MAPK14 the same shown in the volcano plots (Fig. 3A, 4A, and 5C)? The number of interactions do not seem the same. If there is a difference, please clarify the reason.

c) Fig. 2C and 2D:

i) Do these figures show both bait and prey PTMs? (89 PTMs across 19 baits)

ii) In Table S3, prey PTMs are listed for 8 baits. Please summarize prey PTMs within Fig. 2C and/or 2D. Since ~ 30% of PTMs are on prey, this would further support simultaneous PPI and PTM analysis. Most figures focus primarily on bait PTMs.

d) Fig. 2E:

i) Please label x-axis to indicate "acetylation", "methylation" and "phosphorylation" for additional clarity.

ii) The difference between novel and described should be presented in a clearer way. Red outline may be difficult to see.

10) Fig. 3: TRAF2

- a) Fig. 3A and 3B: Please refer to general comments on figure presentation (below).
- b) Fig. 3B-3E: When are UT samples taken relative to 5, 15, and 30 minute time points? Is it essentially time 0?
- c) Fig. 3B: Aside from an increase in TANK, ELP2 shows a clear, relatively static decrease with activation. Is ELP2 change not significant during time course? Does the phosphorylation status change this interaction?
- d) Fig. 3D: 7_Phospho has very small, but significant changes. Is 11_Phospho significant by t-test?
- e) Fig. 3F: ISG15 shows as an interactor, but it is actually a Ub-like covalent modification. Can Ub and other similar proteins be detected like this, instead of "Ub remnant antibody" enrichment? Could the author's comment on any difficulties involved in identifying these novel modifications without incorporating an enrichment step?
- f) Fig. 3J: Should fold differences in primary macrophages be similar to 5F and 5I? Also, we were not clear on the cell line used for 3A-3I? Was it U937?

11) Fig. 4: MAP3K7 - Please refer to general comments on figure presentation (below) for Fig. 4A, 4B and 4D.

Fig. 4A: Please adjust data point labels. They overlap and are not visible at the bottom.

12) Fig. 5: MAPK14 - MIP-APMS enable identification of signaling network rearrangements?

- a) Both p38 and MAPK14 names are used, please be consistent in figures and text.
- b) Fig. 5B: Please indicate the cell line used for the WB in legend.
- c) Fig. 5C: Please refer to general comments on figure presentation (below)
- d) Fig. 5D: Heat map
We were unclear if the values shown are the intensity values obtained for each condition, or if they were compared to a control? It seems DMSO is up/down depending on interaction?
- e) Fig. 5E and 5F: P3C4 is not clearly indicated in the text, legend or materials and methods - "cellular activation" is briefly mentioned in regards to Fig 5E. Please clarify accordingly.
- f) Fig. 5F: Ser2 phosphorylation appears to be unaffected by inhibitors and would be a good control along with total p38. Is there a p38 Ser2 antibody? In Fig. 5B, is phospho p38 against pY182?

Supplement:

1) Fig. S1

- a) S1A: GFP panel is difficult to see.
- b) S1C: Label data points overlap. Please adjust to clarify.

2) Characterization of induced cell lines:

S2A: Is this based on peptides against bait protein itself? Is "bait expression" the sum of endogenous, untagged protein and tagged bait?

3) Comparison of enrichment strategies: We are interested in the details of the various enrichment strategies and would appreciate the inclusion of additional details.

- a) Please add details on epitope tags compared in Materials and Methods (e.g. 1x FLAG or 3x FLAG; 1xStrep-tag II or 2xStrep-tag II; 6xHis).
- b) How do backgrounds of FLAG, Strep, and His beads compare?
- c) Fig. S3A:

- i) Were on-bead digests for FLAG beads and His elutions with imidazole performed and analyzed? If so, how do they compare to on-bead digests of Strep and His beads and FLAG and Strep elutions?
- ii) Were proportional amounts of bead and elution analyzed?
- iii) How complete were the various bead elutions? Would incomplete elution result in lower numbers of significantly interacting proteins?
- d) Fig. S3B: Sequence coverage is not that different. Are there differences in intensities?
- e) Fig. S3F: Please label Y-axis.

4) Fig. S4D: WB for MAPK14 phospho-p38 following PAM3CSK4 activation.

- a) MAPK14 (right) shows clear p38 increase, while wild type (left) is not as clear.
- b) Do two bands in MAPK14 represent both untagged (endogenous) and tagged MAPK14?
- c) It is important to show total p38 (MAPK14) across all samples to support the conclusion that the phosphorylation increase is not primarily a change in protein levels.

5) Fig. S5

- a) Fig. S5A: Why is ISG15 increased under denaturing conditions? Should it be the same as native. Since there is less total protein under denaturing conditions, is more ISG15 able to be detected by MS?
- b) Fig. S5D and S5E: MAP3K7 and TRAF2
Similar to Fig. 5D, we were unclear if the values shown are the intensity values obtained for each condition, or if they were compared to a control? It seems DMSO is up/down depending on interaction?

Minor corrections:

Please adjust following on page 5:

- 1) Change attI to attL.
- 2) NEB Builder to NEBuilder
- 3) Huttlin, Ting et al. 2015 is "doubly cited"; one should be deleted

On page 8:

- 1) "... and Induces ..."

General comments on figure presentation: Applies to Figures 3-5 and S3A, S6-S19.

1) Is a volcano plot the best way to show interactome of specific baits (bait vs all others)? It would be nice for all interactomes to be presented as in Figure 2B.

2) Suggestions to clarify labeling of interactomes, volcano plots and heat maps:

- a) Novel vs Known is clearer than Interactor (blue) vs STRING/Biogrid (green). (Fig. 2A, 2B, 3A, 3B, 4A, 4B and 5C)
- b) Do numbers have any significance aside from being a way to label data points on volcano plot and use heat map labels as a legend? Bait is not always first.
- c) Numbers overlap in volcano plots in Fig. 4A and supplement. Please adjust accordingly.
- d) For PPI heat maps, it might be useful to add color conventions defined in volcano plot to protein labels.

Revision: MSB-2020-10125

Identification of Covalent Modifications Regulating Immune Signaling Complex Composition and Phenotype

We thank the reviewers for their valuable comments and constructive feedback. We have carefully addressed all reviewer's concerns point by point. In the revised manuscript, we include a series of novel experiments and analyses, as suggested. We have also changed several sections in the main text and methods to better present the focus and novelty of our work. We hope you agree that our extensive revision has strengthened the manuscript significantly.

Major improvements include:

- (1) Systematic comparison of open search (e.g. MsFragger, PEAKS/ Taggraph, dependent peptides of MaxQuant) to specific search for identification of post-translational modifications
- (2) In-depth analysis of the interdependency of PPIs and PTMs in TLR2 activation time-course experiments
- (3) Combination of MIP-APMS with GlyGly enrichment to identify GlyGly-sites on TRAF2
- (4) Detailed validation and explanation of our experimental strategy to distinguish between bait/prey and background proteins, including a comparison to the SAINT algorithm
- (5) Confirmation of CRISPR-KOs of MAP3K7 interactors by Western Blot
- (6) Validation of TLR2 cell activation and inhibition with MAPK14 small molecules by Western blots using antibodies against p38 and phospho-p38
- (7) Addition, redesign and optimization of Figures, including an interaction network for every bait protein visualizing interconnected interaction hubs

A point-to-point response is provided below and all edits are highlighted in the text.

Reviewer #1:

The current manuscript by Frauenstein et al. presents a proteomics approach to dissect molecular signaling pathways through the simultaneous detection of protein complexes and post-translational modification. The authors call this methodology MIP-APMS (Modifications, Interactions and Phenotypes by Affinity Purification Mass Spectrometry) and demonstrate proof-of-concept through application of 19 baits (i.e. protein complexes) in monocytes (both immortalized and primary). MIP-APMS combines tagging of bait proteins of interest through a rapid lentivirus approach (His tags used here), purification of protein complexes, LC-MS-based proteins detection and bioinformatic analysis of obtained results. The methodology, mainly through tricks with spectral matching, attempts to detect both protein complexes and all possible post-translational modifications at the same time. Computational analysis of the resulting data is aimed to provide detailed insights into immune cell relevant signaling nodes by providing both information on protein complex identify and post-translational modifications. Rapid genetic and/or

pharmacologic intervention will then enable to specifically investigate specific signaling nodes. Some higher-level proof-of-concept results are shown in the manuscript.

Overall this is an interesting and technically rigorous manuscript that attempts to find an appropriate balance between technology, systems biology and proof-of-concept follow-up of novel discoveries. This manuscript presents a large amount of data and is appropriate for Molecular Systems Biology. Below I suggest a couple of minor modifications for the authors to further strengthen this manuscript.

Comments:

1) It's certainly interesting that putative interactors and PTMs are analyzed at the same time. Pulldowns are somewhat less complex than cellular lysates which would then suggest that PTMs can be detected and in an unbiased manner (i.e. without enrichment). I suggest that the authors describe the data a bit more careful.

a. These pulldowns are actually quite complex: > 4000 proteins seem to be detected in each bait purification. This will certainly limit the number of PTMs that can be effectively detected in each analysis. Many will be missed and would only be detectable if tedious enrichments are performed (as discussed by the authors).

This is an interesting point, which we now elaborate further. We agree, that the proteomic complexity may limit the number of identified sub-stoichiometric PTMs on bait/ prey proteins to some degree. However, we detected PTMs on more than 70% of the bait proteins and also dynamic regulations upon cell activation. In response to this comment and a question raised by reviewer 2, we have now performed a 'tedious' additional enrichment after His-IMAC to detect selected substoichiometric PTMs on selected baits. In particular, we performed GlyGly enrichment post TRAF2 IP and identified two covalently modified sites (see Fig. 3I). We have added a passage in the discussion, to raise the awareness to challenges associated with MIP-APMS. It reads:

"As MIP-APMS does not include a second enrichment step, the method preferentially quantifies abundant PTMs on bait and prey proteins. Ubiquitinylation and neddylation are known as sub-stoichiometric PTMs and special biochemical enrichment or MS methods are commonly used for their detection (Kim, Bennett et al. 2011, Wagner, Beli et al. 2011, Bustos, Bakalarski et al. 2012, Hansen, Tanzer et al. 2021). We show that MIP-APMS combined with Gly-Gly enrichment facilitates the bait-centric identification ubiquitinylation or isgylation sites, exemplified for TRAF2."

b. I assume the dependent peptide algorithm in MaxQuant is similar to an open search (i.e. MSFragger). It might be interesting to run such an algorithm for comparative reasons - at least on a subset of these data to provide additional confirmation.

We thank the reviewer for this question. The dependent search algorithm implemented in MaxQuant detects mass shifts in comparison to unmodified peptides, which can be the result of unknown chemical modifications, protease cleavages or amino acid substitutions. For the revision, we have compared the

dependent peptide algorithm of MaxQuant and other open search algorithms (MSFragger and Peaks/Taggraph). Exemplified for MAPK14, we show that most modifications were identified by MSFragger, followed by Peaks/Taggraph and MaxQuant (Figure EV4G). We observed the highest overlap between MSFragger and MaxQuant (300), followed by MSFragger and Peaks/Taggraph (102) and MaxQuant and Peaks/Taggraph (65) (Figure EV4H). 26 peptides were identified by all three algorithms, corresponding to 21 unique modifications. We highlight the 26 reproducibly identified modifications across algorithms (dependent peptides, PEAKS/taggraph and MSFragger) in Figure EV4I. Interestingly, when we compared the MaxQuant specific search to the open search engines, of the 15 PTMs identified on MAPK14 (Phosphorylation, Acetylation and Methylation) only 6 were detected with PEAKS/Taggraph and 2 by PEAKS/Taggraph and specific search (Figure EV4J).

We are pasting the new panels of Figure EV4 below:

The corresponding legend reads:

“(G) Number of modifications on MAPK14 detected with the indicated open search algorithms in at least one replicate. Open search was performed in MaxQuant (dependent peptide mode), MS Fragger and Peaks/Taggraph on the drug mode of action dataset of MAPK14 (Figure 5). Mass offsets detected at

distinct amino acid positions were kept separately. (H) Venn diagram showing the overlap of modifications identified with open search algorithms in at least one replicate. (I) Number of distinct modifications on MAPK14, identified in at least 70% of the replicates with MaxQuant (red), Peaks/Taggraph (violet), MaxQuant & Peaks/Taggraph (green) and with less than 70% valid values (turquoise) in all searches. (J) Comparison of MaxQuant specific searches for MAP3K7, MAPK14 and TRAF2 to dependent peptide analysis (MaxQuant) and PEAKS/Taggraph. Identified and quantified sites are depicted in turquoise, not identified sites in red.”

We have edited the main text accordingly:

“Furthermore, an unbiased analysis of covalent peptide modifications using the dependent peptide algorithm in MaxQuant, the string-based search algorithm Taggraph - based on a de novo search in PEAKS and MS-Fragger (Devabhaktuni, Lin et al. 2019) revealed a series of less well-described covalent modifications on MAPK14 (Figure EV4G). 26 modifications were shared between search engines (2.3 % of all modifications for dependent peptides, 1.5 % PEAKS/taggraph and 0.9 % MS-Fragger, Figure EV4H). Out of these 26 modifications, 6 were reproducibly identified and quantified in all replicates (Figure EV4I). To distinguish biologically regulated from other - for example - sample preparation-introduced modifications, we quantified the identified modifications upon cell activation with specific searches in MaxQuant. Notably, only MAPK14 phosphorylation was differentially regulated between conditions. Moreover, acetylation, methylation and phosphorylation detected on TRAF2, MAPK14 and MAP3K7 with specific searches were missed by open searches (dependent peptides of MaxQuant and PEAKS/Taggraph) (Figure EV4J). This demonstrates that MIP-APMS can discover novel PTMs in signaling complexes, however, comparisons across search engines and confirmation with specific search strategies are advisable to increase confidence.”

c. Supplemental Table S1 was not completely clear. I assume the authors ran multiple replicates and possibly also a background controls (i.e. lysates without the His-tag)? It might be appropriate to show this table with more granularity and show results from all the replicates, rather than a merged dataset?

We have addressed this point and we are now reporting all replicates and respective controls in Supplementary Table 1/ Table EV1. Additionally, we have updated the other Supplementary Tables and report the LFQ intensity values for all replicates.

2) Figure 2F/G: These Figures and resulting data need more explanation. As presented data interpretation is unclear. What exactly went into this heatmap? How do we get from the correlation clusters to these temporal, quantitative profiles?

We thank the reviewer for this question and have revisited the analysis and visualization of this panel. Now, all PPIs (357) and PTMs (37) identified on baits and preys from time course experiments, with at least 70% valid values in all replicates are included in the heat map. The median intensity (log₂) of each PPI and time-point was calculated and z-scored. For the hierarchical clustering, we performed a row-wise correlation analysis (Pearson correlation) and plotted smoothed medians (in blue) of 7 emerging clusters for all PPIs and PTMs (method=loess, $y \sim x$, confidence interval=0.95). We now show all 7 clusters in Fig 2G, as pasted below.

We have also included an additional Supplementary Table (Table EV3, tab PPI PTM interdependency) showing the results of the correlation analysis and clusters for each PPI and PTM. Additionally, we have updated the Figure legend and the methods section for clarity. We have updated the main text as such:

“Next, to study PTM and PPI interdependency, we correlated all PTM and PPI intensities and clustered them unbiasedly over the time-course of TLR2 activation (Fig. 2G, Table EV3). We detect the dynamic co-regulation on both molecular layers (PPIs and PTMs), identifying correlating and anti-correlating PTMs and PPIs during signaling pathway activation. We identified seven clusters with distinct kinetics, some peaking early (Fig. 2H, Cluster: 4,5) and others late (Fig. 2H, Cluster: 7) upon pathway activation, as well as up- (Fig. 2H, Cluster: 4,5,6,1) vs. down-regulated (Fig. 2H, Cluster: 1,2,3) PTMs and PPIs. Interestingly, interactors identified in more than one MIP-APMS experiment (e.g., CDC37: Cluster 1) were in close network proximity. Our approach facilitated an unbiased discovery of time-resolved molecular connections between dynamic PTMs and PPIs, exemplified by the correlated interaction of MAP3K8 interactors (NFKB1, NFKB2) and NFKB1 phosphorylation (Cluster 4,7), or the anti-correlated relationships of phosphorylation of the C-terminal kinase domain of AKT1 and the interaction with CDC37 (Cluster1, 5). This demonstrates that the sensitivity and robustness of MIP-APMS enables the simultaneously determination of cellular signaling network rearrangements by PPIs, PTMs, and their interplay.”

The Figure legend now reads:

“(G) Unsupervised clustering (Pearson correlation) of the z-scored intensity profiles of all PPIs (357

) and PTMs (37) upon TLR2 activation, partitioned in seven clusters. (H) Dynamic profiles of co-regulated PTMs and PPIs with close network proximity, from the indicated clusters; median z-scored intensity of each time-point (blue: median, grey: confidence interval =0.95, method: loess), n, number of proteins in clusters

1 to 7. Selected proteins from each cluster are indicated, with the bait proteins in parentheses. See also Figure EV3, Table EV2 for PPIs and Table EV3 for PTMs.”

The method section now reads:

“Unsupervised clustering

Intensities of dynamically regulated PPIs (357) and PTMs (178) upon TLR 2 activation were filtered for at least 70% valid values and normalized per time-point (PPIs to bait protein intensity and PTMs to protein intensity of the modified protein as explained above). The median of each time-point was calculated and then Z-scored. Pearson correlation was calculated between each of the PPIs and PTMs and results were visualized by hierarchical clustering. The data was clustered and median z-scored intensities (Confidence interval: 0.95) were plotted against the time-course of TLR2 activation (method=loess, $y \sim x$). N shows the number of PPIs/PTMs corresponding to each cluster.”

3) There are several data normalizations throughout the manuscript that require more explanation. a. For example, the data in Fig 3A (“The interactome of TRAF2 (measured 16x in biological replicates) compared against all other pull-downs in the control group”). **Does this mean that a pulldown of a given bait is compared to the pulldowns from all the other baits in this study?** So basically, bait controls are used to infer bait specific association. **Are there some steps to remove simple non-specific bead interaction?** These data may have been buried somewhere in the manuscript, but were unclear to this reviewer.

We take the opportunity to explain our bioinformatics strategy for interactor calling in more detail. We use a concept termed AE-MS introduced by (Keilhauer, Hein et al. 2015), which is based on the premise, that the majority of quantified proteins in the interactome derives from non-specifically bead-binding proteins (= background) under low detergent (= low stringent) conditions. In AE-MS, background proteins - as opposed to real bait/ prey proteins - have stable intensities across experiments. Accordingly, standard statistical testing of all protein LFQs per bait versus all LFQs from all other MIP-APMS experiments discriminates significantly enriched outlier proteins versus background binders. Notably, in contrast to other approaches, low affinity-binding proteins are preserved under low stringency washing conditions. Moreover, quantification by a label-free (such as MaxLFQ) algorithm - which assumes a certain degree of similarity between samples - is more accurate. Accordingly, we did not apply additional in silico processing steps, to remove non-specific bead binders.

We now show three additional subfigures in the Supplement to illustrate the distinct differences between specific and background protein LFQ intensities and p-values.

The new Figure EV4A-C legend reads:

“(A) Pie-chart showing the proportion of quantified bait and prey proteins (blue) in comparison to background binding proteins (red) (B) Boxplots with enrichment differences (log2) of bait and prey versus background proteins. (C) Boxplots with p-values (-log10) of bait and prey proteins versus background proteins.”

Additionally, we have updated the main-text for clarity. It now reads:

“We generated 19 transgenic monocytic U937 cell lines and analyzed them with MIP-APMS, as described above. This identified and quantified an average of 4106 proteins per measurement, including non-specifically binding proteins as expected for non-stringent AP-MS conditions (Trinkle-Mulcahy, Boulon S Fau - Lam et al. , Rees, Lowe et al. 2011). We observed high median intra-bait and inter-bait Pearson correlations (>0.9) between biological replicates (Figure EV3I) and between different cell lines (Figure EV3J). This highlights the overall reproducibility of the devised workflow. To discriminate specifically interacting proteins from background binders common to all baits, we compared enrichments from single vs. all other cell lines with a standard statistical test (two-sided T-test) at a stringent false discovery rate (FDR) of 1% to correct for multiple-hypothesis testing (Hein, Hubner et al. 2015, Keilhauer, Hein et al. 2015, Hubel, Urban et al. 2019). This resulted in a small fraction of significantly interacting proteins (378 proteins in total, with a median of 16 interactors per bait) compared to a large proportion of background binders (Table EV1, Figure EV4A), Notably, distinct protein intensity differences and p-values clearly distinguish specific bait and prey from unspecific background proteins (Figure EV4B, C).”

b. Fig 3C: For these dynamic protein network analyses a certain prey is normalized to the bait at different time-points/treatments. Intuitively this makes sense, but is the abundance of the bait actually stable in these dynamic experiments and if not how would this change results?

We reanalysed the data to address this interesting point. We have calculated the difference between the replicates of each bait protein and plotted the distribution in a boxplot (Figure EV3H: Bait intensity variation). The LFQ intensity of the bait proteins is largely stable across the replicates and the median difference in time series experiments varies by 0 +/- 0.89. This tight correlation between bait and prey intensity profiles has been previously observed by Keilhauer et al. (Keilhauer, Hein et al. 2015). As we normalize prey intensities to bait intensity (as explained in detail in materials and methods), even in case a bait protein changed abundance (e.g. in case of degradation upon cellular activation), we assume MIP-APMS is not prone to false-positive dynamic interactor calling.

Overall this is an interesting manuscript that presents a pipeline to rapidly generate tagged bait proteins through viral transduction. The focus in on 19 baits in immortalized and primary immune cells, providing a significant resource for immunologist, this alone makes this paper useful to the community. Asides from individual candidates, the novelty of this study is the parallel generation and analysis of dynamic protein complexes (and PTMs). Proof-of-concept validation provide confidence of the generated data and the MIP-APMS technology.

We thank the reviewer for the enthusiasms for our work.

Reviewer #2:

In this manuscript, Frauenstein et al describe a method in which they pull down a number of known key players in signalling pathways downstream of Toll-like receptors (TLR). They analyse the pulldowns not only for protein changes but also for post-translational modifications. They provide a number of examples, including 19 proteins followed by dynamics around TRAF2 complex changes upon TLR activation. They then show that knocking out proteins of the TAK1 complex affects NFkB signalling and finally, they show that three different inhibitors of MAPK14 affect the protein complex of this protein.

Major points:

1. The authors try to sell AP MS and the subsequent search for PTMs in the pulldowns as a new method. This has been done since many years in the community and therefore, this is not a new method and it definitely does not need a new acronym. If required, this reviewer can start looking for papers that are at least 10 years old where this was done. It appears that the authors believe that their "streamlined" cell line generation is worth highlighting, but then many others, including Anne-Claude Gingras have shown similar approaches for AP-MS.

We appreciate this comment. We do acknowledge that PTMs were detected previously in AP-MS experiments and have already cited 20 papers both, AP-MS and PTM-MS only in the introduction, an altogether four publications from Anne-Claude Gingras. To still aim to address this point, we have extended the text as follows to more clearly endorse previous work (Page 3, paragraph 4):

“By contrast, PTM analyses generally focus on a single modification type (e.g., phosphorylation), wherein modified peptides of all cellular proteins are affinity-enriched. Alternatively, in order to classify PTMs on specific proteins, affinity purification mass spectrometry (APMS) approaches with stringent washing and lysis conditions have been performed at the expense of PPI elucidation (Stutz, Kolbe et al. 2017, Pankow, Bamberger et al. 2019, Karayel, Tonelli et al. 2020).”

Conversely, we also include a dedicated passage about previously performed AP-MS approaches including transduction, citing the Gingras and Gygi labs (Page 5, paragraph 3):

“As shown previously, employing lentiviral transduction for amphotrophic gene transfer extends the scope from readily transfectable cell lines, e.g., human embryonic kidney (HEK) cells, to non-dividing and terminally differentiated cells of primary origin (Huttlin, Ting et al. 2015, Samavarchi-Tehrani, Abdouni et al. 2018).”

While carefully re-screening the literature, however, we did not find a study where the dynamic interplay of PTMs and PPIs is studied systematically upon cell activation. In particular for immune or primary cells, streamlined approaches are challenging and have not been reported. We hope the reviewer agrees that our method describes a distinct step towards MS-based investigations of signal transduction mechanisms in physiological and primary cell systems.

We are surprised about the comment referring to the use of acronyms for methods. We do not intend to promote our method by giving it a name. We were inspired by common practice in the field of proteomics

to name methods or tools, e.g. AE-MS (Keilhauer, Hein et al. 2015), EASYPhos (Humphrey, Karayel et al. 2018), CiRCus (Seefried, Schmidt et al. 2019), SEC-SWATH-MS (Bludau, Heusel et al. 2020).

In our opinion, names facilitate unambiguous denominations and discussions among researchers. Accordingly, MIP-APMS is 'just' a name with no further implications what our method provides or does not provide. We carefully re-examined the manuscript to omit statements, where merits of MIP-APMS may be perceived subjectively. In this context, we replaced the term 'framework' by 'method' or 'strategy' throughout the text. Importantly, we do not mention MIP-APMS in the title (although this would help to reduce the word count), but use it as an abbreviation for the method in the manuscript.

2. There is an overall connecting story missing. The manuscript reads as a collection of many, many experiments that are somehow pulled together.

We interpret this comment as a suggestion to optimize the flow and provide additional guidance to the reader. We have addressed the comment in two aspects. To assist the reader in navigating through our manuscript, we have now included a graphical roadmap connecting individual experiments in Figure EV1A, as pasted below.

Moreover, we have carefully edited and expanded the final section of the introduction to highlight our focus on monocyte immune activation and inhibition, and how our method provides unique insights.

“We evaluated and technically optimized all steps of MIP-APMS, comprising (1) the epitope-tagging of proteins of interest and mammalian cell transduction, (2) affinity purification conditions for optimal interaction network and PPI enrichment, (3) followed by MS-based PTM and PPI quantification and identification and (4) ultimate biochemical and phenotypic validation of interactors and PTMs in primary human immune cells. Integration of multiple MIP-APMS experiments generates dynamic signal transduction networks and pinpoints time-resolved co-regulations of PTMs and PPIs in sequential signal transduction steps. We show the discovery potential of our pipeline by interrogating dynamically assembling protein communities in human monocyte immune signaling using Toll-like receptor (TLR) 2 activation and MAP kinase MAPK14 inhibition as paradigms. Our screen encompassing 19 protein

complexes identified more than 50 previously undescribed PTMs, including phosphorylation, acetylation, methylation, isgylation as well as other less well described chemical modifications and elucidated an interaction network spanning more than 300 PPIs. We used the modular concept of MIP-APMS to test emerging data-driven hypotheses to validate PTMs and PPIs regulating immune signaling in reporter and primary cells. In this way, MIP-APMS enables the streamlined validation of cross-talk between different layers of protein regulation with broad applicability.”

Minor points:

1. Supplementary tables should contain the intensity values of all replicates. Suppl Table 1 for example has only the "difference" and the p-value

As suggested, we have now included the LFQ intensity values of all replicates in the supplementary tables.

2. Having values with 5 digits in the supplementary tables makes relatively little sense as the last 3 are definitely not accurate.

We have updated the supplementary table and rounded the numbers to two digits by changing the view settings.

3. The transduction part of the paper is valuable, but too long. The authors may want to shorten this part significantly.

We thank the reviewer for this suggestion. We have now significantly shortened the transduction part of the manuscript. It now reads:

“Universal Cloning and Transduction Strategy. To enable interrogation of signaling cascades, we employed a cost-effective method for epitope-tagging of proteins with a restriction enzyme-free approach, called restriction enzyme-free seamless ligation cloning extract (SLiCE) cloning (Zhang, Werling et al. 2012). A modified weak phosphoglycerate kinase (PGK) promoter controls the expression of the GOIs, which are flanked by attL sites. Thereby, our vector system is compatible with commercial DNA assembly cloning strategies such as the NEBuilder platform or Gateway, which had been used before (Lambert, Tucholska et al. 2014). As shown previously, employing lentiviral transduction for amphotrophic gene transfer extends the scope from readily transfectable cell lines, e.g., human embryonic kidney (HEK) cells, to non-dividing and terminally differentiated cells of primary origin (Huttlin, Ting et al. 2015, Samavarchi-Tehrani, Abdouni et al. 2018). In particular for application with primary immune cells, transduction is advantageous as other methods can activate innate immune signaling pathways and induce cell death (Fernandes-Alnemri, Yu et al. 2009, Hornung, Ablasser et al. 2009, Gaidt, Ebert et al. 2017). As a relevant and challenging experimental model system, we chose human monocytes, because these cells are not easily transfectable and execute a broad spectrum of cellular programs by the dynamic intracellular propagation of molecular signals downstream of cell surface receptors. For method development and phenotypic screening, we employed the monocytic cell-line U937 and validated our results with primary cells. We achieved 92 (+/- 5)% cellular transduction efficiency after antibiotic marker selection (Figure EV1B). We further demonstrated the universality of our approach with primary human macrophages differentiated

from peripheral monocytes (Figure EV1C, Figure EV1D, Table EV1) and primary human T-cells (Figure EV1E, Table EV1).

4. I don't understand Figure 2G. which protein pulldowns are we looking at that see reductions in the PTMs and the complex partners?

We thank the reviewer for this question. We have now updated Figure 2H and G to show a total of seven co-regulated clusters. We also included a more detailed description of the analysis in the main text and methods section, as this was also requested by reviewer 1. In these panels, we show a selection, of regulated PPIs and PTMs, with the respective bait protein indicated in brackets. In addition, Table EV3 tab Interdependencies PPIs and PTMs now shows, which proteins and PTMs are localized in each cluster.

We have updated the main text as follows:

“Next, to study PTM and PPI interdependency, we correlated all PTM and PPI intensities and clustered them unbiasedly over the time-course of TLR2 activation (Fig. 2G, Table EV3). We detect the dynamic co-regulation on both molecular layers (PPIs and PTMs), identifying correlating and anti-correlating PTMs and PPIs during signaling pathway activation. We identified seven clusters with distinct kinetics, some peaking early (Fig. 2H, Cluster: 4,5) and others late (Fig. 2H, Cluster: 7) upon pathway activation, as well as up- (Fig. 2H, Cluster: 4,5,6,1) vs. down-regulated (Fig. 2H, Cluster: 1,2,3) PTMs and PPIs. Interestingly, interactors identified in more than one MIP-APMS experiment (e.g., CDC37: Cluster 1) were in close network proximity. Our approach facilitated an unbiased discovery of time-resolved molecular connections between dynamic PTMs and PPIs, exemplified by the correlated interaction of MAP3K8 interactors (NFKB1, NFKB2) and NFKB1 phosphorylation (Cluster 4,7), or the anti-correlated relationships of phosphorylation of the C-terminal kinase domain of AKT1 and the interaction with CDC37 (Cluster1, 5). This demonstrates that the sensitivity and robustness of MIP-APMS enables the simultaneously determination of cellular signaling network rearrangements by PPIs, PTMs, and their interplay.”

The Figure legend now reads:

“(G) Unsupervised clustering (Pearson correlation) of the z-scored intensity profiles of all PPIs (357) and PTMs (37) upon TLR2 activation, partitioned in seven clusters. (H) Dynamic profiles of co-regulated PTMs and PPIs with close network proximity, from the indicated clusters; median z-scored intensity of each time-point (blue: median, grey: confidence interval =0.95, method: loess), n, number of proteins in clusters 1 to 7. Selected proteins from each cluster are indicated, with the bait proteins in parentheses. See also Figure EV3, Table EV2 for PPIs and Table EV3 for PTMs.”

The method section now reads:

“Unsupervised clustering

Intensities of dynamically regulated PPIs (357) and PTMs (178) upon TLR 2 activation, were filtered for at least 70% valid values and normalized per time-point (PPIs to bait protein intensity and PTMs to protein intensity of the modified protein as explained above). The median of each time-point was calculated and then Z-scored. Pearson correlation was calculated between each of the PPIs and PTMs and results were visualized by hierarchical clustering. The data was divided into seven clusters and median z-scored

intensities (Confidence interval: 0.95) were plotted against the time-course of TLR2 activation (method=loess, $y \sim x$). N shows the number of PPIs/PTMs corresponding to each cluster.”

5. Figure 3A: please make sure that 8,9,10 aren't overlaying in the volcano plot.

We have adjusted the plot, as suggested.

6. Figure 3C&E. some of the points in this figure are identical in their values. Did the authors have indeed such a reproducibility?

We have carefully re-inspected and re-plotted all Figures and now also show the intensity values of all replicates in the respective supplementary tables. The table includes the measured and normalized values. Indeed, upon normalization to bait protein amount as described in materials and methods, we sometimes achieve exceptional reproducibility.

7. Figure 3: please add the protein ID to D. it is not obvious which proteins these PTMs belong to.

As suggested, we have now added an ID to D.

8. ISGylation of Traf2: it is quite harsh mutating away lysines. Did the authors find any GlyGly modifications on these residues?

We thank the reviewer for this question. We have now performed GlyGly enrichment on a TRAF2 MIP-APMS experiment and were able to identify GlyGly sites on TRAF2 (see Fig. 3I). We have updated the main text, accordingly:

“ISG15 is a ubiquitin-like protein, that covalently modifies target proteins on lysine residues in a process called isgylation (Loeb and Haas 1992, Zhang and Zhang 2011). After tryptic digest, isgylated peptides harbor GlyGly modifications on lysines, that can be readily detected by LC-MS/MS. As we did not directly detect GlyGly-modified peptides, we combined MIP-APMS with GlyGly enrichment and indeed identified two GlyGly modification sites on TRAF2 (Positions K27, K320) (Fig. 3I). To deduce the impact of isgylation on the TRAF2 interaction network, we performed site-directed mutagenesis of TRAF2 lysines and subjected the K->R mutant cell-lines to MIP-APMS. Out of the total 32 K->R mutants, 5 showed strong (more than 4x) and significant depletion of ISG15 in the TRAF2 complex (Fig. 3J, Table EV4). Interestingly, the most regulated site - K320 – was also identified by our initial GlyGly enrichment, suggesting an ISGylation of TRAF2.”

Additionally, we have updated the discussion:

“As MIP-APMS does not include a second enrichment step, the method preferentially quantifies abundant PTMs on bait and prey proteins. Ubiquitylation, neddylation and isgylation are known as substoichiometric PTMs and special biochemical enrichment or MS methods are commonly used for their detection (Kim, Bennett et al. 2011, Wagner, Beli et al. 2011, Bustos, Bakalarski et al. 2012, Hansen, Tanzer et al. 2021). We show that MIP-APMS combined with Gly-Gly enrichment facilitates the bait-centric identification of ubiquitin-like modification sites, exemplified for TRAF2.”

9. Considering that the authors didn't prove that the change in NFkB activation in K>R mutants is really down to ISGylation, I suggest that the statement is a little less strong: Please change "Functional analysis of the K->R mutants revealed reduction in NFkB activation for K277R, K320R, K364R and K389R mutants, indicating ISG15 as a positive regulator downstream of TLR2." To "Functional analysis of the K->R mutants revealed reduction in NFkB activation for K277R, K320R, K364R and K389R mutants, suggesting that ISGylation of TRAF2 may act as a positive regulator downstream of TLR2."

We thank the reviewer for this comment. We have changed the wording as requested.

10. Figure 4 and the associated paragraph: it is long known (PMID: 10675530) that TAK1 plays a major role in NFkB signalling and that the knockout has reduced NFkB activation (PMID: 16186825). While making all these CRISPR knockouts was certainly a lot of work, this chapter does not seem to provide much novelty.

No doubt, TAK1 is well known as a major regulator of NFkB signaling. This was one of the reasons, why we were interested in characterizing its interactome in the first place. As we found several proteins not previously described as TAK1 interaction partners (ARHGEF18, FOSB), we were curious, whether their knockout by CRISPR would impact NFkB signaling. Accordingly, in the CRISPR-KO experiment, MAP3K7 (TAK1), TLR2 and MYD88 serve as positive controls for the novel candidate protein KOs. As this apparently was not so clearly stated, we have rephrased the passage and also included the suggested reference in the main text. It now reads:

"To further explore the utility of MIP-APMS for discovery of new interactors, we evaluated functional interactions of MAP3K7. MAP3K7 (TAK1) is a central kinase of the MAPK signaling pathway, with crucial roles in the activation of TRAF6 downstream of TLRs and other receptors (B-cell receptor, TNF receptor) (Landström 2010) and known as a major regulator of NFkB signaling (Sato, Sanjo et al. 2005)."

11. How many replicates were made of every pulldown. It wasn't clear from the manuscript.

We thank the reviewer for this question. We have now included a tab in the Table EV1 indicating how many replicates were carried out per pulldown. 16 replicates were carried out for the pulldowns of BCL10, BIRC2, CASP7, IFIH1, MAP3K7, RIPK2, RIPK3, TRAF1, TRAF6. 15 replicates were carried out for AKT1, MAP3K8, MAPK14, TIRAP, TRAF2. 14 replicates were carried out for MYD88 and SYK. 4 replicates were performed for CASP4, CASP5 and IKBKE.

12. What did the authors do when there was only 1 or 2 missing values? Did the authors still impute values?

We take the opportunity to clarify this point. Missing values were replaced by random numbers calculated from a normal down-shifted distribution (down-shift: 1.8), also in the case of only 1 or 2 missing values per protein. In this regard, our analysis is very stringent as potentially more interactors could be reported when applying distinct strategies to compute missing values. This information is also included in the Methods section.

13. Please add a list to PRIDE explaining what each RAW file is. I could not find this.

We thank the reviewer for raising this topic. We have now added an additional supplementary table for Pride raw files.

Reviewer #3:

Summary:

In their manuscript "Identification of Covalent Modifications Regulating Immune Signaling Complex Composition and Phenotype," Frauenstein et al. have created a framework to define "protein communities" - PPIs and PTMs associated with affinity purified complexes - and investigate their rearrangement in response to specific perturbations. To fully test the boundaries of these methods, the authors have chosen to interrogate molecular signaling in human monocyte cell lines and primary cells. Starting from the cloning of GOIs into His-tagged lentiviral constructs for transduction into a broad spectrum of cell types, to the optimization of affinity purification and sample preparation for MS, and MS analysis to simultaneously investigate PPIs and the PTMs of purified proteins, these steps collectively form MIP-APMS (Modifications, Interactions and Phenotypes by Affinity Purification Mass Spectrometry), a "fast and cost-effective pipeline" to characterize PPIs and PTMs in an iterative manner.

The authors have delineated a thoughtful strategy to clone GOIs for AP-MS studies. Using a previously published method for cloning, SLiCE, a lentivirus vector was modified to temper gene expression (modified weak PGK promoter) and mediate compatibility with commercial cloning systems (flanking ~ 100 bp attL sites). A major advantage of this protocol is that PTM analysis requires no additional enrichment steps; instead, modifications are preserved through the mild lysis conditions employed, and their identification is enabled through the high sequence coverage obtained for bait proteins. Dependent searches, performed through PEAKS and MaxQuant, identify ~ 50 novel PTMs. Incorporation of a variable modification search allowed the identification of dynamic N-term phosphorylation of TRAF2, with verification of these sites via phosphor variants revealing significant interactome remodeling. Frauenstein et al. further demonstrate the utility of MIP-APMS to profile drugs and identify potential off target effects by probing alterations of PPI networks following treatment with small molecule inhibitors.

Recommendation:

Although the underlying premise is commendable, the actual execution and results fall short of expectations due, in part, to technical limitations; moreover, the individual parts are not generally novel while their sum is overstated with conclusions overreaching at times. Beyond PPIs, MIPAPMS reveals both bait and prey PTMs. However, the correlations between PPIs and PTMs are generally not convincingly clear in the time course experiments. Nevertheless, some of the approaches and subsequent findings are interesting (especially isgylation of TRAF2); therefore, we would recommend a qualified acceptance pending significant revisions to clarify text and figures and additional experiments to further support conclusions.

We thank the reviewer for the thorough evaluation and appreciation of our approaches and findings. We have carefully addressed all points, clarified text and Figures and performed additional experiments to support our conclusions as suggested.

Major Comments:

1) The author's do not incorporate any negative controls into their AP-MS workflow, instead removing proteins common across samples as background interactors.

We thank the reviewer for raising this point, which was also touched upon by reviewer 1. We take the opportunity to explain our bioinformatics strategy for interactor calling in more detail. We use a concept termed AE-MS introduced by (Keilhauer, Hein et al. 2015), which is based on the premise, that the majority of quantified proteins in the interactome derives from non-specifically bead-binding proteins (= background) under low detergent (= low stringent) conditions. In AE-MS, background proteins - as opposed to real bait/ prey proteins - have stable intensities across experiments. Accordingly, standard statistical testing of all protein LFQs per bait versus all LFQs from all others MIP-APMS experiments discriminates significantly enriched outlier proteins versus background binders. Notably, in contrast to other approaches, low affinity-binding proteins are preserved under low stringency washing conditions. Moreover, quantification by a label-free (such as MaxLFQ) algorithm - which assumes a certain degree of similarity between samples - is more accurate. Accordingly, we did not apply additional in silico processing steps, to remove non-specific bead interactors.

We now show three additional subfigures in the Supplement to illustrate the distinct differences between specific and background protein LFQ intensities and p-values.

The new Figure EV4A-C legend reads:

“(A) Pie-chart showing the proportion of quantified bait and prey proteins (blue) in comparison to background binding proteins (red) (B) Boxplots with enrichment differences (log₂) of bait and prey versus background proteins. (C) Boxplots with p-values (-log₁₀) of bait and prey proteins versus background proteins.”

Additionally, we have updated the main-text for clarity. It now reads:

We generated 19 transgenic monocytic U937 cell lines and analyzed them with MIP-APMS, as described above. This identified and quantified an average of 4106 proteins per measurement, including non-specifically binding proteins as expected for non-stringent AP-MS conditions (Trinkle-Mulcahy, Boulon S Fau - Lam et al. , Rees, Lowe et al. 2011). We observed high median intra-bait and inter-bait Pearson correlations (>0.9) between biological replicates (Figure EV3I) and between different cell lines (Figure EV3J). This highlights the overall reproducibility of the devised workflow. To discriminate specifically interacting proteins from background binders common to all baits, we compared enrichments from single vs. all other cell lines with a standard statistical test (two-sided T-test) at a stringent false discovery rate

(FDR) of 1% to correct for multiple-hypothesis testing (Hein, Hubner et al. 2015, Keilhauer, Hein et al. 2015, Hubel, Urban et al. 2019). This resulted in a small fraction of significantly interacting proteins (378 proteins in total, with a median of 16 interactors per bait) compared to a large proportion of background binders (Table EV1, Figure EV4A), Notably, distinct protein intensity differences and p-values clearly distinguish specific bait and prey from unspecific background proteins (Figure EV4B, C).

a) Do the author's incorporate intensity or abundance differences between samples in determining what gets labeled a background interactor?

We integrate differences in intensities and abundance as described by (Keilhauer, Hein et al. 2015). Briefly, in a two-step procedure, first, a two-sided T-test discriminates significantly interacting from background binding proteins, and second, a two-sided T-test discriminates dynamically changing interactors. To clarify this point, we have added a detailed description of the interactor calling in the methods section on page 29 paragraph 4.

b) Interconnected proteins (e.g. TRAF2 and BIRC2) are also baits; how does this affect determination of background proteins?

We appreciate the comment. We determine interactors unbiasedly by statistical testing without a priori knowledge. Bait proteins or interconnecting proteins are not treated differently. We now illustrate how this analysis strategy impacts interconnectedness, using SYK and MAPK14 as examples, as they share 32 common interactors: We compared enrichments and p-values from statistical tests (two-sided T-test) from single vs. all other cell lines against single vs. all other cell lines excluding SYK/ MAPK14. In the first example, where both MAPK14 and SYK are in the same control group (large control group), enrichment and p-value (-log₁₀) of bait and prey proteins are smaller than for the second example, where MAPK14 and SYK are removed from the control group (small control group) (Figure EV4D, pasted below). Accordingly, our workflow prioritizes bait-specific interactors, as PPIs present in multiple interactomes may not be enriched significantly.

The Figure legend reads:

“Impact of interconnected interactors on enrichment differences and p-values: Interactor calling was performed for indicated proteins using big (with SYK and MAPK14) and small (without SYK and MAPK14) control groups. Differences of all significant interactors and p-values are shown individually for SYK and MAPK14 pulldowns.”

We have updated the main text, accordingly:

“MIP-APMS prioritizes bait-specific preys, as proteins enriched in multiple experiments – including interconnected interactors - show lower enrichment differences and p-values (Figure EV4D) by unbiased statistical interactor calling (see Methods).”

c) Similarly, were any scoring systems used when determining interactors (SAINT, compPASS)? We would suggest incorporating negative controls along with established algorithms (SAINT, compPASS) for determining interactors, particularly as the authors are introducing MIP-APMS as a novel methodology. Also, are there any challenges in incorporating PTM analysis with these algorithms? We would appreciate a comment from the authors.

We thank the reviewer for this interesting comment. We have now integrated a SAINT analysis for a subset of our data. By uploading our data to the SAINT-based Crapome pipeline (<https://reprint-apms.org/>), we analyzed the pulldown of MAPK14 versus control (cell line transduced with His-Tag only). We used standard parameters and spectral counts for quantification, although the latter is less quantitative than ion intensities. We compared the results by plotting the SAINT likelihood (probability of true interaction between two proteins) against the p-value obtained from our T-test-based approach (Figure EV4E, pasted below). The SAINT and T-test approach reveal five common top outliers. The T-test based approach is slightly more sensitive as it captures one additional known interactor of MAPK14 and two novel

interactors, for which low SAINT probabilities are reported. The integration of PTM data into the SAINT algorithms would require the upload of PTM data together with interaction data, which is currently not supported.

Figure EV4E shows the comparison between the SAINT algorithm and the T-test based approach, we use for MIP-APMS:

The new Figure legend reads:

“(E) Saint probability (probability of the interactor being a true interactor) plotted against the p-value from the Student’s T-test comparing MAPK14 versus control (transduced with His-Tag only). Significant interactors from T-test analysis are colored in turquoise, SAINT interactors (75% probability of interactor being a true interactor) are above the horizontal line at 0.75. Known interactors are colored green.”

We have updated the main text:

“Additionally, we compared our LFQ intensity and T-test-based strategy to the results of the SAINT algorithm (spectral count based) exemplary for MAPK14 and identified largely similar interactors (Figure EV4E).”

We have updated the methods section, accordingly:

“MAPK14 His IPs and controls (U937 transduced with His-Tag) were performed in triplicates and uploaded to the SAINT based Crapome server (<https://reprint-apms.org>)(Mellacheruvu, Wright et al. 2013). As Experiment Type, we selected single step epitope tag AP-MS and spectral counts as quantitation Type. As external controls, we selected PBMC (Cell/tissue Type), agarose (affinity support) and Q Exactive (Instrument type). The primary empirical fold change score (FC-A) was calculated by user controls using average for combining replicates (Number virtual controls=10). The secondary fold-change score was calculated by all controls (user + external controls) using geometric mean for combining replicates (Number virtual controls=3). The probabilistic SAINT Score was calculated by user controls (combining replicates: average) and 10 virtual controls. Saint options were 2000 n-burn, 4000 n-iter, 0 LowMode, 1 MinFold and 1 Normalize.”

2) The authors identify ~4,000 interactors in an experiment, which seems high. Can this be attributed to the less-stringent lysis methods used to capture PTMs?

We thank the reviewer for these questions, which were also touched upon by reviewer 1. We indeed identify around 4000 proteins per IP, however, we acknowledge, that most of these proteins are background binders. A reason for the high background is the low stringent lysis/ washing method. We now show that the intensity difference and p-values of background binding proteins are largely stable and distinct from specifically interacting proteins (Figure EV4B, EV4C). Notably, in contrast to other approaches, such as tandem affinity purifications, low affinity-binding proteins are preserved under low stringency washing conditions and quantification by a label-free (such as MaxLFQ) algorithm - which assumes a certain degree of similarity between samples - is more accurate.

We have updated the main text accordingly:

“We generated 19 transgenic monocytic U937 cell lines and analyzed them with MIP-APMS, as described above. This identified and quantified an average of 4106 proteins per measurement, including non-specifically binding proteins as expected for non-stringent AP-MS conditions (Trinkle-Mulcahy, Boulon S Fau - Lam et al. , Rees, Lowe et al. 2011).”

3) Do the author's variable modification searches (phosphorylation, acetylation, methylation) match the modification searches captured by the dependent searches using MaxQuant and PEAKS?

We thank the reviewer for this interesting question. We have now compared the outputs of the MaxQuant modification specific search, the dependent peptide search of MaxQuant and the PEAKS/taggraph search for the PTMs on MAPK14, TRAF2 and MAP3K7 in detail. Unexpectedly, of the 15 PTMs identified with specific searches, we only recapitulate 6 with the dependent search and PEAKS/taggraph and 2 additional PTMs in the intersection of PEAKS/taggraph and specific searches.

To illustrate the comparison between open and specific searches, we now include Figure EV4J:

The new Figure legend reads:

“(J) Comparison of MaxQuant specific searches for MAP3K7, MAPK14 and TRAF2 to dependent peptide analysis (MaxQuant) and PEAKS/Taggraph. Identified and quantified sites are depicted in turquoise, not identified sites in red.”

We have updated the main text accordingly:

“Moreover, acetylation, methylation and phosphorylation detected on TRAF2, MAPK14 and MAP3K7 with specific searches were missed by open searches (dependent peptides of MaxQuant and PEAKS/Taggraph) (Figure EV4J). This demonstrates that MIP-APMS can discover novel PTMs in signaling complexes, however, comparisons across search engines and confirmation with specific search strategies are advisable to increase confidence.”

4) Fig. S4D and Fig. 5D: It is important to show total p38 (MAPK14) across all samples and that phosphorylation increase is not primarily a change in protein.

We thank the reviewer for this request. We have now included the protein intensity of MAPK14 to the respective Figures 5F and 5G – we assume the reviewer meant Figure 5F and 5G, which shows the dynamic change of interactors (5F) and PTMs (5G) in response to drug perturbation of MAPK14. Our analysis strategy of dynamic interactors/ dynamic PTMs includes the normalization of interactor intensity/ modified peptide intensity to the total protein amount of MAPK14 before statistical testing to adjust for protein abundance changes, as reported in detail in the methods section. For dynamic PTM analysis, phosphorylation of pS2 can be used as an internal control of the normalization, as quantification of this site remains unaffected.

5) Fig. 4: MAP3K7

a) Fig. 4B and 4C:

i) Do grey boxes in Fig. 4C indicate missing values?

Yes, grey boxes indicate missing values. We have clarified this in the legend, now.

ii) In general, no clear upward or downward PPI trends are observed across time points (except perhaps 427_Phospho). Does this reflect the rapid dynamics of the signaling pathways?

Based on PPI studies in other experimental settings, both, gain and loss of PPIs can be expected upon pathway activation. While PPI dynamics occur at various temporal scales, our method allows a temporal resolution of 5 min intervals. On average, we detected two statistically significant dynamic PPIs and one dynamic PTM per bait (as summarized now in Figure EV5A; Tables EV2 and EV3).

iii) Do asterisks indicate significant hits at all "timepoints"?

Asterisks indicate a significant regulation for at least one time point. Individual significantly regulated time points are shown in Table EV2, EV3. We have updated the Figure legends for clarity. It now reads:

“(C) Heatmap of MAP3K7 PTMs (phosphorylation) upon activation, with significant hits (t-test, p-value < 0.05) in at least one time-point denoted with an asterisk.”

b) Fig. 4B and 4C: The authors indicated interplay between PPIs and PTMs, specifically highlighting the loss of **TAB1 and SNX17 PPIs due to MAP3K7 Ser389 phosphorylation**. Changes do not clearly correlate in the same time points or to the same degree

c) In the absence of follow up experiments to confirm this connection (such as the S to D/G point mutations in Fig. 3), this is a bit premature.

We agree and have edited the paragraph to avoid overinterpretation of the data. It now reads:

“Upon TLR2 activation, TAB1 and SNX17 were depleted from the MAP3K7 complex (Fig. 4C), while phosphorylation of Ser389 on MAP3K7 increased (Fig. 4D). This revealed dynamic regulation of both PTMs and PPIs during pathway execution.”

ii) We were also curious about ARHGEF18 and 389_Pi. This looks similar to SNX17 in heat map, but is not indicated to be significant.

A p-value < 0.05 was used for calling significance. Accordingly, ARHGEF18 dynamics and phosphorylation of 427 were not significant with p-values of 0.23 (-log₁₀: 0.59) and 0.39 (-log₁₀: 0.40), respectively. All protein and peptide intensities and p-values can now be inspected in the Tables EV2 and 3.

c) Fig. 4D: CRISPR KO in monocytic U937 NFKB reporter cells - It is important to confirm gene KO/loss of protein by TIDE analysis/western blot, especially since effect is only partial. We did not find any mention of this in the text, legend, or materials and methods.

We thank the reviewer for this comment. We now show Western Blots to verify protein knockouts with an orthogonal method (see Figure EV5F).

We have updated the main text, accordingly:

‘We verified CRISPR-KO of ARHGEF18 and FOSB by Western Blot analysis (Figure EV5F).’

We also updated the Figure legend:

“(F) Western Blot analysis of CRISPR-KO reporter cell-lines with antibodies against MAP3K7, ARHGEF18, FOSB. Tubulin was used as a loading control.”

Minor Comments:

1) Can the authors comment on the effect the dependent search has on false discovery rates (FDR) - as this is a less complex dataset than analysis of whole cell lysate, for example, is the effect on FDR minimized?

The FDR for dependent peptides is determined separately from the FDR on base peptides. Also, here a target-decoy method is applied, which counts both forward and reverse hits and applies a cutoff when an FDR of 1% is reached. Naturally, this FDR estimate becomes imprecise if only a low number of dependent peptides is found. The limitation is given by estimating the number of reverse hits. Since we found in total 80 000 dependent peptides and > 1100 on MAPK14 we should be able to estimate 1% of these with sufficient accuracy.

2) How similar are the results obtained with PEAKS and MaxQuant?

We thank the reviewer for this question. We now compare the dependent peptide algorithm of MaxQuant and other open search algorithms (MSFragger and Peaks/ Taggraph). Exemplified for MAPK14, we show that most modifications were identified by MSFragger, followed by Peaks/Taggraph and MaxQuant (Figure EV4G). We observed the highest overlap between MSFragger and MaxQuant (300), followed by MSFragger and Peaks/Taggraph (102) and MaxQuant and Peaks/Taggraph (65) (Figure EV4H). 26 peptides were identified by all three algorithms, corresponding to 21 unique modifications. We highlight the 26 reproducibly identified modifications across algorithms (dependent peptides, PEAKS/taggraph and MSFragger) in Figure EV4I.

We are pasting the new panels of Figure EV4 below:

The corresponding legend reads:

“(G) Number of modifications on MAPK14 detected with the indicated open search algorithms in at least one replicate. Open search was performed in MaxQuant (dependent peptide mode), MS Fragger and Peaks/Taggraph on the drug mode of action dataset of MAPK14 (Figure 5). Mass offsets detected at distinct amino acid positions were kept separately. (H) Venn diagram showing the overlap of modifications identified with open search algorithms in at least one replicate. (I) Number of distinct modifications on MAPK14, identified in at least 70% of the replicates with MaxQuant (red), Peaks/Taggraph (violet), MaxQuant & Peaks/Taggraph (green) and with less than 70% valid values (turquoise) in all searches.”

We have edited the main text accordingly:

“Furthermore, an unbiased analysis of covalent peptide modifications using the dependent peptide algorithm in MaxQuant, the string-based search algorithm Taggraph based on a de novo search in PEAKS and MS-Fragger (Devabhaktuni, Lin et al. 2019) revealed a series of less well-described covalent modifications on MAPK14 (Figure EV4G). 26 modifications were shared between search engines (2.3 % of all modifications for dependent peptides, 1.5 % PEAKS/taggraph and 0.9 % MS-Fragger, Figure EV4H). Out of these 26 modifications, 6 were reproducibly identified and quantified in all replicates (Figure EV4I). To distinguish biologically regulated from other - for example - sample preparation-introduced modifications, we quantified the modifications upon cell activation with specific searches in MaxQuant. Notably, only MAPK14 phosphorylation was differentially regulated between conditions. Moreover, acetylation, methylation and phosphorylation detected on TRAF2, MAPK14 and MAP3K7 with specific searches were missed by open searches (dependent peptides of MaxQuant and PEAKS/Taggraph) (Figure EV4J). This demonstrates that MIP-APMS can discover novel PTMs in signaling complexes, however, comparisons

across search engines and confirmation with specific search strategies are advisable to increase confidence.”

3) Are the author's employing any sort of stringency cutoffs (e.g. minimum number of peptides) for PTMs identified using the variable search - this information is provided at the protein level, but I'm not clear if the same cutoffs are applied for PTMs ("Each quantified protein had to be identified with more than one peptide and in more than 60% of replicates of at least one cell line to be considered valid")?

PTMs are filtered by a 1% FDR on the level of peptide-spectrum matches (PSMs), as reported in the Methods section. As PTM identifications and quantifications are based on unique spectral evidences, further cut-off criteria are conceptually challenging. Accordingly, we report PTMs on all peptides passing the PSM scoring in Table EV3. Significant PTM changes require a minimum of 3 quantifications in at least one of the two compared time-points. We updated the methods section to clarify our strategy. It now reads:

“Conversely, intensities of modified peptides of each replicate were normalized to the intensity of the respective protein intensity, decreasing the total coefficient of variation (data not shown). Only PTMs quantified in at least 3 replicates of at least one time point were considered for quantitative analyses. No imputation was performed.”

4) It would be helpful to include a table with the number of PTMs and intensity for each modification. Is there any downside to high bait enrichment (do prey IDs suffer?)

High bait intensity favors the detection of PTMs and can therefore be perceived as advantage. We now include the intensity values of the modifications from the TLR2 time-course experiment in the Table EV3.

5) Please specify western blot protocol and source of antibodies used in Fig. S4D and Fig. 5D in Materials and Methods.

We have included the relevant information in Materials and Methods. It now reads:

“One million U937 cells were stimulated, washed in PBS, and lysed in buffer (4% SDS, 40 mM HEPES (pH 7.4, 10 mM DTT) supplemented with protease inhibitors (Sigma-Aldrich, 4693159001). Samples were centrifuged (16 000 g, 10 min), LDS sample buffer was added to a final concentration of 1x and the supernatant was incubated (5min, 95%). Proteins were separated on 12% Novex Tris-glycine gels (Thermo Fisher Scientific, XP00120BOX) and transferred onto PVDF membranes (Merck Millipore, IPVH00010) or Nitrocellulose membranes (Amersham, 10600002). Membranes were blocked in 5% BSA in PBST, and antibodies were diluted in 2% BSA in PBST. Antibodies used for immunoblotting were as follows (diluted 1:1000): Phospho-p38 MAPK (Thr180/Tyr182) Antibody (Cell Signaling, 9211), GAPDH (14C10) Rabbit mAb (Cell Signaling, 2118), p38 MAPK (R&D, AF8691), ARHGEF18 (Sigma, HPA042689), MAP3K7 (R&D, MAB5307), FOSB (R&D, AF2214), Anti-rabbit IgG, HRP-linked Antibody (Cell Signaling, 7074).”

6) MIP-APMS: the P stands for Phenotypes, the authors also talk about "biochemical and phenotypic validation in primary human immune cells". To us, "phenotype" doesn't sound

right here. We would rather say "functional validation", "intracellular pathway dissection" or "network topologies".

We thank the reviewer for the suggestions. 'P' refers to (functional and molecular) phenotypes, which can be tested with our method. The latter is key difference to established workflows, focusing on biochemical evidences in signaling pathways and networks. Accordingly, we prefer to keep the name "MIP-APMS".

7) Page 10, first line: "To further explore the utility of MIP-APMS for discovery, ..." We would appreciate more precision about which kind of discovery.

We have changed the text, as suggested. It now reads:

"To further explore the utility of MIP-APMS for discovery of new interactors, we evaluated functional interactions of MAP3K7."

8) Fig. 1D: Please explain in more detail in the main text for clarity.

We edited the text as suggested, which now reads:

"Dynamic Signaling Network Analysis. To study how signaling networks rearrange upon cellular activation, we integrated quantitative PTM and PPI information from multiple MIP-APMS experiments. This enabled quantitative analysis of sequential steps in signal transduction, since it allowed for dynamic PTM and PPI crosstalk to be resolved providing a basis to identify molecular switches in signal transduction networks. We observed enrichments and de-enrichment of prey proteins in protein complex of interest together with dynamically regulated PTMs on both bait and prey proteins (Fig. 1D, regulation up/down)."

9) Fig. 2:

a) Fig. 2A: Please add more space between bar charts to separate 'novel vs unknown' (blue/green) from total interactors (grey) or place bait labels between bar charts.

We have introduced the requested changes.

b) Fig. 2B: i) Please reference Table S2 in the main text and/or figure legend for a clear list of proteins in the 19 interactomes.

We have introduced the requested changes. Table EV2 is now referenced in the main text and in the Figure legend of Figure 2.

ii) Upon closer examination, found that nodes are actually labeled; however, blue text could not be discerned over blue background for novel interactions. Please change the text color to make it visible.

As requested, we have changed the text color. It should be better visible now. Also, we now show excerpts of this graph for each individual pulldown in the respective Figures (Fig.3, Fig.4, Fig.5, Appendix Figure S1-S14) to better visualize the interconnectedness between different pulldowns.

iii) Are the data for TRAF2, MAP3K8, and MAPK14 the same shown in the volcano plots (Fig. 3A, 4A, and 5C)? The number of interactions do not seem the same. If there is a difference, please clarify the reason.

We thank the reviewer for noticing this inconsistency. We have updated the input for the network analysis to match the volcano plots for TRAF2, MAP3K8 and MAPK14.

c) Fig. 2C and 2D:

i) Do these figures show both bait and prey PTMs? (89 PTMs across 19 baits)

Yes, these Figures show both bait and prey PTMs. We have specified this now in the Figure legend.

ii) In Table S3, prey PTMs are listed for 8 baits. Please summarize prey PTMs within Fig. 2C and/or 2D. Since ~ 30% of PTMs are on prey, this would further support simultaneous PPI and PTM analysis. Most Figures focus primarily on bait PTMs.

We thank the reviewer for raising this point. We have included a new panel in Figure 2 (Figure 2D). It shows the percentage of methylations, acetylations and phosphorylations identified on bait/ prey proteins.

We have updated the Figure legend, accordingly:

“D) Percentage of PTMs identified on bait proteins and interactors.”

We have also updated the main text:

“While the majority of PTMs were detected on bait proteins, some (31 PTMs on ten proteins) were also detected on prey proteins (26% of all known PTMs) (Fig.2D).”

d) Fig. 2E:

i) Please label x-axis to indicate "acetylation", "methylation" and "phosphorylation" for additional clarity.

We have introduced the requested changes.

ii) The difference between novel and described should be presented in a clearer way. Red outline may be difficult to see.

We have changed the representation of the data as requested.

10) Fig. 3: TRAF2

a) Fig. 3A and 3B: Please refer to general comments on Figure presentation (below).

We have changed the Figures as requested.

b) Fig. 3B-3E: When are UT samples taken relative to 5, 15, and 30 minute time points? Is it essentially time 0?

UT samples are untouched control cells, which serve a control for all time points in the time course experiments. These samples are not treated with Pam3CSK4 and can be considered a 0 min time point.

c) Fig. 3B: Aside from an increase in TANK, ELP2 shows a clear, relatively static decrease with activation. Is ELP2 change not significant during time course? Does the phosphorylation status change this interaction?

ELP2 enrichment does not change significantly during the time-course. There is a decrease in the median of the abundance, but this decrease is not highly reproducible, see Table EV2. This is the reason, why these dynamics are insignificant at a p-value < 0.05. In the phosphomimetic experiments, ELP2 dynamics are also not significant. Accordingly, we do not define this interaction as phosphorylation dependent (Table EV4).

d) Fig. 3D: 7_Phospho has very small, but significant changes. Is 11_Phospho significant by t-test?

Indeed, also the 11_Phospho site is not significantly regulated, as the site was not identified reproducibly (Table EV3) and is of lower abundance than 7_Phospho. All intensities and p-values can now be inspected in Table EV3.

e) Fig. 3F: ISG15 shows as an interactor, but it is actually a Ub-like covalent modification. Can Ub and other similar proteins be detected like this, instead of "Ub remnant antibody" enrichment? Could the author's comment on any difficulties involved in identifying these novel modifications without incorporating an enrichment step?

This is an interesting point. MIP-APMS can detect ubiquitin and other ubiquitin-like proteins as interactors. To distinguish between ubiquitin and ubiquitin-like proteins functioning as PTMs or interactors, we have pursued multiple strategies including size-exclusion based binding assays, site-directed mutagenesis and MIP-APMS under denaturing conditions. However, in the TRAF2 experiments, we have not observed ubiquitin as a significant interactor, albeit we detect it in the background. We do, however, detect several proteins of the ubiquitin/ neddylation machinery on RIPK3, BCL10 and MAP3K8 as significant interactors: UBA3, UBAP2L, UBE2K, UBE2S.

We have updated the discussion in response to this question:

“As MIP-APMS does not include a second enrichment step, the method preferentially quantifies abundant PTMs on bait and prey proteins. Ubiquitylation, neddylation and isgylation are known as substoichiometric PTMs and special biochemical enrichment or MS methods are commonly used for their detection (Kim, Bennett et al. 2011, Wagner, Beli et al. 2011, Bustos, Bakalarski et al. 2012, Hansen, Tanzer et al. 2021). We show that MIP-APMS combined with Gly-Gly enrichment facilitates the bait-centric identification of ubiquitin-like modification sites, exemplified for TRAF2.”

f) Fig. 3J: Should fold differences in primary macrophages be similar to 5F and 5I?

We thank the reviewer for this interesting question. Did the reviewer mean 3F and 3I? We answer the question assuming this were the implied Figures. As expected for distinct cellular systems, we don't observe exactly the same fold differences as in 3F and 3I. However, we observe the same trend for the

K389R mutant (downwards) and S11D mutant (upwards). In contrast to cell lines, primary cells cannot be kept viable and functional for extended time frames, presumably selection and biological effects are less pronounced.

Also, we were not clear on the cell line used for 3A-3I? Was it U937?

Yes, we have employed the monocytic cell-line U937 in experiments 3A-I. The new Figure legend reads:

“Experiments in A, B, C, D, E, F, G, H, I were performed in U937 cell-lines.”

11) Fig. 4: MAP3K7 - Please refer to general comments on figure presentation (below) for Fig. 4A, 4B and 4D. Fig. 4A: Please adjust data point labels. They overlap and are not visible at the bottom.

We have introduced the requested changes.

12) Fig. 5: MAPK14 - MIP-APMS enable identification of signaling network rearrangements?

a) Both p38 and MAPK14 names are used, please be consistent in figures and text.

We have introduced the requested changes and now only refer to MAPK14.

b) Fig. 5B: Please indicate the cell line used for the WB in legend.

We have introduced the requested changes.

c) Fig. 5C: Please refer to general comments on figure presentation (below)

We have introduced the requested changes.

d) Fig. 5D: Heat map: We were unclear if the values shown are the intensity values obtained for each condition, or if they were compared to a control? It seems DMSO is up/down depending on interaction?

We thank the reviewer for picking up this inconsistency. The intensity values were not normalized to DMSO control. We have now harmonized Figure 5E (former Figure 5D) and Figures EV5G, EV5H (former Figure S5G, H), to the data representation of the other Figures.

e) Fig. 5E and 5F: P3C4 is not clearly indicated in the text, legend or materials and methods - "cellular activation" is briefly mentioned in regards to Fig 5E. Please clarify accordingly.

We have updated the methods section, accordingly. It now reads:

“Cells (5 Mio) were seeded in deep-well 24-well plates. The cells were treated with MAPK14 inhibitors (sorafenib: 10 μ M; skepinone-L: 80 nM; and JX-401: 10 μ M) for 2 h at 37°C under 5% CO₂ in quadruplicate. Inhibitor-treated cells and controls either harvested directly or were activated with PAM3CSK3 (P3C4, 0.5 μ g/mL; Invivogen) for 30 min at 37°C under 5% CO₂. Cells were harvested by centrifugation, and frozen until MIP-APMS.”

We have updated the Figure legend, accordingly. It now reads:

“(F) Intensity profiles of the MAPK14 interactors RPS6KA4 and MAPKAPK2 after treatment with different MAPK14 inhibitors, normalized to MAPK14 bait intensity. Drug mode of action was analyzed in the presence (P3C4, 0.5 ug/mL, 30 min) or absence of P3C4. (G) Intensity profiles of MAPK14 phosphorylation on positions Ser2 and Tyr182 after treatment with different MAPK14 inhibitors, normalized to MAPK14 bait intensity. Drug mode of action was analyzed in the presence (P3C4, 0.5 ug/mL, 30 min) or absence of P3C4.”

f) Fig. 5F: Ser2 phosphorylation appears to be unaffected by inhibitors and would be a good control along with total p38. Is there a p38 Ser2 antibody? In Fig. 5B, is phosphor p38 against pY182?

To our knowledge, there is no antibody available for MAPK14 Ser2. The antibody, we have used, specifically detects pY182.

Supplement:

1) Fig. S1

a) S1A: GFP panel is difficult to see.

We have now added quantitative information to the panel and counted % positive GFP cells in three replicates.

b) S1C: Label data points overlap. Please adjust to clarify.

We corrected the labelling.

2) Characterization of induced cell lines:

S2A: Is this based on peptides against bait protein itself? Is "bait expression" the sum of endogenous, untagged protein and tagged bait?

Protein copy numbers per cell scale according to LFQ intensities (Wiśniewski, Hein et al. 2014). This analysis is based on the fixed ratio between the total histone signal and cellular DNA. An accurate determination of the DNA content in a proteomic sample helps to directly determine the number of cells. The MS signal derived from histones is used as a natural standard in a whole proteome dataset. For quantification, we used the LFQ intensities of the bait proteins (which is based on the peptide intensities). We summarize the intensities of endogenous – untagged – and tagged protein. We now state this explicitly in the Figure legend of Figure EV2A. It now reads:

“Evaluation of bait overexpression on the proteomes of 13 bait cell lines (5752 total protein IDs). Comparison of the copy numbers of overexpressed bait proteins in the non-overexpressing cells (U937 WT) and bait-overexpressing U937 cells. Copy numbers of bait proteins were calculated with the Perseus Plugin based on the LFQ intensity of the respective bait protein. LFQ intensities of endogenous protein and the overexpressed bait proteins are summarized.”

3) Comparison of enrichment strategies: We are interested in the details of the various enrichment strategies and would appreciate the inclusion of additional details. a) Please add details on epitope tags compared in Materials and Methods (e.g. 1x FLAG or 3x FLAG; 1xStrep-tag II or 2xStrep-tag II; 6xHis).

We have included the requested changes. We have used for His-Tag: 9x His, for Flag-tag: 1x Flag, for Strep-tag: 1x Strep-tag II.

b) How do backgrounds of FLAG, Strep, and His beads compare?

We have now included a comparison between backgrounds of His, Strep and Flag beads (Figure EV3B). In comparison to Strep and Flag beads, His beads have the highest number of background proteins.

We have updated the Figure legend. It now reads:

“Comparison of background proteins between His, Flag and Strep beads. Proteins that were identified and quantified at least once were included in the Venn Diagram.”

We have updated the main text. It now reads:

“Highest number of background proteins were identified for His enrichment and we identified > 1000 shared proteins between all three enrichments (Figure EV3B).”

c) Fig. S3A: i) Were on-bead digests for FLAG beads and His elutions with imidazole performed and analyzed? If so, how do they compare to on-bead digests of Strep and His beads and FLAG and Strep elutions?

We have not performed elutions for Flag and His beads, accordingly, we cannot make definitive claims. However, lower intensity of bait protein and higher standard deviation, argues against an elution-based workflow. Moreover, as this is an additional manual step, it may induce additional variation. Additionally, the LFQ profits from a largely comparable background between samples, overall suggesting an advantage of on-bead digests.

ii) Were proportional amounts of bead and elution analyzed?

Yes, proportional amounts of bead and elution were analyzed, which we now state in the Methods section.

iii) How complete were the various bead elutions? Would incomplete elution result in lower numbers of significantly interacting proteins?

It can be assumed that incomplete elution leads to lower number of significantly interacting proteins or reduced bait abundance. We observe that MAPK14 intensity is lower for elution than for strep beads.

d) Fig. S3B: Sequence coverage is not that different. Are there differences in intensities?

We have now analysed the intensities, as requested. Intensities are also comparable between different IP strategies. However, it seems that by including an additional elution step, the reproducibility of bait enrichment is slightly reduced, as standard deviations increase.

This panel is now included in Figure EV3D. We have updated the Figure legend. It now reads:

“(D) Protein intensity (log₂) of MAPK14 after enrichment with different affinity matrices. Each enrichment was performed with equal bead amounts (50 uL each) and 5 Mio cells as input. Each bar represents the median of three experiments.”

We have updated the main text: It now reads:

“However, highest median bait-protein sequence coverage (Figure EV3C), highest intensity of MAPK14 (Figure EV3D) and highest number of significantly interacting proteins were achieved with His-IMAC.”

e) Fig. S3F: Please label Y-axis.

We have introduced the requested changes.

4) Fig. S4D: WB for MAPK14 phospho-p38 following PAM3CSK4 activation.

a) MAPK14 (right) shows clear p38 increase, while wild type (left) is not as clear.

We have repeated the Western blot and used total p38 as requested above as a control.

b) Do two bands in MAPK14 represent both untagged (endogenous) and tagged MAPK14?

Yes, we strongly assume, the second band represents tagged MAPK14.

c) It is important to show total p38 (MAPK14) across all samples to support the conclusion that the phosphorylation increase is not primarily a change in protein levels.

We have now included total MAPK14 as control (see above, Question 4a).

5) Fig. S5

a) Fig. S5A: Why is ISG15 increased under denaturing conditions? Should it be the same as native. Since there is less total protein under denaturing conditions, is more ISG15 able to be detected by MS?

There is indeed more ISG15 enriched under denaturing conditions. We speculate that under denaturing conditions also more ISG15 peptides can be detected by MS as the total proteomic complexity is decreased.

b) Fig. S5D and S5E: MAP3K7 and TRAF2

Similar to Fig. 5D, we were unclear if the values shown are the intensity values obtained for each condition, or if they were compared to a control? It seems DMSO is up/down depending on interaction?

As explained above, (point by point response, page 30, Question 12d) we have now normalized the DMSO control to harmonize the presentation with all other Figures.

Minor corrections:

Please adjust following on page 5:

- 1) Change attl to attL.
- 2) NEB Builder to NEBuilder
- 3) Huttlin, Ting et al. 2015 is "doubly cited"; one should be deleted

On page 8:

- 1) "... and I nduces ..."

We have corrected all typos.

General comments on figure presentation: Applies to Figures 3-5 and S3A, S6-S19.

1) Is a volcano plot the best way to show interactome of specific baits (bait vs all others)? It would be nice for all interactomes to be presented as in Figure 2B.

Volcano plots present an accurate and honest visualization form for all quantitative information of enriched bait and prey proteins and their significance. The reader can directly infer from a volcano plot, how strong an interactor was enriched. We prefer to use volcano plots for the presentation of pairwise comparisons. In addition, we now display each interactome in a small network.

2) Suggestions to clarify labeling of interactomes, volcano plots and heat maps:
a) Novel vs Known is clearer than Interactor (blue) vs STRING/Biogrid (green). (Fig. 2A, 2B, 3A, 3B, 4A, 4B and 5C)

We have introduced the requested changes.

b) Do numbers have any significance aside from being a way to label data points on volcano plot and use heat map labels as a legend? Bait is not always first.

Numbers indicate enrichment ranks. Sometimes, bait proteins are not the most highly enriched proteins in the sample, but interacting proteins are. We now include a statement in the Figure legend of the volcano plots. It reads:

“The results of the T-tests are represented in volcano plots, which show the protein enrichment versus the significance of the enrichment. Numbers indicate enrichment ranks with the heatmap labels of (C) serving as the legend.”

c) Numbers overlap in volcano plots in Fig. 4A and supplement. Please adjust accordingly.

We have checked carefully and corrected all overlaps, accordingly.

d) For PPI heat maps, it might be useful to add color conventions defined in volcano plot to protein labels.

We have also added color conventions of the volcano plots in the PPI heatmaps.

References:

- Bustos, D., C. E. Bakalarski, Y. Yang, J. Peng and D. S. Kirkpatrick (2012). "Characterizing ubiquitination sites by peptide-based immunoaffinity enrichment." Molecular & cellular proteomics : MCP **11**(12): 1529-1540.
- Devabhaktuni, A., S. Lin, L. Zhang, K. Swaminathan, C. G. Gonzalez, N. Olsson, S. M. Pearlman, K. Rawson and J. E. Elias (2019). "TagGraph reveals vast protein modification landscapes from large tandem mass spectrometry datasets." Nature Biotechnology **37**(4): 469-479.
- Fernandes-Alnemri, T., J.-W. Yu, P. Datta, J. Wu and E. S. Alnemri (2009). "AIM2 activates the inflammasome and cell death in response to cytoplasmic DNA." Nature **458**(7237): 509-513.
- Gaidt, M. M., T. S. Ebert, D. Chauhan, K. Ramshorn, F. Pinci, S. Zuber, F. O'Duill, J. L. Schmid-Burgk, F. Hoss, R. Buhmann, G. Wittmann, E. Latz, M. Subklewe and V. Hornung (2017). "The DNA Inflammasome in Human Myeloid Cells Is Initiated by a STING-Cell Death Program Upstream of NLRP3." Cell **171**(5): 1110-1124.e1118.
- Hansen, F. M., M. C. Tanzer, F. Brüning, I. Bludau, C. Stafford, B. A. Schulman, M. S. Robles, O. Karayel and M. Mann (2021). "Data-independent acquisition method for ubiquitinome analysis reveals regulation of circadian biology." Nature Communications **12**(1): 254.
- Hein, Marco Y., Nina C. Hubner, I. Poser, J. Cox, N. Nagaraj, Y. Toyoda, Igor A. Gak, I. Weisswange, J. Mansfeld, F. Buchholz, Anthony A. Hyman and M. Mann (2015). "A Human Interactome in Three Quantitative Dimensions Organized by Stoichiometries and Abundances." Cell **163**(3): 712-723.
- Hornung, V., A. Ablasser, M. Charrel-Dennis, F. Bauernfeind, G. Horvath, D. R. Caffrey, E. Latz and K. A. Fitzgerald (2009). "AIM2 recognizes cytosolic dsDNA and forms a caspase-1-activating inflammasome with ASC." Nature **458**(7237): 514-518.
- Hubel, P., C. Urban, V. Bergant, W. M. Schneider, B. Knauer, A. Stukalov, P. Scaturro, A. Mann, L. Brunotte, H. H. Hoffmann, J. W. Schoggins, M. Schwemmler, M. Mann, C. M. Rice and A. Pichlmair (2019). "A protein-interaction network of interferon-stimulated genes extends the innate immune system landscape." Nature Immunology **20**(4): 493-502.
- Huttlin, E. L., L. Ting, R. J. Bruckner, F. Gebreab, M. P. Gygi, J. Szpyt, S. Tam, G. Zarraga, G. Colby, K. Baltier, R. Dong, V. Guarani, L. P. Vaites, A. Ordureau, R. Rad, B. K. Erickson, M. Wühr, J. Chick, B. Zhai, D. Kolippakkam, J. Mintseris, R. A. Obar, T. Harris, S. Artavanis-Tsakonas, M. E. Sowa, P. De Camilli, J. A. Paulo, J. W. Harper and S. P. Gygi (2015). "The BioPlex Network: A Systematic Exploration of the Human Interactome." Cell **162**(2): 425-440.
- Karayel, O., F. Tonelli, S. Virreira Winter, P. E. Geyer, Y. Fan, E. M. Sammler, D. Alessi, M. Steger and M. Mann (2020). "Accurate MS-based Rab10 phosphorylation stoichiometry determination as readout for LRRK2 activity in Parkinson's disease." Molecular & Cellular Proteomics: mcp.RA120.002055.
- Keilhauer, E. C., M. Y. Hein and M. Mann (2015). "Accurate Protein Complex Retrieval by Affinity Enrichment Mass Spectrometry (AE-MS) Rather than Affinity Purification Mass Spectrometry (AP-MS)." Molecular & Cellular Proteomics : MCP **14**(1): 120-135.
- Kim, W., E. J. Bennett, E. L. Huttlin, A. Guo, J. Li, A. Possemato, M. E. Sowa, R. Rad, J. Rush, M. J. Comb, J. W. Harper and S. P. Gygi (2011). "Systematic and quantitative assessment of the ubiquitin-modified proteome." Molecular cell **44**(2): 325-340.
- Lambert, J.-P., M. Tucholska, T. Pawson and A.-C. Gingras (2014). "Incorporating DNA shearing in standard affinity purification allows simultaneous identification of both soluble and chromatin-bound interaction partners." Journal of Proteomics **100**: 55-59.
- Landström, M. (2010). "The TAK1-TRAF6 signalling pathway." The International Journal of Biochemistry & Cell Biology **42**(5): 585-589.
- Loeb, K. R. and A. L. Haas (1992). "The interferon-inducible 15-kDa ubiquitin homolog conjugates to intracellular proteins." **267**(11): 7806-7813.

Mellacheruvu, D., Z. Wright, A. L. Couzens, J.-P. Lambert, N. A. St-Denis, T. Li, Y. V. Miteva, S. Hauri, M. E. Sardu, T. Y. Low, V. A. Halim, R. D. Bagshaw, N. C. Hubner, A. al-Hakim, A. Bouchard, D. Faubert, D. Fermin, W. H. Dunham, M. Goudreault, Z.-Y. Lin, B. G. Badillo, T. Pawson, D. Durocher, B. Coulombe, R. Aebersold, G. Superti-Furga, J. Colinge, A. J. R. Heck, H. Choi, M. Gstaiger, S. Mohammed, I. M. Cristea, K. L. Bennett, M. P. Washburn, B. Raught, R. M. Ewing, A.-C. Gingras and A. I. Nesvizhskii (2013). "The CRAPome: a contaminant repository for affinity purification–mass spectrometry data." Nature Methods **10**: 730.

Pankow, S., C. Bamberger and J. R. Yates (2019). "A posttranslational modification code for CFTR maturation is altered in cystic fibrosis." Science Signaling **12**(562): eaan7984.

Rees, J. S., N. Lowe, I. M. Armean, J. Roote, G. Johnson, E. Drummond, H. Spriggs, E. Ryder, S. Russell, D. St Johnston and K. S. Lilley (2011). "In vivo analysis of proteomes and interactomes using Parallel Affinity Capture (iPAC) coupled to mass spectrometry." Molecular & cellular proteomics : MCP **10**(6): M110.002386-M002110.002386.

Samavarchi-Tehrani, P., H. Abdouni, R. Samson and A.-C. Gingras (2018). "A Versatile Lentiviral Delivery Toolkit for Proximity-dependent Biotinylation in Diverse Cell Types *." Molecular & Cellular Proteomics **17**(11): 2256-2269.

Sato, S., H. Sanjo, K. Takeda, J. Ninomiya-Tsuji, M. Yamamoto, T. Kawai, K. Matsumoto, O. Takeuchi and S. Akira (2005). "Essential function for the kinase TAK1 in innate and adaptive immune responses." Nature Immunology **6**(11): 1087-1095.

Stutz, A., C.-C. Kolbe, R. Stahl, G. L. Horvath, B. S. Franklin, O. van Ray, R. Brinkschulte, M. Geyer, F. Meissner and E. Latz (2017). "NLRP3 inflammasome assembly is regulated by phosphorylation of the pyrin domain." The Journal of experimental medicine **214**(6): 1725-1736.

Trinkle-Mulcahy, L., Y. W. Boulon S Fau - Lam, R. Lam Yw Fau - Urcia, F.-M. Urcia R Fau - Boisvert, F. Boisvert Fm Fau - Vandermoere, N. A. Vandermoere F Fau - Morrice, S. Morrice Na Fau - Swift, U. Swift S Fau - Rothbauer, H. Rothbauer U Fau - Leonhardt, A. Leonhardt H Fau - Lamond and A. Lamond "Identifying specific protein interaction partners using quantitative mass spectrometry and bead proteomes." (1540-8140 (Electronic)).

Wagner, S. A., P. Beli, B. T. Weinert, M. L. Nielsen, J. Cox, M. Mann and C. Choudhary (2011). "A proteome-wide, quantitative survey of in vivo ubiquitylation sites reveals widespread regulatory roles." Molecular & cellular proteomics : MCP **10**(10): M111.013284-M013111.013284.

Wiśniewski, J. R., M. Y. Hein, J. Cox and M. Mann (2014). "A "Proteomic Ruler" for Protein Copy Number and Concentration Estimation without Spike-in Standards." Molecular & Cellular Proteomics **13**(12): 3497.

Zhang, D. and D.-E. Zhang (2011). "Interferon-stimulated gene 15 and the protein ISGylation system." Journal of interferon & cytokine research : the official journal of the International Society for Interferon and Cytokine Research **31**(1): 119-130.

Zhang, Y., U. Werling and W. Edelmann (2012). "SLICE: a novel bacterial cell extract-based DNA cloning method." Nucleic Acids Research **40**(8): e55-e55.

1st Jul 2021

Manuscript Number: MSB-2020-10125R

Title: Identification of Covalent Modifications Regulating Immune Signaling Complex Composition and Phenotype

Author: Annika Frauenstein

Stefan Ebner

Fynn Hansen

Ankit Sinha

Phulphagar Kshiti

Kirby Swatek

Daniel Hornburg

Matthias Mann

Felix Meissner

Dear Dr Meissner,

Thank you for sending us your revised manuscript. We have now heard back from the reviewer who agreed to re-evaluate your study. The reviewer thinks that the concerns raised by all three reviewers have been convincingly addressed and that the study is now suitable for publication.

Before we can formally accept your manuscript, we would ask you to address the following editorial-level issues:

1. Please reduce the keyword number to five.
2. Please rename "Declaration of interests" into "Conflicts of interest".
3. Remove "data not shown". As per our guidelines, on "Unpublished Data" the journal does not permit citation of "Data not shown". All data referred to in the paper should be displayed in the main or Expanded View figures.
4. Data availability: I noticed that the datasets are not yet public. Please make sure that they will be made publicly accessible upon acceptance of the manuscript. Please remove the reviewer password, as the datasets will be made public.
5. Please remove yellow highlights from the manuscript and the appendix file.
6. I have slightly modified the synopsis text (see attached). Please let me know if it is fine like this or if you would like to introduce further modifications.
7. Our data editors have seen the manuscript, and they have made some comments and suggestions that need to be addressed (see attached). Please send back a revised version (in track change mode), as we will need to go through the changes.

Please resubmit your revised manuscript online, with a covering letter listing amendments and responses to each point raised by the referees. Please resubmit the paper ****within one month****

and ideally as soon as possible. If we do not receive the revised manuscript within this time period, the file might be closed and any subsequent resubmission would be treated as a new manuscript. Please use the Manuscript Number (above) in all correspondence.

When you resubmit your manuscript, please download our CHECKLIST (<https://bit.ly/EMBOPressAuthorChecklist>) and include the completed form in your submission. *Please note* that the Author Checklist will be published alongside the paper as part of the transparent process (<https://www.embopress.org/page/journal/17444292/authorguide#transparentprocess>)

Click on the link below to submit your revised paper.

Link Not Available

I look forward to receiving your revised manuscript soon.

Kind regards,
Jingyi

Jingyi Hou
Editor
Molecular Systems Biology

If you do choose to resubmit, please click on the link below to submit the revision online before 31st Jul 2021.

IMPORTANT: When you send your revision, we will require the following items:

1. the manuscript text in LaTeX, RTF or MS Word format
2. a letter with a detailed description of the changes made in response to the referees. Please specify clearly the exact places in the text (pages and paragraphs) where each change has been made in response to each specific comment given
3. three to four 'bullet points' highlighting the main findings of your study
4. a short 'blurb' text summarizing in two sentences the study (max. 250 characters)
5. a 'thumbnail image' (550px width and max 400px height, Illustrator, PowerPoint or jpeg format), which can be used as 'visual title' for the synopsis section of your paper.
6. Please include an author contributions statement after the Acknowledgements section (see <https://www.embopress.org/page/journal/17444292/authorguide#manuscriptpreparation>)
7. Please complete the CHECKLIST available at (<https://bit.ly/EMBOPressAuthorChecklist>). Please note that the Author Checklist will be published alongside the paper as part of the transparent process (<https://www.embopress.org/page/journal/17444292/authorguide#transparentprocess>).
8. Please note that corresponding authors are required to supply an ORCID ID for their name upon submission of a revised manuscript (EMBO Press signed a joint statement to encourage ORCID

adoption) (<https://www.embopress.org/page/journal/17444292/authorguide#editorialprocess>).

Currently, our records indicate that the ORCID for your account is 0000-0003-1000-7989.

Link Not Available

The system will prompt you to fill in your funding and payment information. This will allow Wiley to send you a quote for the article processing charge (APC) in case of acceptance. This quote takes into account any reduction or fee waivers that you may be eligible for. Authors do not need to pay any fees before their manuscript is accepted and transferred to the publisher.

*** PLEASE NOTE *** As part of the EMBO Press transparent editorial process initiative (see our Editorial at <https://dx.doi.org/10.1038/msb.2010.72> , Molecular Systems Biology will publish online a Review Process File to accompany accepted manuscripts. When preparing your letter of response, please be aware that in the event of acceptance, your cover letter/point-by-point document will be included as part of this File, which will be available to the scientific community. More information about this initiative is available in our Instructions to Authors. If you have any questions about this initiative, please contact the editorial office (msb@embo.org).

Reviewer #1:

I went carefully over the entire manuscript again, including all the new data and the rebuttal by the authors.

While the manuscript is complex, I believe the authors have made an excellent effort to address the comments from previous reviews. While more work needs to be done in the future, this manuscript should be of interest to the MSB community. the quality of the data is high and the manuscript is certainly in line with recent PPI papers published in Nature and Cell from some of the groups that were specifically mentioned by some of the other reviewers.

I support the acceptance and publication of this paper in MSB.

The authors performed the requested editorial changes.

8th Jul 2021

Manuscript number: MSB-2020-10125RR

Title: Identification of Covalent Modifications Regulating Immune Signaling Complex Composition and Phenotype

Thank you again for sending us your revised manuscript. We are now satisfied with the modifications made and I am pleased to inform you that your paper has been accepted for publication.

*** PLEASE NOTE *** As part of the EMBO Publications transparent editorial process initiative (see our Editorial at <https://dx.doi.org/10.1038/msb.2010.72>), Molecular Systems Biology publishes online a Review Process File with each accepted manuscripts. This file will be published in conjunction with your paper and will include the anonymous referee reports, your point- by-point response and all pertinent correspondence relating to the manuscript. If you do NOT want this File to be published, please inform the editorial office at msb@embo.org within 14 days upon receipt of the present letter.

Should you be planning a Press Release on your article, please get in contact with msb@wiley.com as early as possible, in order to coordinate publication and release dates.

LICENSE AND PAYMENT:

All articles published in Molecular Systems Biology are fully open access: immediately and freely available to read, download and share.

Molecular Systems Biology charges an article processing charge (APC) to cover the publication costs. You, as the corresponding author for this manuscript, should have already received a quote with the article processing fee separately.

Please let us know in case this quote has not been received.

Once your article is at Wiley for editorial production you will receive an email from Wiley's Author Services system, which will ask you to log in and will present you with the publication license form for completion. Within the same system the publication fee can be paid by credit card, an invoice or pro forma can be requested.

Payment of the publication charge and the signed Open Access Agreement form must be received before the article can be published online.

Molecular Systems Biology articles are published under the Creative Commons licence CC BY, which facilitates the sharing of scientific information by reducing legal barriers, while mandating attribution of the source in accordance to standard scholarly practice.

Proofs will be forwarded to you within the next 2-3 weeks.

Thank you very much for submitting your work to Molecular Systems Biology.

Kind regards,
Jingyi

Jingyi Hou
Editor
Molecular Systems Biology

Corresponding Author Name: Felix Meissner

Journal Submitted to: MSB

Manuscript Number: MSB-2020-10125